# Near-Optimal Online Learning for Multi-Agent Submodular Coordination: Tight Approximation and Communication Efficiency

**Qixin Zhang**[1]    **Zongqi Wan**[4]    **Yu Yang**[2]    **Li Shen**[3*]    **Dacheng Tao**[1*]

[1]Nanyang Technological University, 639798, Singapore    [2]City University of Hong Kong, 518057, China
[3]Sun Yat-sen University, Shenzhen 518107, China    [4]Chinese Academy of Sciences, Beijing 100190, China
`qxzhang4-c@my.cityu.edu.hk; wanzongqi20s@ict.ac.cn; yuyang@cityu.edu.hk`
`mathshenli@gmail.com; dacheng.tao@gmail.com`

## Abstract

Coordinating multiple agents to collaboratively maximize submodular functions in unpredictable environments is a critical task with numerous applications in machine learning, robot planning and control. The existing approaches, such as the OSG algorithm, are often hindered by their poor approximation guarantees and the rigid requirement for a fully connected communication graph. To address these challenges, we firstly present a **MA-OSMA** algorithm, which employs the multi-linear extension to transfer the discrete submodular maximization problem into a continuous optimization, thereby allowing us to reduce the strict dependence on a complete graph through consensus techniques. Moreover, **MA-OSMA** leverages a novel surrogate gradient to avoid sub-optimal stationary points. To eliminate the computationally intensive projection operations in **MA-OSMA**, we also introduce a projection-free **MA-OSEA** algorithm, which effectively utilizes the KL divergence by mixing a uniform distribution. Theoretically, we confirm that both algorithms achieve a regret bound of $\widetilde{O}(\sqrt{\frac{C_T T}{1-\beta}})$ against a $(\frac{1-e^{-c}}{c})$-approximation to the best comparator in hindsight, where $C_T$ is the deviation of maximizer sequence, $\beta$ is the spectral gap of the network and $c$ is the joint curvature of submodular objectives. This result significantly improves the $(\frac{1}{1+c})$-approximation provided by the state-of-the-art OSG algorithm. Finally, we demonstrate the effectiveness of our proposed algorithms through simulation-based multi-target tracking.

## 1 Introduction

Recent years have witnessed an upsurge in research focused on leveraging submodular functions to coordinate the actions of multiple agents in accomplishing tasks that are spatially distributed. A compelling example is the dynamic deployment of mobile sensors, particularly unmanned aerial vehicles (UAVs), for multi-target tracking (Zhou et al., 2018; Corah & Michael, 2021) as depicted in Figure 1. In this scenario, at each critical moment of decision, every mobile sensor needs to determine its trajectory and velocity through interactions with others to effectively track all moving points of interest. The primary challenges of this tracking challenge lie in the unpredictability of the targets' movements and the limited sensing capabilities of agents. To address these issues, various modeling techniques have been developed, including one based on dynamically maximizing a sequence of submodular functions that capture the spatial relationship between sensors and moving targets (Xu et al., 2023; Rezazadeh & Kia, 2023; Robey et al., 2021). As a result, the problem of target tracking can be cast into a specific instance of multi-agent online submodular maximization(MA-OSM) problem. Besides target tracking, the MA-OSM problem also offers a versatile framework for a variety of complex tasks such as area monitoring (Schlotfeldt et al., 2021; Li et al., 2023), environmental mapping (Atanasov et al., 2015; Liu et al., 2021), data summarization (Mirzasoleiman et al., 2016a;b) and task assignment (Qu et al., 2019). Motivated by these practical use cases, this paper delves into the multi-agent online submodular maximization problem.

---

*Corresponding Authors

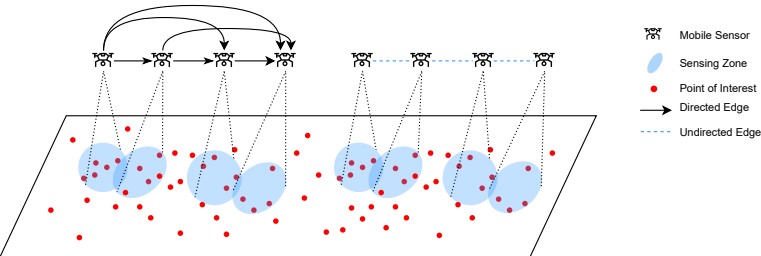

Figure 1: Left: Multi-target tracking with $4$ mobile sensors over a *complete* directed acyclic communication network. Right: Multi-target tracking with $4$ sensors over a *connected* undirected graph.

To tackle the aforementioned MA-OSM problem, Xu et al. (2023) have recently proposed an *online sequential greedy* (OSG) algorithm, building upon the foundations of the classical greedy method (Fisher et al., 1978). Nevertheless, this online algorithm suffers from two notable limitations: **i)** **Sub-optimal Approximation:** In contrast with the tight $(\frac{1-e^{-c}}{c})$-approximation ratio (Vondrák, 2010; Bian et al., 2017a), OSG only can guarantee a sub-optimal $(\frac{1}{1+c})$-approximation where $c \in [0, 1]$ is the joint curvature of submodular objectives; **ii) Requirement of a Fully Connected Communication Network:** OSG begins by assigning a unique order to each agent and then requires every agent to have full access to the decisions made by all predecessors, which leads to a *complete* directed acyclic communication graph (Refer to the left side of Figure 1). As the number of agents grows, the communication overheads associated with this operation may become prohibitively high. Furthermore, Grimsman et al. (2018) have pointed out that the approximation guarantee of OSG continuously degrades as the communication graph becomes less dense. This highlights the necessity of a *complete* communication graph for maintaining the effectiveness of OSG. Given these disadvantages of OSG algorithm, the objective of this paper is to address the following question:

*Is it possible to devise an online algorithm with tight $(\frac{1-e^{-c}}{c})$-approximation for* MA-OSM *problem over a connected and sparse communication network?*

In this paper, we provide an affirmative answer to this question by presenting two online algorithms, i.e., **MA-OSMA** and **MA-OSEA**, both of which not only can achieve the optimal $(\frac{1-e^{-c}}{c})$-approximation guarantee but also reduce the strict requirement for a *complete* communication graph.

Specifically, our proposed algorithms incorporate three key innovations. First, we utilize the multi-linear extension to convert the discrete submodular maximization into a continuous optimization problem, which enables us to reduce the rigid requirement for a complete communication graph via the well-established consensus techniques in the field of decentralized optimization. Second, we develop a surrogate function for the multi-linear extension of submodular functions with curvature $c$, which empowers us to move beyond the sub-optimal $(\frac{1}{1+c})$-approximation stationary points. Last but not least, for each agent, we implement a distinct strategy to update the selected probabilities associated with its own actions and those of other agents, which only requires agents to assess the marginal gains of actions within their own action sets, thereby reducing the practical requirement on the observational scope of each agent. To summarize, we make the following contributions.

- We construct a surrogate function for the multi-linear extension of submodular functions with curvature $c \in [0, 1]$. The stationary points of this surrogate can guarantee a tight $(\frac{1-e^{-c}}{c})$-approximation to the maximum value of the multi-linear extension, significantly outperforming the $(\frac{1}{1+c})$-approximation provided by the stationary points of the original multi-linear extension itself.

- We propose a new algorithm **MA-OSMA**, which seamlessly integrates consensus techniques, lossless rounding and the surrogate function previously discussed. Moreover, we prove that **MA-OSMA** enjoys a regret bound of $O\left(\sqrt{\frac{C_T T}{1-\beta}}\right)$ against a $(\frac{1-e^{-c}}{c})$-approximation to the best comparator in hindsight, where $C_T$ is the deviation of maximizer sequence and $\beta$ is the spectral gap of the communication network. Subsequently, we present a *projection-free* variant of **MA-OSMA**, titled **MA-OSEA**, which effectively utilizes the KL divergence by mixing a uniform distribution. We also prove that **MA-OSEA** can attain a $(\frac{1-e^{-c}}{c})$-regret bound of $\widetilde{O}\left(\sqrt{\frac{C_T T}{1-\beta}}\right)$. A detailed comparison of our **MA-OSMA** and **MA-OSEA** with existing studies is presented in Table 1.

| Method | Approx.Ratio | Graph($G$) | Regret | Projection-free? | Reference |
|--------|-------------|-----------|--------|------------------|-----------|
| OSG | $\left(\frac{1}{1+c}\right)$ | **complete** | $\widetilde{O}\left(\sqrt{C_T T}\right)$ | ✔ | Xu et al. (2023) |
| OSG | $\left(\frac{1}{1+\alpha(G)}\right)$ | connected | $\widetilde{O}\left(\sqrt{C_T T}\right)$ | ✔ | Grimsman et al. (2018); Xu et al. (2023) |
| **MA-OSMA** | $\left(\frac{1-e^{-c}}{c}\right)$ | connected | $O\left(\sqrt{\frac{C_T T}{1-\beta}}\right)$ | ✘ | Theorem 3 & Remark 8 |
| **MA-OSEA** | $\left(\frac{1-e^{-c}}{c}\right)$ | connected | $\widetilde{O}\left(\sqrt{\frac{C_T T}{1-\beta}}\right)$ | ✔ | Theorem 5 & Remark 9 |

Table 1: Comparison with prior works. $T$ is the horizon length, $c \in [0, 1]$ is the joint curvature of submodular objectives, $C_T$ is the deviation of maximizer sequence, $\beta$ is the second largest magnitude of the eigenvalues of the weight matrix, $\alpha(G)$ is the number of nodes in the largest independent set in communication graph $G$ where $\alpha(G) \geq 1$ and $\widetilde{O}(\cdot)$ hides $\log(T)$ term.

- We conduct a simulation-based evaluation of our proposed algorithms within a multi-target tracking scenario. Our experiments demonstrate the effectiveness of our **MA-OSMA** and **MA-OSEA**.

**Related Work.** Due to space limits, we only focus on the most relevant studies. A more comprehensive discussion is provided in Appendix A.1. Multi-agent submodular maximization (MA-SM) problem involves coordinating multiple agents to collaboratively maximize a submodular utility function, with numerous applications in sensor coverage (Krause et al., 2008; Prajapat et al., 2022) and multi-robot planning (Singh et al., 2009; Zhou & Tokekar, 2022). A commonly used solution for MA-SM problem heavily depends on the distributed implementation of the classic *sequential greedy* method (Fisher et al., 1978), which can ensure a $\left(\frac{1}{1+c}\right)$-approximation (Conforti & Cornuéjols, 1984). However, this distributed algorithm requires each agent to have full access to the decisions of all previous agents, thereby forming a *complete* directed communication graph. Subsequently, several studies (Grimsman et al., 2018; Gharesifard & Smith, 2017; Marden, 2016) have investigated how the topology of the communication network affects the performance of the distributed greedy method. Particularly, Grimsman et al. (2018) pointed out that the worst-case performance of the distributed greedy algorithm will deteriorate in proportion to the size of the largest independent group of agents in the communication graph. Given that the majority of applications occur in time-varying environments, Xu et al. (2023) proposed the *online sequence greedy*(OSG) algorithm for online MA-SM problem, which also ensures a sub-optimal $\left(\frac{1}{1+c}\right)$-approximation over a *complete* communication graph.

## 2 PRELIMINARIES AND PROBLEM FORMULATION

**Notations.** Throughout this paper, $\mathbb{R}$ and $\mathbb{R}_+$ denote the set of real numbers and non-negative real numbers, respectively. For any positive integer $K$, $[K]$ stands for the set $\{1, \ldots, K\}$. Let $\|\cdot\|$ represent a norm for vectors and its dual norm be denoted by $\|\cdot\|_*$. Specially, $\|\cdot\|_1$ and $\|\cdot\|_2$ denote the $l_1$ norm and $l_2$ norm for vectors, respectively. $\langle\cdot,\cdot\rangle$ denotes the inner product. The lowercase boldface (e.g. $\mathbf{x}$) denotes a column vector with a suitable dimension and the uppercase boldface (e.g. $\mathbf{W}$) for a matrix. The $i$-th component of a vector $\mathbf{x}$ will be denoted $x_i$ and the element in the $i$-th row of the $j$-th column of a matrix $\mathbf{W}$ will be denoted by $w_{ij}$. Moreover, $\lambda_i(\mathbf{W})$ denotes the $i$-th largest eigenvalue of matrix $\mathbf{W}$. $\mathbf{I}_n$ and $\mathbf{1}_n$ represent the identity matrix and the $n$-dimensional vector whose all entries are 1, respectively. Additionally, for any vector $\mathbf{x} \in \mathbb{R}^n$ and $S \subseteq [n]$, the $[\mathbf{x}]_S$ denotes the projection of $\mathbf{x}$ onto the set $S$, i.e., $[\mathbf{x}]_S = (x_{i_1}, \ldots, x_{i_{|S|}}) \in \mathbb{R}^s$ for any $S = \{i_1, \ldots, i_{|S|}\} \subseteq [n]$.

**Submodularity and curvature.** Let $\mathcal{V}$ be a finite set and $f : 2^V \to \mathbb{R}_+$ be a set function mapping subsets of $\mathcal{V}$ to the non-negative real line. The function $f$ is said to be submodular iff $f(S \cup \{e\}) - f(S) \geq f(T \cup \{e\}) - f(T)$ for any $S \subseteq T \subseteq V$ and $e \in V \setminus T$. In this paper, we will consider submodular functions that are *monotone*, meaning that for any $S \subseteq T \subseteq V$, $f(S) \leq f(T)$, and *normalized*, that is, $f(\emptyset) = 0$. To better reflect the diminishing returns property of submodular functions, Conforti & Cornuéjols (1984) introduced the concept of *curvature*, which is defined as $c := 1 - \min_{S \subseteq \mathcal{V}, e \notin S} \frac{f(S \cup \{e\}) - f(S)}{f(\{e\})}$. Moreover, we can infer $c \in [0, 1]$ for submodular functions.

### 2.1 PROBLEM FORMULATION

In this subsection, we introduce the multi-agent online submodular maximization problem, commonly abbreviated as multi-agent submodular coordination.

In MA-OSM, we generally consider a group of $N$ different agents denoted as $\mathcal{N} = \{1, 2, \ldots, N\}$, interacting over a connected communication graph $G(\mathcal{N}, \mathcal{E})$. In addition, each agent $i$ within $\mathcal{N}$ is equipped with a *unique* and *discrete* set of actions, denoted by $\mathcal{V}_i$. This implies that these action sets are mutually disjoint, i.e., $\mathcal{V}_i \cap \mathcal{V}_j = \emptyset$ for any two distinct agents $i \in \mathcal{N}$ and $j \in \mathcal{N}$. At each time step $t \in [T]$, every agent $i \in \mathcal{N}$ separately selects an action $a_{t,i}$ from the individual action set $\mathcal{V}_i$. After committing to these choices, the environment reveals a monotone submodular function $f_t$ defined over the aggregated action space $\mathcal{V} := \cup_{i \in \mathcal{N}} \mathcal{V}_i$. Then, the agents receive the utility $f_t(\cup_{i \in \mathcal{N}} \{a_{t,i}\})$. As a result, the objective of agents at any given moment is to maximize their collective gains as much as possible, that is to say, we need to solve the following submodular maximization problem in a multi-agent manner at each round:

$$\max f_t(\mathcal{A}), \quad \text{s.t.} \quad |\mathcal{A} \cap \mathcal{V}_i| \leq 1, \forall i \in \mathcal{N}. \tag{1}$$

Compared to the standard *centralized* submodular maximization problem, this MA-OSM problem brings additional challenges: **1) Unpredictable Objectives and Actions:** Agents must make decisions at each moment without prior knowledge of future submodular utility functions and the insight into other agents' actions; **2) Limited Feedback:** In many real-world scenarios, each agent is typically endowed with a narrow perceptual or detection scope, which only allows it to sense the environmental changes within its surroundings. For instance, in the target tracking problem of Figure 1, every sensor usually overlooks these targets beyond its sensing circle. Broadly speaking, the local information observed by one agent is inadequate for precisely assessing the actions of most other agents who are not in close vicinity. To capture this, various studies (Xu et al., 2023; Rezazadeh & Kia, 2023; Robey et al., 2021; Qu et al., 2019) related to MA-OSM problem commonly confine each agent $i \in \mathcal{N}$ to a local marginal gain oracle $\mathcal{O}_t^i : \mathcal{V}_i \times 2^{\mathcal{V}} \to \mathbb{R}_+$ after $f_t$ is revealed, where $\mathcal{O}_t^i(a, \mathcal{A}) := f_t(\mathcal{A} \cup \{a\}) - f_t(\mathcal{A})$ for any $a \in \mathcal{V}_i$ and $\mathcal{A} \subseteq \mathcal{V}$. This restriction means that, at each time $t \in [T]$, agents only can receive the limited feedback about the marginal evaluations of actions within their own action set, rather than the full information of $f_t$. In this paper, we also impose this restriction on each agent.

Given the NP-hardness of maximizing a submodular function subject to a general constraint (Vondrák, 2013; Bian et al., 2017b), we adopt the dynamic $\alpha$-*regret* to evaluate the algorithm performance for MA-OSM problem in this paper, which is defined as follows (Kakade et al., 2007; Streeter & Golovin, 2008; Chen et al., 2018):

$$\mathbf{Reg}_\alpha^d(T) = \alpha \sum_{t=1}^{T} f_t(\mathcal{A}_t^*) - \sum_{t=1}^{T} f_t(\cup_{i \in \mathcal{N}} \{a_{t,i}\}),$$

where $\mathcal{A}_t^*$ is the maximizer of Eq.(1) and $a_{t,i}$ is the action taken via the agent $i \in \mathcal{N}$ at time $t \in [T]$.

## 3 MULTI-LINEAR EXTENSION AND ITS PROPERTIES

Compared to discrete optimization, continuous optimization has a plethora of efficient tools and algorithmic frameworks. As a result, a common approach in discrete optimization is based on a continuous relaxation to embed the corresponding discrete problem into a solvable continuous optimization. In the subsequent section, we will present a canonical relaxation technique for submodular functions, known as *multi-linear extension* (Calinescu et al., 2011; Chekuri et al., 2014). To better illustrate this extension, we suppose $|\mathcal{V}| = n$ and set $\mathcal{V} := [n] = \{1, \ldots, n\}$ throughout this paper.

**Definition 1.** *For a set function $f : 2^{\mathcal{V}} \to \mathbb{R}_+$, we define its multi-linear extension as*

$$F(\mathbf{x}) = \sum_{\mathcal{A} \subseteq \mathcal{V}} \left( f(\mathcal{A}) \prod_{a \in \mathcal{A}} x_a \prod_{a \notin \mathcal{A}} (1 - x_a) \right) = \mathbb{E}_{\mathcal{R} \sim \mathbf{x}} \left( f(\mathcal{R}) \right), \tag{2}$$

*where $\mathbf{x} = (x_1, \ldots, x_n) \in [0, 1]^n$ and $\mathcal{R} \subseteq \mathcal{V}$ is a random set that contains each element $a \in \mathcal{V}$ independently with probability $x_a$ and excludes it with probability $1 - x_a$. We write $\mathcal{R} \sim \mathbf{x}$ to denote that $\mathcal{R} \subseteq \mathcal{V}$ is a random set sampled according to $\mathbf{x}$.*

From the Eq.(2), we can view multi-linear extension at any point $\mathbf{x} \in [0, 1]^n$ as the expected utility of independently selecting each action $a \in \mathcal{V}$ with probability $x_a$. With this tool, we can cast the previous discrete problem Eq.(1) into a continuous maximization which learns the selected probability for each action $a \in \mathcal{V}$, that is, for any $t \in [T]$, we consider the following continuous optimization:

$$\max_{\mathbf{x} \in [0,1]^n} F_t(\mathbf{x}), \quad \text{s.t.} \quad \sum_{a \in \mathcal{V}_i} x_a \leq 1, \forall i \in \mathcal{N}, \tag{3}$$

where $F_t(\mathbf{x})$ is the multi-linear extension of $f_t$. When $f_t$ is submodular, the maximization problem Eq.(3) is both non-convex and non-concave (Bian et al., 2020). Thanks to recent advancements in optimizing complex neural networks, a large body of empirical and theoretical evidence has shown that numerous gradient-based algorithms, such as projected gradient methods and Frank Wolfe, can efficiently address the general non-convex or non-concave problem. Specifically, under certain mild assumptions, many first-order gradient algorithms can converge to a stationary point of the corresponding non-convex or non-concave objective (Nesterov, 2013; Lacoste-Julien, 2016; Jin et al., 2017; Agarwal et al., 2017; Hassani et al., 2017). Motivated by these findings, we proceed to investigate the stationary points of the multi-linear extension of submodular functions.

### 3.1 CHARACTERIZING STATIONARY POINTS

We begin with the definition of a stationary point for maximization problems.

**Definition 2.** *A vector $\mathbf{x} \in \mathcal{C}$ is called a stationary point for the differentiable function $G : [0,1]^n \to \mathbb{R}_+$ over the domain $\mathcal{C} \subseteq [0,1]^n$ if $\max_{\mathbf{y} \in \mathcal{C}} \langle \mathbf{y} - \mathbf{x}, \nabla G(\mathbf{x}) \rangle \leq 0$.*

Stationary points are of great interest as they characterize the fixed points of a multitude of gradient-based methods. Next, we quantify the performance of the stationary points of multi-linear extension relative to the maximum value, i.e.,

**Theorem 1** (Proof is deferred to Appendix B). *If $f : 2^{\mathcal{V}} \to \mathbb{R}_+$ is a monotone submodular function with curvature $c$, then for any stationary point $\mathbf{x}$ of its multi-linear extension $F : [0,1]^n \to \mathbb{R}_+$ over domain $\mathcal{C} \subseteq [0,1]^n$, we have*

$$F(\mathbf{x}) \geq \left( \frac{1}{1+c} \right) \max_{\mathbf{y} \in \mathcal{C}} F(\mathbf{y}).$$

**Remark 1.** *The ratio $\frac{1}{1+c}$ is tight for the stationary points of the multi-linear extension of submodular function with curvature $c$, because there exists a special instance of multi-linear extension with a $\left(\frac{1}{2}\right)$-approximation stationary point when $c = 1$ (Hassani et al., 2017).*

Theorem 1 suggests that applying various gradient-based methods directly to multi-linear extension only can ensure a $\frac{1}{1+c}$-approximation guarantee. However, the best-known approximation ratio for maximizing a monotone submodular function with curvature $c$ is $\frac{1-e^{-c}}{c}$ (Vondrák, 2010; Bian et al., 2017a). As depicted in Figure 2, there exists a non-negligible gap between $\frac{1}{1+c}$ and $\frac{1-e^{-c}}{c}$. The question arises: *Is it feasible to bridge this significant gap?* Recently, numerous studies have successfully leveraged a classic technique named *Non-Oblivious Search* (Alimonti, 1994; Khanna et al., 1998; Filmus & Ward, 2012; 2014) to output superior solutions by constructing an effective surrogate function. Inspired by this idea, we also

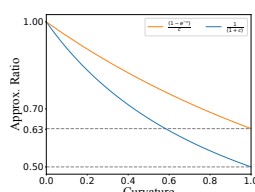

Figure 2: $\frac{1}{1+c}$ v.s. $\frac{1-e^{-c}}{c}$.

aspire to devise a surrogate function that can enhance the approximation guarantees for the stationary points of multi-linear extension. In line with the works (Zhang et al., 2022; 2024; Wan et al., 2023), we consider a type of surrogate function $F^s(\mathbf{x})$ whose gradient at point $\mathbf{x}$ assigns varying weights to the gradient of multi-linear extension at $z * \mathbf{x}$, given by $\nabla F^s(\mathbf{x}) = \int_0^1 w(z) \nabla F(z * \boldsymbol{x}) \mathrm{d}z$ where $w(z)$ is the positive weight function over $[0,1]$ and $*$ denotes the multiplication of scalars and vectors. After carefully selecting the weight function $w(z)$, we can have that:

**Theorem 2** (Proof is deferred to Appendix C). *If the weight function $w(z) = e^{c(z-1)}$ and the function $F : [0,1]^n \to \mathbb{R}_+$ is the multi-linear extension of a monotone submodular function $f : 2^{\mathcal{V}} \to \mathbb{R}_+$ with curvature $c$, we have, for any $\mathbf{x}, \mathbf{y} \in [0,1]^n$,*

$$\langle \mathbf{y} - \mathbf{x}, \nabla F^s(\mathbf{x}) \rangle = \left\langle \mathbf{y} - \mathbf{x}, \int_0^1 e^{c(z-1)} \nabla F(z * \mathbf{x}) \mathrm{d}z \right\rangle \geq \left( \frac{1-e^{-c}}{c} \right) F(\mathbf{y}) - F(\mathbf{x}). \quad (4)$$

**Remark 2.** *Theorem 2 illustrate that the stationary points of surrogate function $F^s(\mathbf{x})$ can provide a better $\left( \frac{1-e^{-c}}{c} \right)$-approximation than the stationary points of the original multi-linear extension $F$.*

**Remark 3.** *Unlike prior work on surrogate functions regarding the multi-linear extension of submodular functions (Zhang et al., 2022; 2024), Theorem 2 takes into account the impact of curvature. Specifically, when the curvature $c = 1$, our result Eq.(4) is consistent with those of Zhang et al. (2022; 2024). To the best of our knowledge, we are the first to explore the stationary points of the multi-linear extension of submodular functions with different curvatures.*

## 3.2 Constructing an Unbiased Gradient for surrogate function

In this subsection, we present how to estimate the gradient $\nabla F^s(\mathbf{x}) = \int_0^1 e^{c(z-1)} \nabla F(z * \boldsymbol{x}) \mathrm{d}z$ using the function values of $f$. Given that $F$ is the multi-linear extension of set function $f$, we can show $\frac{\partial F(\mathbf{x})}{\partial x_i} = \mathbb{E}_{\mathcal{R} \sim \mathbf{x}} \Big( f(\mathcal{R} \cup \{i\}) - f(\mathcal{R} \setminus \{i\}) \Big)$ (Calinescu et al., 2011). That is to say, the partial derivative of multi-linear extension $F$ at each variable $x_i$ equals the expected marginal contribution for the action $\{i\}$. Consequently, after sampling a random number $z$ from the probability distribution of r.v. $\mathcal{Z}$ where $P(\mathcal{Z} \leq b) = (\frac{c}{1-e^{-c}}) \int_0^b e^{c(z-1)} \mathrm{d}z = \frac{e^{c(b-1)} - e^{-c}}{1 - e^{-c}}$ for any $b \in [0, 1]$ and then generating a random set $\mathcal{R}$ according to $z * \mathbf{x}$, we can estimate $\nabla F^s(\mathbf{x})$ by the following equation:

$$\widetilde{\nabla} F^s(\mathbf{x}) = \Big( \frac{1 - e^{-c}}{c} \Big) \Big( f(\mathcal{R} \cup \{1\}) - f(\mathcal{R} \setminus \{1\}), \dots, f(\mathcal{R} \cup \{n\}) - f(\mathcal{R} \setminus \{n\}) \Big) \quad (5)$$

## 4 Methodology

The mirror method, a sophisticated optimization framework, utilizes the notion of Bregman divergence in lieu of Euclidean distance for the projection step, thereby unifying a spectrum of first-order algorithms (Nemirovsky & Yudin, 1983). In this section, we present two multi-agent variants of the online mirror ascent (Hazan et al., 2016; Jadbabaie et al., 2015; Shahrampour & Jadbabaie, 2017), which is specifically crafted to tackle the MA-OSM problem introduced in Section 2.1.

### 4.1 Multi-Agent Online Surrogate Mirror Ascent

Given the core role of Bregman divergence in the mirror ascent method, we begin with an in-depth review of this concept, that is,

**Definition 3** (Bregman Divergence). *Let $\phi : \Omega \to \mathbb{R}$ is a continuously-differentiable, 1-strongly convex function defined on a convex set $\Omega \subseteq [0, 1]^n$. Then the Bregman divergence with respect to $\phi$ is defined as:*

$$\mathcal{D}_\phi(\mathbf{x}, \mathbf{y}) := \phi(\mathbf{x}) - \phi(\mathbf{y}) - \langle \nabla \phi(\mathbf{y}), \mathbf{x} - \mathbf{y} \rangle. \quad (6)$$

Two well-known examples of Bregman divergence include the Euclidean distance, which arises from the choice of $\phi(\mathbf{x}) = \frac{\|\mathbf{x}\|_2^2}{2}$ and the Kullback-Leibler (KL) divergence, associated with $\phi(\mathbf{x}) = \sum_{i=1}^n x_i \log(x_i)$. Note that both forms of $\phi(\mathbf{x})$ allow for a coordinate-wise decomposition. Without loss of generality, we make the following assumption.

**Assumption 1.** *$\phi(\mathbf{x})$ is dominated by a one-dimensional strongly convex differentiable function $g : [0, 1] \to \mathbb{R}$, that is, $\phi(\mathbf{x}) = \sum_{i=1}^n g(x_i)$ where $\mathbf{x} = (x_1, \dots, x_n)$.*

Under this assumption, we can re-define the Bregman divergence between two $n$-dimensional vectors $\mathbf{b}$ and $\mathbf{c}$ as: $\mathcal{D}_{g,n}(\mathbf{b}, \mathbf{c}) := \sum_{i=1}^n \Big( g(b_i) - g(c_i) - g'(c_i)(b_i - c_i) \Big)$ where $g'$ denotes the derivative of $g$. Specially, we also have $\mathcal{D}_\phi(\mathbf{x}, \mathbf{y}) = \mathcal{D}_{g,n}(\mathbf{x}, \mathbf{y})$ from Eq.(6). Building on these foundations, we now introduce the Multi-Agent Online Boosting Mirror Ascent (**MA-OSMA**) algorithm for MA-OSM problem, as detailed in Algorithm 1.

In Algorithm 1, at every time step $t \in [T]$, each agent $i \in \mathcal{N}$ maintains a local variable $\mathbf{x}_{t,i} \in [0, 1]^{|\mathcal{V}|}$, which, to some extent, reflects agent $i$'s current beliefs regarding all actions in $\mathcal{V}$. The core of **MA-OSMA** algorithm is primarily composed of four interleaved components: Rounding, Information aggregation, Surrogate gradient estimation and Probabilities update. Specifically, at every iteration $t \in [T]$, each agent $i$ first selects an action $a_{t,i}$ from $\mathcal{V}_i$ based on its current preferences $\mathbf{x}_{t,i}$. Subsequently, agent $i$ receives $\mathbf{x}_{t,j}$ from all neighboring agents and then computes the aggregated beliefs $\mathbf{y}_{t,i}$ as the weighted average of $\mathbf{x}_{t,j}$ for $j \in \mathcal{N}_i$, where $\mathcal{N}_i$ denotes the neighbors of agent $i$. Next, agent $i$ estimates the gradient of the surrogate function of the multi-linear extension of $f_t$ at each coordinate $a \in \mathcal{V}_i$ by employing the methods outlined in Section 3.2. That is, agent $i$ initially samples a random number $z_{t,i}$ from the random variable $\mathcal{Z}$, where $P(\mathcal{Z} \leq b) = \frac{e^{c(b-1)} - e^{-c}}{1 - e^{-c}}$ for any $b \in [0, 1]$, and then approximates $[\nabla F_t^s(\mathbf{x}_{t,i})]_a$ as $\frac{1 - e^{-c}}{c} \big( f_t(\mathcal{R}_{t,i} \cup \{a\}) - f_t(\mathcal{R}_{t,i} \setminus \{a\}) \big)$ for any $a \in \mathcal{V}_i$ where $\mathcal{R}_{t,i}$ is a random set according to $z_{t,i} * \mathbf{x}_{t,i}$. Finally, each agent $i$ adjusts the probabilities of actions in $\mathcal{V}_i$ through a mirror ascent along the direction $[\widetilde{\nabla} F_t^s(\mathbf{x}_{t,i})]_{\mathcal{V}_i}$. As for other actions not in $\mathcal{V}_i$, their probabilities are straightforwardly updated using the aggregated beliefs $\mathbf{y}_{t,i}$.

---

**Algorithm 1** Multi-Agent Online Surrogate Mirror Ascent (**MA-OSMA**)

---

1: **Input:** Number of iterations $T$, the set of agents $\mathcal{N}$, communication graph $G(\mathcal{N}, \mathcal{E})$, weight matrix $\mathbf{W} = [w_{ij}] \in \mathbb{R}^{N \times N}$, 1-strongly decomposable convex function $\phi(\mathbf{x}) = \sum_{i=1}^{n} g(x_i)$, the curvature $c \in [0, 1]$, step size $\eta_t$ for $t \in [T]$;

2: **Initialized:** for any agent $i \in \mathcal{N}$, let $[\mathbf{x}_{1,i}]_j = \frac{1}{|\mathcal{V}_i|}, \ \forall j \in \mathcal{V}_i$ and $[\mathbf{x}_{1,i}]_j = 0, \ \forall j \notin \mathcal{V}_i$

3: **for** $t \in [T]$ **do**

4:     **for** $i \in \mathcal{N}$ **do**

5:         Compute SUM $:= \sum_{a \in \mathcal{V}_i} [\mathbf{x}_{t,i}]_a$   ▷ Rounding (Lines 5-6)

6:         Select an action $a_{t,i}$ from the set $\mathcal{V}_i$ with probability $\frac{[\mathbf{x}_{t,i}]_a}{\text{SUM}}$

7:         Exchange $\mathbf{x}_{t,i}$ with each neighboring node $j \in \mathcal{N}_i$   ▷ Information aggregation (Lines 7-8)

8:         Aggregate the information by setting $\mathbf{y}_{t,i} = \sum_{j \in \mathcal{N}_i \cup \{i\}} w_{ij} \mathbf{x}_{t,j}$

9:         Sampling a random number $z_{t,i}$ from r.v. $\mathcal{Z}$   ▷ Surrogate gradient estimation (Lines 9-11)

10:        Sampling a random set $\mathcal{R}_{t,i} \sim z_{t,i} * \mathbf{x}_{t,i}$

11:        Compute $[\widetilde{\nabla} F_t^s(\mathbf{x}_{t,i})]_a := \frac{1 - e^{-c}}{c}\big(f_t(\mathcal{R}_{t,i} \cup \{a\}) - f_t(\mathcal{R}_{t,i} \setminus \{a\})\big)$ for any $a \in \mathcal{V}_i$

12:        Update $[\mathbf{x}_{t+1,i}]_a = [\mathbf{y}_{t,i}]_a, \ \forall a \notin \mathcal{V}_i$   ▷ Update the probabilities of actions (Lines 12-13)

13:        Update the probabilities of actions of agent $i$ itself by

$$[\mathbf{x}_{t+1,i}]_{\mathcal{V}_i} := \underset{\sum_{k=1}^{n_i} b_k \leq 1}{\arg\min} \left( -\langle [\tilde{\nabla} F_t^s(\mathbf{x}_{t,i})]_{\mathcal{V}_i}, \mathbf{b} \rangle + \frac{1}{\eta_t} \mathcal{D}_{g,n_1}(\mathbf{b}, [\mathbf{y}_{t,i}]_{\mathcal{V}_i}) \right), \qquad (7)$$

where $n_i = |\mathcal{V}_i|$ and $\mathbf{b} = (b_1, \dots, b_{n_i}) \in [0, 1]^{n_i}$

---

The key novelty of Algorithm 1 is twofold: first, it integrates a surrogate gradient estimation for the multi-linear extension of $f_t$, ensuring a tight approximation guarantee; second, it adopts a divide-and-conquer strategy to update the probabilities of all actions in Lines 12-13, which only requires agents to evaluate the marginal benefits of actions within their own action sets. These tactics not only effectively reduce the computational burden for each agent but also partially offset the practical errors caused by the limited observational capabilities of each agent.

### 4.1.1 REGRET ANALYSIS FOR ALGORITHM 1

In this subsection, we present theoretical results for the proposed method **MA-OSMA**. We begin by introducing some standard assumptions about the communication graph $G(\mathcal{N}, \mathcal{E})$, weight matrix $\mathbf{W} \in \mathbb{R}^{N \times N}$, Bregman divergence $\mathcal{D}_\phi$ and the surrogate gradient estimation $\tilde{\nabla} F_t^s$.

**Assumption 2.** *The graph $G$ is connected, i.e., there exists a path from any agent $i \in \mathcal{N}$ to any other agent $j \in \mathcal{N}$. Moreover, the weight matrix $\mathbf{W} = [w_{ij}] \in \mathbb{R}^{N \times N}$ is symmetric and doubly stochastic with positive diagonal, i.e., $\mathbf{W}^T = \mathbf{W}$ and $\mathbf{W}\mathbf{1}_N = \mathbf{1}_N$, where $N$ is the number of agents.*

**Remark 4.** *The connectivity of communication graph $G$ implies the uniqueness of $\lambda_1(\mathbf{W}) = 1$ and also warrants that other eigenvalues of $\mathbf{W}$ are strictly less than one in magnitude (Nedic & Ozdaglar, 2009; Horn & Johnson, 2012; Yuan et al., 2016). In detail, regarding the eigenvalue of $\mathbf{W}$,i.e., $1 = \lambda_1(\mathbf{W}) \geq \lambda_2(\mathbf{W}) \geq \cdots \geq \lambda_N(\mathbf{W}) \geq -1$, then $\beta < 1$,where $\beta = \max(|\lambda_2(\mathbf{W})|, |\lambda_N(\mathbf{W})|)$ is the second largest magnitude of the eigenvalues of $\mathbf{W}$.*

**Assumption 3.** *Let $\mathbf{x}$ and $\{\mathbf{y}_i\}_{i=1}^N$ be vectors in $[0,1]^n$, the Bregman divergence satisfies the separate convexity in the following sense $\mathcal{D}_\phi\left(\mathbf{x}, \sum_{i=1}^N \alpha_i \mathbf{y}_i\right) \leq \sum_{i=1}^N \mathcal{D}_\phi(\mathbf{x}, \alpha_i \mathbf{y}_i)$, where $\sum_{i=1}^N \alpha_i = 1$.*

**Remark 5.** *The separate convexity (Bauschke & Borwein, 2001) is commonly satisfied for most used cases of Bregman divergence. For example, the Euclidean distance and KL-divergence.*

**Assumption 4.** *The Bregman divergence satisfies a Lipschitz condition, i.e., there exists a constant $K$ such that for any $\mathbf{x}, \mathbf{y}, \mathbf{z} \in [0,1]^n$, we have $|\mathcal{D}_\phi(\mathbf{x}, \mathbf{z}) - \mathcal{D}_\phi(\mathbf{y}, \mathbf{z})| \leq K\|\mathbf{x} - \mathbf{y}\|$.*

**Remark 6.** *When the function $\phi$ is Lipschitz with respect to $\|\cdot\|$, the Lipschitz condition on the Bregman divergence is automatically satisfied. Thus, this assumption evidently holds for Euclidean distance. However, KL divergence is not satisfied with Assumption 4, as its gradient will approach infinity on the boundary.*

**Assumption 5.** *For any $t \in [T]$ and $\mathbf{x} \in [0,1]^n$, the stochastic gradient $\widetilde{\nabla} F_t^s(\mathbf{x})$ is bounded and unbiased, i.e., $\mathbb{E}(\widetilde{\nabla} F_t^s(\mathbf{x})|\mathbf{x}) = \nabla F_t^s(\mathbf{x})$ and $\mathbb{E}(\|\widetilde{\nabla} F_t^s(\mathbf{x})\|_*^2) \leq G^2$. Here, $\|\cdot\|_*$ is the dual norm of the general norm $\|\cdot\|$. Moreover, $F_t$ is also $L$-smooth, i.e., $\|\nabla F_t^s(\mathbf{x}) - \nabla F_t^s(\mathbf{y})\|_* \leq L\|\mathbf{x} - \mathbf{y}\|$.*

A detailed discussion regarding Assumption 5 will be presented in the Appendix A.2. Now we are ready to show the main theoretical result about Algorithm 1.

**Theorem 3** (Proof is deferred to Appendix D). *Consider our proposed Algorithm 1, if Assumption 1-5 hold and each set function $f_t$ is monotone submodular with curvature $c$ for any $t \in [T]$, then we can conclude that, when $\alpha = \frac{1-e^{-c}}{c}$,*

$$\mathbb{E}\Big(\mathbf{Reg}_\alpha^d(T)\Big) \leq C_1\Big(\sum_{t=1}^{T}\sum_{\tau=1}^{t}\beta^{t-\tau}\eta_\tau\Big) + \frac{NR^2}{\eta_{T+1}} + KNC_2\sum_{t=1}^{T}\frac{|\mathcal{A}_{t+1}^*\Delta\mathcal{A}_t^*|}{\eta_{t+1}} + \frac{NG}{2}\sum_{t=1}^{T}\eta_t, \quad (8)$$

*where $\mathcal{A}_t^*$ is any maximizer of Eq.(1), $\Delta$ is the symmetric difference of two sets, $C_1 = (4G + LDG)N^{\frac{3}{2}}$, $\|\mathbf{x}\| \leq C_2\|\mathbf{x}\|_1$ for $\mathbf{x} \in [0,1]^n$, $D = \sup_{\mathbf{x},\mathbf{y}\in\mathcal{C}} \|\mathbf{x} - \mathbf{y}\|$, $R^2 = \sup_{\mathbf{x},\mathbf{y}\in\mathcal{C}} \mathcal{D}_\phi(\mathbf{x},\mathbf{y})$, and $\mathcal{C}$ is the constraint set of Eq.(3).*

**Remark 7.** *According to the definition of symmetric difference, i.e., $S\Delta T = (S \setminus T) \cup (T \setminus S)$, we can know that the value $|\mathcal{A}_{t+1}^*\Delta\mathcal{A}_t^*|$ quantifies the deviation between the optimal strategy set at time $t+1$ and the one at time $t$, which, to a certain extent, reflects the environmental fluctuations.*

**Remark 8.** *From Eq.(8), if we set $\eta_t = O\left(\sqrt{\frac{(1-\beta)C_T}{T}}\right)$ where $C_T := \sum_{t=1}^{T}|\mathcal{A}_{t+1}^*\Delta\mathcal{A}_t^*|$ is the deviation of maximizer sequence, we have that $\sum_{t=1}^{T}\mathbb{E}\Big(f_t(\mathcal{A}_t)\Big) \geq \left(\frac{1-e^{-c}}{c}\right)\sum_{t=1}^{T}f_t(\mathcal{A}_t^*) - O\left(\sqrt{\frac{C_T T}{1-\beta}}\right)$, which means that Algorithm 1 can attain a dynamic regret bound of $O(\sqrt{\frac{C_T T}{1-\beta}})$ against a $(\frac{1-e^{-c}}{c})$-approximation to the best comparator in hindsight.*

## 4.2 PROJECTION-FREE MULTI-AGENT ONLINE SURROGATE ENTROPIC ASCENT

The primary computational burden of Algorithm 1 lies in Line 13, where each agent is tasked with a single constrained mirror projection. Despite that this projection can be done very efficiently in linear time using standard methods described in (Pardalos & Kovoor, 1990; Brucker, 1984), the optimal solution to Eq.(7) admits an analytical expression when KL-divergence is selected as the metric. That is, we have the following theorem, whose proof is deferred to Appendix E.

**Theorem 4.** *Let $m$ be a positive integer and $g(x) = x\log(x)$. Then, the optimal solution $\mathbf{x}$ to the problem $\min_{\|\mathbf{b}\|_1\leq 1,\mathbf{b}\in[0,1]^m}\left(\langle\mathbf{z},\mathbf{b}\rangle + \mathcal{D}_{g,m}(\mathbf{b},\mathbf{y})\right)$ satisfies the following conditions: if $\sum_{i=1}^{m}y_i\exp(-z_i) \leq 1$, $x_i = y_i\exp(-z_i)$; otherwise, $x_i = \frac{y_i\exp(-z_i)}{\sum_{i=1}^{m}y_i\exp(-z_i)} \,\forall i \in [m]$.*

However, KL divergence does not meet with the Lipschitz condition in Assumption 4, as its gradient approaches infinity on the boundary. Fortunately, this drawback can be circumvented by mixing a uniform distribution. As a result, we get the *projection-free* Multi-Agent Online Surrogate Entropic Ascent (**MA-OSEA**) algorithm for the MA-OSM problem, as shown in Algorithm 2. Similarly, we also can verify the following regret bound for **MA-OSEA** algorithm.

**Theorem 5** (Proof deferred to Appendix F). *Consider our proposed Algorithm 2, if Assumption 1,2,3 and 5 hold, $\|\cdot\|$ is $l_1$ norm and each set function $f_t$ is monotone submodular with curvature $c$, then we can conclude that, when $\alpha = \frac{1-e^{-c}}{c}$,*

$$\mathbb{E}\Big(\mathbf{Reg}_\alpha^d(T)\Big) \leq C_1\Big(\sum_{t=1}^{T}\sum_{\tau=1}^{t}(\beta-\beta\gamma)^{t-\tau}\eta_\tau\Big) + \frac{NC_2}{\eta_{T+1}} + C_2\sum_{t=1}^{T}\frac{|\mathcal{A}_{t+1}^*\Delta\mathcal{A}_t^*|}{\eta_{t+1}} + \frac{NG}{2}\sum_{t=1}^{T}\eta_t + \sum_{t=1}^{T}\frac{C_3}{\eta_t} + GD\gamma,$$
$$(9)$$

*where $\mathcal{A}_t^*$ is any maximizer of Eq.(1), $C_1 = (4G^2 + LDG)N^{\frac{3}{2}}$, $C_2 = N\log(\frac{n}{\gamma})$, $C_3 = 2N^2\gamma$, $D = \sup_{\mathbf{x},\mathbf{y}\in\mathcal{C}} \|\mathbf{x} - \mathbf{y}\|_1$ and $\mathcal{C}$ is the constraint set of Eq.(3).*

**Remark 9.** *From Eq.(9), if we set $\eta_t = O\left(\sqrt{\frac{(1-\beta)C_T}{T}}\right)$ and $\gamma = O(T^{-2})$ where $C_T = \sum_{t=1}^{T}|\mathcal{A}_{t+1}^*\Delta\mathcal{A}_t^*|$, we have that $\sum_{t=1}^{T}\mathbb{E}\Big(f_t(\mathcal{A}_t)\Big) \geq \left(\frac{1-e^{-c}}{c}\right)\sum_{t=1}^{T}f_t(\mathcal{A}_t^*) - \widetilde{O}\left(\sqrt{\frac{C_T T}{1-\beta}}\right)$.*

---

**Algorithm 2** Multi-Agent Online Surrogate Entropic Ascent (**MA-OSEA**)

---

1: **Input:** Number of iterations $T$, the set of agents $\mathcal{N}$, communication graph $G(\mathcal{N}, \mathcal{E})$, weight matrix $\mathbf{W} = [w_{ij}] \in \mathbb{R}^{N \times N}$, 1-strongly decomposable convex function $\phi(\mathbf{x}) = \sum_{i=1}^{n} x_i \log(x_i)$, the curvature $c \in [0, 1]$, step size $\eta_t$ for $t \in [T]$, mixing parameter $\gamma$;

2: **Initialized:** for any agent $i \in \mathcal{N}$, let $[\mathbf{x}_{1,i}]_j = \frac{1}{|\mathcal{V}_i|}$, $\forall j \in \mathcal{V}_i$ and $[\mathbf{x}_{1,i}]_j = 0$, $\forall j \notin \mathcal{V}_i$

3: **for** $t \in [T]$ **do**

4:     **for** $i \in \mathcal{N}$ **do**

5:         Compute SUM $:= \sum_{a \in \mathcal{V}_i} [\mathbf{x}_{t,i}]_a$   ▷ Rounding (Lines 5-6)

6:         Select an action $a_{t,i}$ from the set $\mathcal{V}_i$ with probability $\frac{[\mathbf{x}_{t,i}]_a}{\text{SUM}}$

7:         Compute $\hat{\mathbf{x}}_{t,i} := (1 - \gamma)\mathbf{x}_{t,i} + \frac{\gamma}{n}\mathbf{1}_n$;   ▷ Mixing (Line 7)

8:         Exchange $\hat{\mathbf{x}}_{t,i}$ with each neighboring node $j \in \mathcal{N}_i$  ▷ Information aggregation (Lines 8-9)

9:         Aggregate the information by setting $\mathbf{y}_{t,i} = \sum_{j \in \mathcal{N}_i \cup \{i\}} w_{ij}\mathbf{x}_{t,j}$

10:       Sampling a random number $z_{t,i}$ from r.v. $\mathcal{Z}$  ▷ Surrogate gradient estimation (Lines 10-12)

11:       Sampling a random set $\mathcal{R}_{t,i} \sim z_{t,i} * \mathbf{x}_{t,i}$

12:       Compute $[\widetilde{\nabla} F_t^s(\mathbf{x}_{t,i})]_a := \frac{1-e^{-c}}{c}\big(f_t(\mathcal{R}_{t,i} \cup \{a\}) - f_t(\mathcal{R}_{t,i} \setminus \{a\})\big)$ for any $a \in \mathcal{V}_i$

13:       Update $[\mathbf{x}_{t+1,i}]_a = [\mathbf{y}_{t,i}]_a$, $\forall a \notin \mathcal{V}_i$   ▷ Update the probabilities of actions (Lines 13-18)

14:       Compute $\text{SUM}_1 := \sum_{a \in \mathcal{V}_i}\left([\mathbf{y}_{t,i}]_a \exp(\eta_t[\tilde{\nabla} F_t^s(\mathbf{x}_{t,i})]_a)\right)$

15:       **if** $\text{SUM}_1 \leq 1$ **then**

16:         $[\mathbf{x}_{t+1,i}]_a := [\mathbf{y}_{t,i}]_a \exp(\eta_t[\tilde{\nabla} F_t^s(\mathbf{x}_{t,i})]_a)$ for any $a \in \mathcal{V}_i$

17:       **else**

18:         $[\mathbf{x}_{t+1,i}]_a := [\mathbf{y}_{t,i}]_a \exp(\eta_t[\tilde{\nabla} F_t^s(\mathbf{x}_{t,i})]_a)/\text{SUM}_1$ for any $a \in \mathcal{V}_i$

---

## 5 NUMERICAL EXPERIMENTS

In this section, we evaluate our proposed Algorithm 1 and Algorithm 2 in simulated multi-target tracking tasks (Corah & Michael, 2021; Xu et al., 2023) with multiple agents.

**Experiment Setup.** We consider a 2D scenario where 20 agents are pursuing 30 moving targets with $T = 2500$ iterations over 50 seconds. At every iteration, each agent selects its direction of movement from "up", "down", "left", "right", or "diagonally". Concurrently, agents also adjust their speeds from a set of 5, 10, or 15 units/s. As for targets, we categorize them into three distinct types: the unpredictable 'Random', the structured 'Polyline', and the challenging 'Adversarial'. Specifically, at each iteration, a 'Random' target randomly changes its movement angle $\theta$ from $[0, 2\pi]$ and moves at a random speed between 5 and 10 units/s. A 'Polyline' target generally maintains its trajectory and only behaves like the 'Random' target at $\{0, \frac{T}{k}, \frac{2T}{k}, \dots, \frac{(k-1)T}{k}\}$-th iteration where $T$ is the predefined total iterations and $k$ is a random number from $\{1, 2, 4\}$. As for the 'Adversarial' target, it acts like a 'Random' target when all agents are beyond 20 units. However, if any agent is within a 20-unit range, the 'Adversarial' target escapes at a speed of 15 units/s for one second, pointing to the direction that maximizes the average distance from all agents. In addition, we initialize the starting positions of all agents and targets randomly within 20-unit radius circle centered at the origin.

**Objective Function.** We begin by defining the ground set $\mathcal{V} = \{(\theta, s, i) : s \in \{5, 10, 15\}\text{units/s}, i \in [20], \theta \in \{\frac{\pi}{4}, \frac{\pi}{2}, \frac{3\pi}{4}, \pi, \dots, 2\pi\}\}$ where $\theta, s, i$ represent the movement angle, speed and the identifier of agents, respectively. Moreover, the symbol $o_t(j)$ denotes the 2D location of target $j \in [30]$ at time $t \in [T]$ and $o_t^a(\theta, s, i)$ stands for the new position of agent $i$ after moving from its location at time $t-1$ in the direction of $\theta$ at a speed of $s$. In order to keep up with all targets, we naturally consider the following submodular objective function for each time $t$: $f_t(\mathcal{A}) = \sum_{j=1}^{30} \max_{(\theta,s,i) \in \mathcal{A}} \frac{1}{d(o_t^a(\theta,s,i), o_t(j))}$ where $d(\cdot, \cdot)$ is the distance between two locations and $\mathcal{A} \subseteq \mathcal{V}$.

**Analysis.** In Figure 3, we assess our proposed **MA-OSMA** and **MA-OSEA** against OSG (Xu et al., 2023) across scenarios with different proportions of 'Random', 'Polyline', and 'Adversarial' targets. The ratios are setting as 'R':'A':'P'=8:1:1 in Figure 3(a)-3(c), 6:3:1 in Figure 3(d)-3(f) and 4:5:1 in Figure 3(g)-3(i). The suffixes in **MA-OSMA** and **MA-OSEA** represent two different choices for communication graphs: 'c' for a complete graph and 'r' for an Erdos-Renyi random graph with average degree 4. From Figure 3(a),3(d) and 3(g), we can find that the running average utility

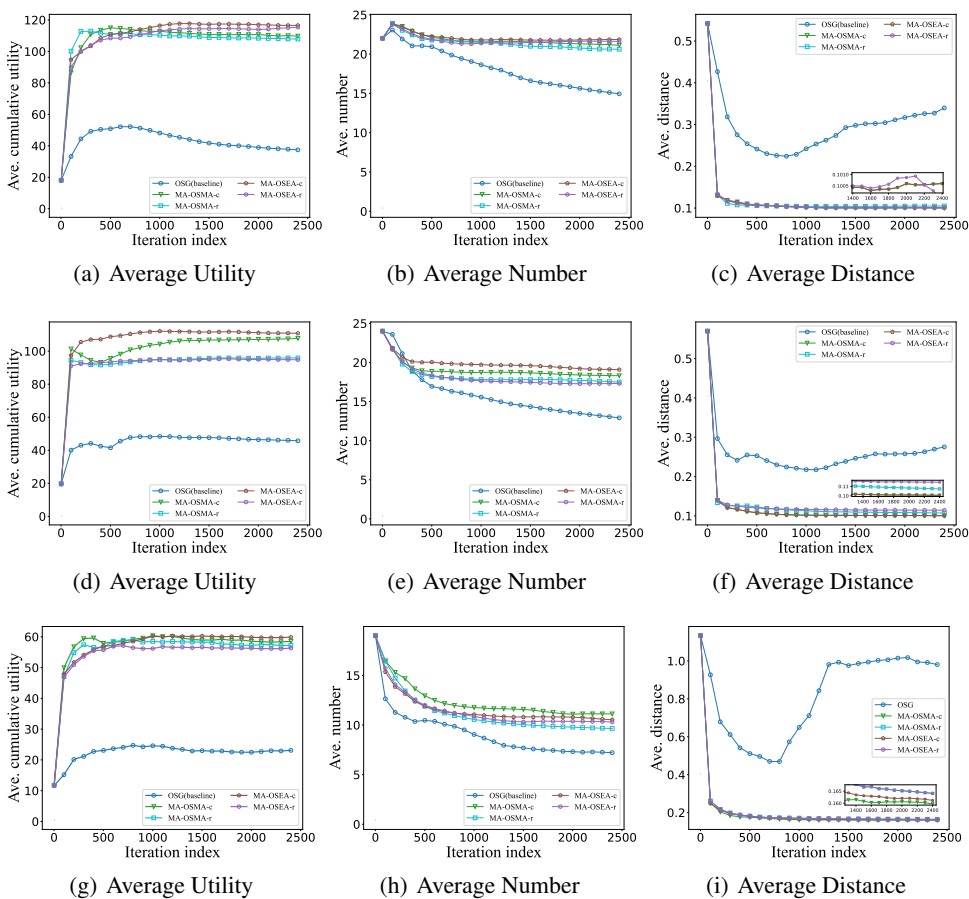

(a) Average Utility      (b) Average Number      (c) Average Distance

(d) Average Utility      (e) Average Number      (f) Average Distance

(g) Average Utility      (h) Average Number      (i) Average Distance

Figure 3: Comparison of average cumulative utility, average number of targets within 5 units, average distance of Top-5 nearest targets of **MA-OSMA-c**,**MA-OSMA-r**,**MA-OSEA-c** and **MA-OSEA-c** with OSG on different multi-target tracking scenarios(averaged over 5 runs).

$\frac{\sum_{\tau=1}^{t} f_\tau(\cup_{i \in \mathcal{N}}\{a_{\tau,i}\})}{t}$ of our proposed **MA-OSMA** and **MA-OSEA** significantly outperforms the OSG algorithm, which is in accord with our theoretical analysis. Similarly, the average number of targets within 5 units for **MA-OSMA** and **MA-OSEA** greatly exceeds that of the OSG, as depicted in Figure 3(b),3(e) and 3(h). Note that, due to 'Adversarial' targets, all curves for the average number exhibit a downward trend. Furthermore, our proposed **MA-OSMA** and **MA-OSEA** also can effectively reduce the average distance as shown in Figure 3(c), 3(f), and 3(i). Note that the algorithms over random graph perform comparably to those on complete graph in all figures, which, to some extent, reflects the communication efficiency of our proposed algorithms.

## 6 CONCLUSIONS

This paper presents two efficient algorithms for the multi-agent online submodular maximization problem. In sharp contrast with the previous OSG method, our proposed algorithms not only enjoy a tight $(\frac{1-e^{-c}}{c})$-approximation but also reduce the need for a complete communication graph. Finally, extensive empirical evaluations are performed to validate the effectiveness of our algorithms.

In many real-world scenarios, the local information gathered by one agent is often contaminated with noise, thereby leading to imperfect assessments of the marginal gains of its own actions. To tackle this challenge, a compelling strategy is to extend our regret analysis to accommodate the estimation errors inherent in marginal evaluations, as exemplified by the work of Corah & Michael (2021). Furthermore, another innovative direction is to generalize Algorithms 1 and 2 to adapt to time-varying and directed network topology (Nedić & Olshevsky, 2014; Nedić et al., 2017), as opposed to the static and undirected structure that is assumed. Lastly, the most promising direction is to design a parameter-free algorithm that eliminates the dependency on curvature of Algorithms 1 and 2.

ACKNOWLEDGMENTS

This research is supported by STI 2030—Major Projects (No. 2021ZD0201405), Shenzhen Basic Research Project (Natural Science Foundation) Basic Research Key Project (NO. JCYJ20241202124430041) and the RIE2025 Industry Alignment Fund – Industry Collaboration Projects (IAF-ICP) (Award I2301E0026), administered by A*STAR, as well as supported by Alibaba Group and NTU Singapore through Alibaba-NTU Global e-Sustainability CorpLab (ANGEL).

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

# A    ADDITIONAL DISCUSSIONS

## A.1    MORE DETAILS ON RELATED WORKS

**Single-Agent Submodular Maximization,** Maximization of submodular functions has recently found numerous applications in machine learning, operations research and economics, including data summarization (Lin & Bilmes, 2010; 2011; Wei et al., 2013; 2015), dictionary learning (Das & Kempe, 2018), product recommendation (Kempe et al., 2003; El-Arini et al., 2009; Mirzasoleiman et al., 2016a; Wang et al., 2022), federated learning (Balakrishnan et al., 2022; Rafiey, 2024; Zhao et al., 2024a;b) and in-context learning (Kumari et al., 2024; Fan et al., 2025; Jin et al., 2024a;b;c). When considering the simple cardinality constraint, the classical works(Fisher et al., 1978; Nemhauser et al., 1978) show that the greedy algorithm can achieve a tight $(1 - 1/e)$-approximation guarantee for monotone submodular maximization problem. As for the general matroid constraint, a continuous greedy algorithm with $(1 - 1/e)$-approximation guarantee is presented in (Calinescu et al., 2011; Chekuri et al., 2014). Especially when the submodular function has curvature $c \in [0, 1]$, Vondrák (2010) pointed out that the continuous greedy algorithm also can achieve an improved $(\frac{1-e^{-c}}{c})$-approximation guarantee.

**Multi-Agent Submodular Maximization.** Multi-agent submodular maximization(MA-SM) problem involves coordinating multiple agents to collaboratively maximize a submodular utility function. A commonly used solution for MA-SM problem heavily depends on the distributed implementation of the classic *sequential greedy* method (Fisher et al., 1978), which can ensure a $(\frac{1}{1+c})$-approximation (Conforti & Cornuéjols, 1984) when the submodular function has curvature $c \in [0, 1]$. However, this distributed algorithm requires each agent to have full access to the decisions of all previous agents, thereby forming a *complete* directed communication graph. Subsequently, several studies (Grimsman et al., 2018; Gharesifard & Smith, 2017; Marden, 2016) have investigated how the topology of the communication network affects the performance of the distributed greedy method. Particularly, Grimsman et al. (2018) pointed out that the worst-case performance of the distributed greedy algorithm will deteriorate in proportion to the size of the largest independent group of agents in the communication graph. In order to overcome these challenges, various studies (Rezazadeh & Kia, 2023; Robey et al., 2021; Du et al., 2022) utilized the multi-linear extension to design algorithms for solving MA-SM problem. Specifically, Du et al. (2022) devised a multi-agent variant of gradient ascent for MA-SM problem, which can attain $\frac{1}{2}OPT - \epsilon$ over connected communication graph at the cost of $O(\frac{n}{\epsilon^2})$ value queries to the submodular function where $OPT$ is the optimal value. After that, to improve the $\frac{1}{2}$-approximation, Robey et al. (2021) developed a multi-agent variant of continuous greedy method(Calinescu et al., 2011; Chekuri et al., 2014) with tight $(1 - 1/e)$-approximation. However, this multi-agent continuous greedy(Robey et al., 2021) requires the exact knowledge of the multi-linear extension function, which will lead to the exponential query complexity. To tackle this drawback, Rezazadeh & Kia (2023) also proposed a stochastic variant of continuous greedy method(Calinescu et al., 2011; Chekuri et al., 2014), which considers the curvature $c$ of submodular objectives and can enjoy $(\frac{1-e^{-c}}{c})OPT - \epsilon$ at the expense of $O(\frac{n \log(\frac{1}{\epsilon})}{\epsilon^3})$ value queries to the submodular function. In our Algorithm 1 and Algorithm 2, if any incoming objective function $f_t$ corresponds to some submodular function $f$ and we return the set $\cup_{i \in \mathcal{N}}\{a_{t,i}\}, \forall t \in [T]$ with probability $\frac{1}{T}$, we also can obtain two methods with the tight $(\frac{1-e^{-c}}{c})$-approximation guarantee for MA-SM problem. Notably, in sharp contrast with $O(\frac{n \log(\frac{1}{\epsilon})}{\epsilon^3})$ value queries in (Rezazadeh & Kia, 2023), the aforementioned variants of Algorithm 1 and Algorithm 2 only require inquiring the submodular objective $O(\frac{n}{\epsilon^2})$ and $O(\frac{n \log(\frac{1}{\epsilon})}{\epsilon^2})$ times to attain $(\frac{1-e^{-c}}{c})OPT - \epsilon$, respectively. We present a detailed comparison of our proposed **MA-OSMA** and **MA-OSEA** with previous studies for MA-SM problem in Table 2.

Because the multi-linear extension of a submodular set function belongs to the general continuous DR-submodular functions, we also review related works about decentralized continuous DR-submodular maximization. As for more detailed content on continuous DR-submodular maximization, please refer to Pedramfar et al. (2023; 2024).

**Decentralized Continuous DR-submodular Maximization.** A differentiable function $F : [0, 1]^n \to \mathbb{R}_+$ is *DR-submodular* if $\nabla F(\mathbf{x}) \le \nabla F(\mathbf{y})$ for any $\mathbf{x} \ge \mathbf{y}$. Mokhtari et al. (2018) is the first to study the decentralized continuous DR-submodular maximization problem, which showed an algorithm

titled DeCG achieving $(1-1/e)OPT-\epsilon$ for *monotone* and *deterministic* continuous DR-submodular maximization after $O(\frac{1}{\epsilon^2})$ iterations and $O(\frac{1}{\epsilon^2})$ communications. When only an unbiased estimate of gradient is available, Mokhtari et al. (2018) also presented a decentralized method named DeSCG for monotone cases, which attains $(1-1/e)OPT-\epsilon$ with $O(\frac{1}{\epsilon^3})$ communications after $O(\frac{1}{\epsilon^3})$ iterations. Next, Xie et al. (2019) presented a deterministic DeGTFW and a stochastic DeSGTFW by applying the gradient tracking techniques(Pu & Nedić, 2021) to DeCG and DeSCG respectively, both of which achieve the $(1-1/e)$-approximation with a faster $O(\frac{1}{\epsilon})$ convergence rate and a lower $O(\frac{1}{\epsilon})$ communications. After that, Gao et al. (2023) utilized the variance reduction technique(Fang et al., 2018) to propose two sample-efficient algorithms, namely, DeSVRFW-gp and DeSVRFW-gt, both of which can reduce the sample complexity of DeSGTFW from $O(\frac{1}{\epsilon^3})$ to $O(\frac{1}{\epsilon^2})$. Lastly, several studies(Zhu et al., 2021; Zhang et al., 2023; Liao et al., 2023) extended these aforementioned offline decentralized frameworks to time-varying DR-submodular objectives. It's important to note that the decentralized submodular maximization problem significantly differs from the multi-agent scenario emphasized in this paper. In the context of decentralized optimization, we typically assume that each local node maintains its own local utility function and the collective goal is to optimize the sum of these local functions. In contrast, within the scope of this paper, we assume that all agents share a common submodular function but each agent is restricted to accessing a unique set of actions.

| Method | Utility | Graph | Query Complexity | Reference |
|---|---|---|---|---|
| Greedy Method | $(\frac{1}{1+c})OPT$ | **complete** | $n$ | Conforti & Cornuéjols (1984) |
| Greedy Method | $(\frac{1}{1+\alpha(G)})OPT$ | connected | $n$ | Grimsman et al. (2018) |
| Projected Gradient Method | $\frac{1}{2}OPT-\epsilon$ | connected | $O(\frac{n}{\epsilon^2})$ | Du et al. (2022) |
| CDCG | $(1-\frac{1}{e})OPT-\epsilon$ | connected | $O(\frac{n2^n}{\epsilon})$ | Robey et al. (2021) |
| Distributed-CG | $(\frac{1-e^{-c}}{c})OPT-\epsilon$ | connected | $O\left(\frac{n\log(\frac{1}{\epsilon})}{\epsilon^3}\right)$ | Rezazadeh & Kia (2023) |
| Algorithm 1 | $(\frac{1-e^{-c}}{c})OPT-\epsilon$ | connected | $O(\frac{n}{\epsilon^2})$ | Theorem 3 & Remark 8 |
| Algorithm 2 | $(\frac{1-e^{-c}}{c})OPT-\epsilon$ | connected | $O\left(\frac{n\log(\frac{1}{\epsilon})}{\epsilon^2}\right)$ | Theorem 5 & Remark 9 |

Table 2: Comparison with prior works for Multi-Agent Submodular Maximization Problem. $OPT$ is the optimal value, $c$ is the curvature of submodular objective, $n := |\mathcal{V}|$ the total number of all available actions, $\alpha(G)$ is the number of nodes in the largest independent set in communication graph $G$ where $\alpha(G) \geq 1$. Note that the column of Query Complexity only considers the setting that each agent selects one action.

## A.2 MORE DISCUSSIONS ON ASSUMPTION 5

In this subsection, we show the Assumption 5 is well-established in both $l_1$ norm and $l_2$ norm.

At first, we review that, in Section 3.2, we define that:

$$\widetilde{\nabla}F^s(\mathbf{x}) = \left(\frac{1-e^{-c}}{c}\right)\Big(f(\mathcal{R}\cup\{1\})-f(\mathcal{R}\setminus\{1\}),\ldots,f(\mathcal{R}\cup\{n\})-f(\mathcal{R}\setminus\{n\})\Big), \quad (10)$$

where $z$ is sampled from the probability distribution of the random variable $\mathcal{Z}$ where $P(\mathcal{Z} \leq b) = (\frac{c}{1-e^{-c}})\int_0^b e^{c(z-1)}dz = \frac{e^{c(b-1)}-e^{-c}}{1-e^{-c}}$ for any $b \in [0,1]$ and $\mathcal{R} \sim z*\mathbf{x}$.

From Eq.(10) and the submodularity of set function $f$, we can know that $\mathbb{E}(\widetilde{\nabla}F^s(\mathbf{x})|\mathbf{x}) = \nabla F^s$ and $\mathbb{E}(\|\widetilde{\nabla}F^s(\mathbf{x})\|_\infty) \leq \frac{1-e^{-c}}{c}m_f$ where $m_f = \max_{a\in\mathcal{V}}f(\{a\})$ denotes maximum singleton value.

Moreover, Hassani et al. (2017) recently have shown that the multi-linear extension $F$ of a submodular function $f$ is $m_f$-smooth under $l_1$ norm, that is, $\|\nabla F(\mathbf{x}) - \nabla F(\mathbf{y})\|_\infty \leq m_f\|\mathbf{x}-\mathbf{y}\|_1$. From the definition of surrogate function, we also know that $\nabla F^s(\mathbf{x}) = \int_0^1 e^{c(z-1)}\nabla F(z*\boldsymbol{x})dz$ such that

$$\|\nabla F^s(\mathbf{x}) - \nabla F^s(\mathbf{y})\|_\infty \leq \int_0^1 e^{c(z-1)}\|\nabla F(z*\mathbf{x}) - \nabla F(z*\mathbf{y})\|_\infty dz$$

$$\leq \int_0^1 ze^{c(z-1)}m_f\|\mathbf{x}-\mathbf{y}\|_1 dz$$

$$= \frac{e^{-c}+c-1}{c^2}m_f\|\mathbf{x}-\mathbf{y}\|_1.$$

Therefore, we can conclude that the surrogate function $F^s$ of the multi-linear extension of a submodular function $f$ with curvature $c \in [0, 1]$ is $\left(\frac{e^{-c} + c - 1}{c^2} m_f\right)$-smooth under $l_1$ norm. Due to $\|\cdot\|_2 \leq \sqrt{n}\|\cdot\|_\infty$ and $\|\cdot\|_1 \leq \sqrt{n}\|\cdot\|_2$, we also can show that $\mathbb{E}(\|\widetilde{\nabla} F^s(\mathbf{x})\|_2) \leq \sqrt{n}\frac{1-e^{-c}}{c} m_f$ and $\|\nabla F^s(\mathbf{x}) - \nabla F^s(\mathbf{y})\|_2 \leq \left(n\frac{e^{-c} + c - 1}{c^2} m_f\right)\|\mathbf{x} - \mathbf{y}\|_2$.

With the previous results, in Assumption 5, we can set $G = \left(\frac{1-e^{-c}}{c}\right) \max_{a \in \mathcal{V}, t \in [T]} \left(f_t(\{a\})\right)$ and $L = \left(\frac{e^{-c} + c - 1}{c^2}\right) \max_{a \in \mathcal{V}, t \in [T]} \left(f_t(\{a\})\right)$ under $l_1$ norm. As for $l_2$ norm, we consider $G = \sqrt{n}\left(\frac{1-e^{-c}}{c}\right) \max_{a \in \mathcal{V}, t \in [T]} \left(f_t(\{a\})\right)$ and $L = n\left(\frac{e^{-c} + c - 1}{c^2}\right) \max_{a \in \mathcal{V}, t \in [T]} \left(f_t(\{a\})\right)$.

### A.3 MORE DISCUSSIONS ON EXPERIMENTS

In this subsection, we highlight some additional details about the experiments.

At first, all experiments are performed in Python 3.6.5 using CVX optimization tool (Grant & Boyd, 2014) on a MacBook Pro with Apple M1 Pro and 16GB RAM. Then, when considering the random graph, we set the weight matrix $\mathbf{W}$ as follow: if the edge $(i, j)$ is an edge of the graph, let $w_{ij} = 1/(1 + \max(d_i, d_j))$ where $d_i$ and $d_j$ are the degree of agent $i$ and $j$, respectively. If $(i, j)$ is not an edge of the graph and $i \neq j$, then $w_{ij} = 0$. Finally, we set $w_{ii} = 1 - \sum_{j \in \mathcal{N}_i} w_{ij}$.

As for the complete graph, we set $w_{ij} = \frac{1}{N}$ where $N$ is the number of agents. In **MA-OSMA** and **MA-OSEA** algorithms, we set $c = 1$ and $\eta_t = \frac{1}{\sqrt{T}}$.

## B PROOF OF THEOREM 1

We begin by reviewing some basic properties about the multi-linear extension of a submodular function.

**Lemma 1** (Calinescu et al. (2011); Bian et al. (2020)). *When $f : \mathcal{V} \to \mathbb{R}_+$ is a monotone submodular function, its multi-linear extension $F(\mathbf{x}) = \sum_{\mathcal{A} \subseteq \mathcal{V}} \left(f(\mathcal{A}) \prod_{a \in \mathcal{A}} x_a \prod_{a \notin \mathcal{A}} (1 - x_a)\right)$ has the following properties:*

1. *$F$ is monotone, that is, $\frac{\partial F(\mathbf{x})}{\partial x_i} \geq 0$ for any $i \in [n]$ and $\mathbf{x} \in [0, 1]^n$;*

2. *$F$ is concave along any non-negative direction $\boldsymbol{d} \in \mathbb{R}_+^n$;*

3. *$\frac{\partial F(\mathbf{x})}{\partial x_i} = \mathbb{E}_{\mathcal{R} \sim \mathbf{x}}\left(f(\mathcal{R} \cup \{i\}) - f(\mathcal{R} \setminus \{i\})\right)$,*

4. *$\nabla F(\mathbf{x}) \geq \nabla F(\mathbf{y})$ for any $\mathbf{x} \leq \mathbf{y}$ and $\mathbf{x}, \mathbf{y} \in [0, 1]^n$.*

With this lemma, we can show the following lemma.

**Lemma 2** (Calinescu et al. (2011); Bian et al. (2020)). *when $F$ is the multi-linear extension of a monotone submodular function $f$, we can conclude that*

$$\langle \mathbf{y} - \mathbf{x}, \nabla F(\mathbf{x})\rangle \geq F(\mathbf{x} \vee \mathbf{y}) + F(\mathbf{x} \wedge \mathbf{y}) - 2F(\mathbf{x}),$$

*where $\mathbf{x} \wedge \mathbf{y} := \min(\mathbf{x}, \mathbf{y})$ and $\mathbf{x} \vee \mathbf{y} := \max(\mathbf{x}, \mathbf{y})$ are component-wise minimum and component-wise maximum, respectively.*

*Proof.* From the second property about the concavity in Lemma 1 and $\mathbf{y} \vee \mathbf{x} \geq \mathbf{x}$ and $\mathbf{x} \wedge \mathbf{y} \leq \mathbf{x}$, we first have that

$$\begin{aligned} \langle \mathbf{y} \vee \mathbf{x} - \mathbf{x}, \nabla F(\mathbf{x})\rangle &\geq F(\mathbf{y} \vee \mathbf{x}) - F(\mathbf{x}), \\ \langle \mathbf{x} \wedge \mathbf{y} - \mathbf{x}, \nabla F(\mathbf{x})\rangle &\geq F(\mathbf{x} \wedge \mathbf{y}) - F(\mathbf{x}). \end{aligned} \qquad (11)$$

Due to $\mathbf{x} + \mathbf{y} = \mathbf{x} \vee \mathbf{y} + \mathbf{x} \wedge \mathbf{y}$, we therefore that that $\langle \mathbf{y} - \mathbf{x}, \nabla F(\mathbf{x})\rangle = \langle \mathbf{y} \vee \mathbf{x} - \mathbf{x}, \nabla F(\mathbf{x})\rangle + \langle \mathbf{x} \wedge \mathbf{y} - \mathbf{x}, \nabla F(\mathbf{x})\rangle \geq F(\mathbf{x} \vee \mathbf{y}) + F(\mathbf{x} \wedge \mathbf{y}) - 2F(\mathbf{x})$. $\square$

From the third property in Lemma 1 and the definition of curvature $c \in [0, 1]$, that is, $c := 1 - \min_{S \subseteq \mathcal{V}, e \notin S} \frac{f(S \cup \{e\}) - f(S)}{f(\{e\})}$, we can conclude that, when $f$ is monotone submodular, $f(S \cup \{e\}) - f(S) \geq (1 - c)f(\{e\})$ for any $S \subseteq \mathcal{V}, e \notin S$ such that, for any $\mathbf{x} \in [0, 1]^n$,

$$\frac{\partial F(\mathbf{x})}{\partial x_i} = \mathbb{E}_{\mathcal{R} \sim \mathbf{x}}\Big(f(\mathcal{R} \cup \{i\}) - f(\mathcal{R} \setminus \{i\})\Big) \geq (1 - c)f(\{i\}) = (1 - c)\frac{\partial F(\mathbf{0}_n)}{\partial x_i},$$

where $\mathbf{0}_n$ is $n$-dimensional zero vector and we general suppose $f(\emptyset) = 0$.

As a result, we can show that

**Lemma 3.** *when $F$ is the multi-linear extension of a monotone submodular function $f$ with curvature $c$, we have*

$$F(\mathbf{x} \vee \mathbf{y}) \geq (1 - c)\Big(F(\mathbf{x}) - F(\mathbf{x} \wedge \mathbf{y})\Big) + F(\mathbf{y}).$$

*Proof.*

$$
\begin{aligned}
F(\mathbf{x} \vee \mathbf{y}) - F(\mathbf{y}) &= \int_{z=0}^{1} \Big\langle \nabla F\Big(\mathbf{y} + z(\mathbf{x} \vee \mathbf{y} - \mathbf{y})\Big), \mathbf{x} \vee \mathbf{y} - \mathbf{y}\Big\rangle \mathrm{d}z \\
&\geq (1 - c)\langle \nabla F(\mathbf{0}_n), \mathbf{x} \vee \mathbf{y} - \mathbf{y}\rangle \\
&\geq (1 - c)\langle \nabla F(\mathbf{x} \wedge \mathbf{y}), \mathbf{x} \vee \mathbf{y} - \mathbf{y}\rangle \\
&= (1 - c)\langle \nabla F(\mathbf{x} \wedge \mathbf{y}), \mathbf{x} - \mathbf{x} \wedge \mathbf{y}\rangle \\
&\geq (1 - c)\Big(F(\mathbf{x}) - F(\mathbf{x} \wedge \mathbf{y})\Big),
\end{aligned}
$$

where the first inequality follows from $\nabla F(\mathbf{y} + z(\mathbf{x} \vee \mathbf{y} - \mathbf{y})) \geq (1 - c)\nabla F(\mathbf{0}_n)$; the second inequality comes from the fourth property in Lemma 1, i.e., $\nabla F(\mathbf{0}_n) \geq \nabla F(\mathbf{x} \wedge \mathbf{y})$. The second equality is due to $\mathbf{x} + \mathbf{y} = \mathbf{x} \vee \mathbf{y} + \mathbf{x} \wedge \mathbf{y}$ and the final inequality from the concavity along non-negative direction. $\square$

Merging Lemma 3 into Lemma 2, we finally have that $\langle \mathbf{y} - \mathbf{x}, \nabla F(\mathbf{x})\rangle \geq F(\mathbf{y}) - (1 + c)F(\mathbf{x})$ such that if $\mathbf{x}$ is a stationary point for $F$ over the domain $\mathcal{C} \subseteq [0, 1]^n$, we have that $0 \geq \langle \mathbf{y} - \mathbf{x}, \nabla F(\mathbf{x})\rangle \geq F(\mathbf{y}) - (1 + c)F(\mathbf{x})$ for any $\mathbf{y} \in \mathcal{C}$, so $F(\mathbf{x}) \geq \frac{1}{1+c} \max_{\mathbf{y} \in \mathcal{C}} F(\mathbf{y})$.

## C  PROOF OF THEOREM 2

In this section, we show the proof of Theorem 2. Before going into the details, we firstly prove the following lemma.

**Lemma 4.** *when $F$ is the multi-linear extension of a monotone submodular function $f$ with curvature $c$, we have, for any $\mathbf{y}, \mathbf{x} \in [0, 1]^n$,*

$$\langle \mathbf{y}, \nabla F(\mathbf{x})\rangle \geq F(\mathbf{y}) - cF(\mathbf{x}).$$

*Proof.* Due to $\mathbf{x} + \mathbf{y} = \mathbf{x} \vee \mathbf{y} + \mathbf{x} \wedge \mathbf{y}$, we have that

$$
\begin{aligned}
\langle \mathbf{y}, \nabla F(\mathbf{x})\rangle &= \langle \mathbf{x} \vee \mathbf{y} - \mathbf{x}, \nabla F(\mathbf{x})\rangle + \langle \mathbf{x} \wedge \mathbf{y}, \nabla F(\mathbf{x})\rangle \\
&\geq F(\mathbf{x} \vee \mathbf{y}) - F(\mathbf{x}) + (1 - c)\langle \mathbf{x} \wedge \mathbf{y}, \nabla F(\mathbf{0}_n)\rangle \\
&\geq F(\mathbf{x} \vee \mathbf{y}) - F(\mathbf{x}) + (1 - c)F(\mathbf{x} \wedge \mathbf{y}) \\
&\geq F(\mathbf{y}) - cF(\mathbf{x}),
\end{aligned}
$$

where the first inequality follows from $\nabla F(\mathbf{x}) \geq (1 - c)\nabla F(\mathbf{0}_n)$ and the concavity along non-negative direction, i.e., $\langle \mathbf{x} \vee \mathbf{y} - \mathbf{x}, \nabla F(\mathbf{x})\rangle \geq F(\mathbf{x} \vee \mathbf{y}) - F(\mathbf{x})$; the final inequality comes from Lemma 3. $\square$

Next, we verify Theorem 2.

*Proof.* Firstly, we prove that

$$
\left\langle \mathbf{x}, \int_0^1 e^{c(z-1)} \nabla F(z * \mathbf{x}) \mathrm{d}z \right\rangle
$$

$$
= \int_0^1 e^{c(z-1)} \langle \mathbf{x}, \nabla F(z * \mathbf{x}) \rangle \mathrm{d}z
$$

$$
= \int_0^1 e^{c(z-1)} \mathrm{d}\Big( F(z * \mathbf{x}) \Big) \tag{12}
$$

$$
= e^{c(z-1)} * F(z * \mathbf{x})|_{z=0}^{z=1} - \int_0^1 F(z * \mathbf{x}) \mathrm{d}\Big( e^{c(z-1)} \Big)
$$

$$
= F(\mathbf{x}) - c \int_0^1 e^{c(z-1)} F(z * \mathbf{x}) \mathrm{d}z.
$$

Then, we show an upper bound for $\langle \mathbf{y}, \int_0^1 e^{c(z-1)} \nabla F(z * \mathbf{x}) \mathrm{d}z \rangle$,

$$
\left\langle \mathbf{y}, \int_0^1 e^{c(z-1)} \nabla F(z * \mathbf{x}) \mathrm{d}z \right\rangle = \int_0^1 e^{c(z-1)} \langle \mathbf{y}, \nabla F(z*\mathbf{x}) \rangle \mathrm{d}z \geq \int_0^1 e^{c(z-1)} (F(\mathbf{y}) - cF(z*\mathbf{x})) \mathrm{d}z,
$$
$$
\tag{13}
$$

where the inequality follows from Lemma 4. Combining Eq.(12) with Eq.(13), we have

$$
\left\langle \mathbf{y} - \mathbf{x}, \int_0^1 e^{c(z-1)} \nabla F(z * \mathbf{x}) \mathrm{d}z \right\rangle \geq \left( \int_0^1 e^{c(z-1)} \mathrm{d}z \right) F(\mathbf{y}) - F(\mathbf{x}) = \left( \int_0^1 e^{c(z-1)} \mathrm{d}z \right) F(\mathbf{y}) - F(\mathbf{x}).
$$

Note the $\int_0^1 e^{c(z-1)} \mathrm{d}z = \frac{1-e^{-c}}{c}$. $\qquad\qquad\qquad\square$

## D    PROOF OF THEOREM 3

In this section, we show the proof of Theorem 3. Before going into the details, we firstly review a standard lemma for the mirror projection.

**Lemma 5** (Chen & Teboulle (1993); Jadbabaie et al. (2015))**.** *Let $\phi : [0,1]^n \to \mathbb{R}$ be a 1-strongly convex function with respect to the norm $\| \cdot \|$ and $D_\phi(\mathbf{x}, \mathbf{y})$ represent the Bregman divergence with respect to $\phi$, respectively. Then, any update of the form*

$$
\mathbf{x}^+ = \min_{\mathbf{y} \in \mathcal{C}} \langle \boldsymbol{b}, \mathbf{y} \rangle + \mathcal{D}_\phi(\mathbf{y}, \mathbf{x}),
$$

*satisfies the following inequality*

$$
\langle \mathbf{x}^+ - \mathbf{z}, \boldsymbol{b} \rangle \leq \mathcal{D}_\phi(\mathbf{z}, \mathbf{x}) - \mathcal{D}_\phi(\mathbf{z}, \mathbf{x}^+) - \mathcal{D}_\phi(\mathbf{x}^+, \mathbf{x}),
$$

*for any $\mathbf{z} \in \mathcal{C}$ where $\mathcal{C}$ is a convex domain in $[0,1]^n$.*

Next, we verify that each local variable $\mathbf{x}_{t,i}$ of Algorithm 1 is included in the constraint of continuous problem Eq.(3) for any $t \in [T]$ and $i \in \mathcal{N}$.

**Lemma 6.** *In Algorithm 1, if we set the constraint $\mathcal{C} = \{\mathbf{x} \in [0,1]^n : \sum_{a \in \mathcal{V}_i} x_a \leq 1, \forall i \in \mathcal{N}\}$ and Assumption 2 holds, we have that, for any $t \in [T]$ and $i \in \mathcal{N}$, we have that, for any $t \in [T]$ and $i \in \mathcal{N}$, $\mathbf{x}_{t,i} \in \mathcal{C}$ and $\mathbf{y}_{t,i} \in \mathcal{C}$.*

*Proof.* We prove this lemma by induction. At first, from the Line 2 in Algorithm 1, we know that $\mathbf{x}_{1,i} \in \mathcal{C}$ for any $i \in \mathcal{N}$. Moreover, due to Assumption 2 and Line 8, we also can infer $\mathbf{y}_{1,i} = \sum_{j \in \mathcal{N}_i \cup \{i\}} w_{ij} \mathbf{x}_{1,j} \in \mathcal{C}$ for any $i \in \mathcal{N}$. For any $1 \leq t < T$, if we assume $\mathbf{x}_{t,i} \in \mathcal{C}$ and $\mathbf{y}_{t,i} \in \mathcal{C}$ for any $i \in \mathcal{N}$. Then from the Line 12 in Algorithm 1, we know $\sum_{a \in \mathcal{V}_j} [\mathbf{x}_{t+1,i}]_a = \sum_{a \in \mathcal{V}_j} [\mathbf{y}_{t,i}]_a \leq 1$ for any $j \neq i$ and $i \in \mathcal{N}_i$. Furthermore, Line 13 implies that $[\mathbf{x}_{t+1,i}]_{\mathcal{V}_i}$ is the projection over the constraint $\sum_{k=1}^{n_i} b_k \leq 1$, so $\sum_{a \in \mathcal{V}_i} [\mathbf{x}_{t+1,i}]_a \leq 1$. we can conclude that $\mathbf{x}_{t+1,i} \in \mathcal{C}$. Due to Assumption 2, we can further show $\mathbf{y}_{t+1,i} \in \mathcal{C}$. $\qquad\square$

With Lemma 6, we integrate the probability update procedures for actions $i \in \mathcal{V}_i$ and those not in $\mathcal{V}_i$ within Algorithm 1, i,e, Lines 12-13.

**Lemma 7.** *In Algorithm 1, if we set the constraint $\mathcal{C} = \{\mathbf{x} \in [0,1]^n : \sum_{a \in \mathcal{V}_i} x_a \leq 1, \forall i \in \mathcal{N}\}$ and Assumption 1 and 2 hold, we have that, for any $t \in [T]$ and $i \in \mathcal{N}$,*

$$\mathbf{x}_{t+1,i} = \arg\min_{\mathbf{x} \in \mathcal{C}} \left( - \langle \widetilde{\nabla} F_t^s(\mathbf{x}_{t,i}) \odot \mathbf{1}_{\mathcal{V}_i}, \mathbf{x} \rangle + \frac{1}{\eta_t} \mathcal{D}_\phi(\mathbf{x}, \mathbf{y}_{t,i}) \right),$$

*where $\odot$ denotes the coordinate-wise multiplication, i.e., the $i$-th element of vector $\mathbf{x} \odot \mathbf{y}$ is $x_i y_i$, and $\mathbf{1}_{\mathcal{V}_i}$ denotes a $n$-dimensional vector where the entries at $\mathcal{V}_i$ is equal to 1 and all others are 0.*

*Proof.* Firstly, we define the solution of the problem $\min_{\mathbf{x} \in \mathcal{C}} \left( - \langle \widetilde{\nabla} F_t^s(\mathbf{x}_{t,i}) \odot \mathbf{1}_{\mathcal{V}_i}, \mathbf{x} \rangle + \frac{1}{\eta_t} \mathcal{D}_\phi(\mathbf{x}, \mathbf{y}_{t,i}) \right)$ as $\mathbf{o}^* \in [0,1]^n$. Due to Assumption 1,

$$
\begin{aligned}
\mathbf{o}^* &= \arg\min_{\mathbf{x} \in \mathcal{C}} \left( - \langle \widetilde{\nabla} F_t^s(\mathbf{x}_{t,i}) \odot \mathbf{1}_{\mathcal{V}_i}, \mathbf{x} \rangle + \frac{1}{\eta_t} \mathcal{D}_\phi(\mathbf{x}, \mathbf{y}_{t,i}) \right) \\
&= \arg\min_{\mathbf{x} \in \mathcal{C}} \left( - \langle [\widetilde{\nabla} F_t^s(\mathbf{x}_{t,i})]_{\mathcal{V}_i}, [\mathbf{x}]_{\mathcal{V}_i} \rangle + \frac{1}{\eta_t} \mathcal{D}_{g,n_i}([\mathbf{x}]_{\mathcal{V}_i}, [\mathbf{y}_{t,i}]_{\mathcal{V}_i}) + \frac{1}{\eta_t} \mathcal{D}_{g,n-n_i}([\mathbf{x}]_{\mathcal{V} \backslash \mathcal{V}_i}, [\mathbf{y}_{t,i}]_{\mathcal{V} \backslash \mathcal{V}_i}) \right).
\end{aligned}
$$

As a result, we can conclude that

$$[\mathbf{o}^*]_{\mathcal{V}_i} = \arg\min_{\sum_{a \in \mathcal{V}_i} b_a \leq 1} \left( - \langle [\widetilde{\nabla} F_t^s(\mathbf{x}_{t,i})]_{\mathcal{V}_i}, \mathbf{b} \rangle + \frac{1}{\eta_t} \mathcal{D}_{g,n_i}(\mathbf{b}, [\mathbf{y}_{t,i}]_{\mathcal{V}_i}) \right),$$

$$[\mathbf{o}^*]_{\mathcal{V} \backslash \mathcal{V}_i} = \arg\min_{\sum_{a \in \mathcal{V}_j} z_a \leq 1, \forall j \in \mathcal{N} \backslash \{i\}} \left( \mathcal{D}_{g,n_i}(\mathbf{z}, [\mathbf{y}_{t,i}]_{\mathcal{V} \backslash \mathcal{V}_i}) \right) = [\mathbf{y}_{t,i}]_{\mathcal{V} \backslash \mathcal{V}_i}, \quad \text{(Note that } \mathbf{y}_{t,i} \in \mathcal{C} \text{ (See lemma 6))}$$

$$\tag{14}$$

where $\mathbf{b} \in [0,1]^{n_i}$ and $\mathbf{z} \in [0,1]^{n-n_i}$. From the Eq.(14) and Lines 12-13 in Algorithm 1, we get the $\mathbf{o}^* = \mathbf{x}_{t+1,i}$. $\qquad\square$

In the following part, we define some commonly used symbols for the proof of Theorem 3:

$$\bar{\mathbf{x}}_t := \frac{\sum_{i=1}^N \mathbf{x}_{t,i}}{N}, \quad \mathbf{x}_t^{cate} := [\mathbf{x}_{t,1}; \mathbf{x}_{t,2}; \dots; \mathbf{x}_{t,N}] \in \mathbb{R}^{n*N};$$

$$\bar{\mathbf{y}}_t := \frac{\sum_{i=1}^N \mathbf{y}_{t,i}}{N}, \quad \mathbf{y}_t^{cate} := [\mathbf{y}_{t,1}; \mathbf{y}_{t,2}; \dots; \mathbf{y}_{t,N}] \in \mathbb{R}^{n*N};$$

$$\mathbf{r}_{t,i} := \mathbf{x}_{t+1,i} - \mathbf{y}_{t,i}, \quad \mathbf{r}_t^{cate} := [\mathbf{r}_{t,1}; \mathbf{r}_{t,2}; \dots; \mathbf{r}_{t,N}] \in \mathbb{R}^{n*N};$$

With these symbols, we can verify that

**Lemma 8.** *If we set the constraint $\mathcal{C} = \{\mathbf{x} \in [0,1]^n : \sum_{a \in \mathcal{V}_i} x_a \leq 1, \forall i \in \mathcal{N}\}$ and Assumption 1 and 5 hold, we have that*
$$\mathbb{E}(\|\mathbf{r}_{t,i}\|) = \mathbb{E}(\|\mathbf{x}_{t+1,i} - \mathbf{y}_{t,i}\|) \leq G\eta_t.$$

*Proof.* According to Lemma 7, we have that

$$\mathbf{x}_{t+1,i} = \arg\min_{\mathbf{x} \in \mathcal{C}} \left( - \langle \widetilde{\nabla} F_t^s(\mathbf{x}_{t,i}) \odot \mathbf{1}_{\mathcal{V}_i}, \mathbf{x} \rangle + \frac{1}{\eta_t} \mathcal{D}_\phi(\mathbf{x}, \mathbf{y}_{t,i}) \right),$$

where $\mathcal{C} = \{\mathbf{x} \in [0,1]^n : \sum_{a \in \mathcal{V}_i} x_a \leq 1, \forall i \in \mathcal{N}\}$.

From the Lemma 5, we have

$$\eta_t \langle \mathbf{x}_{t+1,i} - \mathbf{x}, -\tilde{\nabla} F_t^s(\mathbf{x}_{t,i}) \odot \mathbf{1}_{\mathcal{V}_i} \rangle \leq \mathcal{D}_\phi(\mathbf{x}, \mathbf{y}_{t,i}) - \mathcal{D}_\phi(\mathbf{x}, \mathbf{x}_{t+1,i}) - \mathcal{D}_\phi(\mathbf{x}_{t+1,i}, \mathbf{y}_{t,i}), \tag{15}$$

for any $\mathbf{x} \in \mathcal{C}$. If we set $\mathbf{x} = \mathbf{y}_{t,i}{}^1$ in Eq.(15), we have that

$$\eta_t \langle \mathbf{x}_{t+1,i} - \mathbf{y}_{t,i}, \tilde{\nabla} F_t^s(\mathbf{x}_{t,i}) \odot \mathbf{1}_{\mathcal{V}_i} \rangle \geq \mathcal{D}_\phi(\mathbf{y}_{t,i}, \mathbf{x}_{t+1,i}) + \mathcal{D}_\phi(\mathbf{x}_{t+1,i}, \mathbf{y}_{t,i}) \geq \|\mathbf{x}_{t+1,i} - \mathbf{y}_{t,i}\|,$$

where the final inequality follows from the 1-strongly convex of $\phi$.

From the Young inequality, we have that $\eta_t \langle \mathbf{x}_{t+1,i} - \mathbf{y}_{t,i}, \tilde{\nabla} F_t^s(\mathbf{x}_{t,i}) \odot \mathbf{1}_{\mathcal{V}_i} \rangle \leq \frac{\|\mathbf{x}_{t+1,i} - \mathbf{y}_{t,i}\|}{2} + \frac{\|\eta_t \tilde{\nabla} F_t^A(\mathbf{x}_{t,i}) \odot \mathbf{1}_{\mathcal{V}_i}\|_*}{2}$ such that $\mathbb{E}(\|\mathbf{x}_{t+1,i} - \mathbf{y}_{t,i}\|) \leq \mathbb{E}(\|\eta_t \tilde{\nabla} F_t^s(\mathbf{x}_{t,i}) \odot \mathbf{1}_{\mathcal{V}_i}\|_*) \leq \eta_t G$ from the Assumption 5. $\qquad\square$

---

[1]Note that we prove $\mathbf{y}_{t,i} \in \mathcal{C}$ in Lemma 6.

With this Lemma 8, we next derive an upper bound about the deviation between $\mathbf{x}_{t+1,i}, \mathbf{y}_{t+1,i}$ and the average $\bar{\mathbf{x}}_{t+1}$.

**Lemma 9.** *Under the Assumption 1, 2 and 5, we have that, for any $t \in [T]$ and $i \in \mathcal{N}$,*

$$\mathbb{E}(\|\mathbf{x}_{t+1,i} - \bar{\mathbf{x}}_{t+1}\|) \leq \sum_{\tau=1}^{t} \sqrt{N}\beta^{t-\tau}\eta_\tau G,$$

$$\mathbb{E}(\|\mathbf{y}_{t+1,i} - \bar{\mathbf{x}}_{t+1}\|) \leq \sum_{\tau=1}^{t} \sqrt{N}\beta^{t-\tau}\eta_\tau G,$$

*where $\beta = \max(|\lambda_2(\mathbf{W})|, |\lambda_N(\mathbf{W})|)$ is the second largest magnitude of the eigenvalues of the weight matrix $\mathbf{W}$.*

*Proof.* From the definition of $\mathbf{r}_{t,i}$, we can conclude that

$$\mathbf{x}_{t+1,i} = \mathbf{r}_{t,i} + \mathbf{y}_{t,i} = \mathbf{r}_{t,i} + \sum_{j \in \mathcal{N}_i \cup \{i\}} w_{ij}\mathbf{x}_{t,j}, \tag{16}$$

where the final equality follows from Line 8 in Algorithm 1.

As a result, from the Eq.(16), we can show that

$$\begin{aligned}
\mathbf{x}_{t+1}^{cate} &= \mathbf{r}_t^{cate} + (\mathbf{W} \otimes \mathbf{I}_n)\mathbf{x}_t^{cate} \\
&= \sum_{\tau=1}^{t} (\mathbf{W} \otimes \mathbf{I}_n)^{t-\tau}\mathbf{r}_\tau^{cate} \\
&= \sum_{\tau=1}^{t} (\mathbf{W}^{t-\tau} \otimes \mathbf{I}_n)\mathbf{r}_\tau^{cate},
\end{aligned} \tag{17}$$

where the symbol $\otimes$ denotes the Kronecker product.

If we also define $\bar{\mathbf{x}}_t^{cate} = [\bar{\mathbf{x}}_t; \bar{\mathbf{x}}_t; \ldots; \bar{\mathbf{x}}_t] \in \mathbb{R}^{n*N}$ and from the Eq.(17), we also have that

$$\begin{aligned}
\bar{\mathbf{x}}_{t+1}^{cate} &= \left(\frac{\mathbf{1}_N\mathbf{1}_N^T}{N} \otimes \mathbf{I}_N\right)\mathbf{x}_{t+1}^{cate} \\
&= \sum_{\tau=1}^{t} \left(\frac{\mathbf{1}_N\mathbf{1}_N^T}{N} \otimes \mathbf{I}_n\right)\mathbf{r}_\tau^{cate}.
\end{aligned} \tag{18}$$

Then, from the Eq.(17) and Eq.(18), we have that , for any $i \in \mathcal{N}$,

$$\mathbf{x}_{t+1,i} - \bar{\mathbf{x}}_{t+1} = \sum_{\tau=1}^{t}\sum_{j \in \mathcal{N}_i \cup \{i\}} \left([\mathbf{W}^{t-\tau}]_{ij} - \frac{1}{N}\right)\mathbf{r}_{\tau,j}. \tag{19}$$

Eq.(19) indicates that

$$\begin{aligned}
\mathbb{E}\left(\|\mathbf{x}_{t+1,i} - \bar{\mathbf{x}}_{t+1}\|\right) &= \mathbb{E}\left(\left\|\sum_{\tau=1}^{t}\sum_{j \in \mathcal{N}_i \cup \{i\}} ([\mathbf{W}^{t-\tau}]_{ij} - \frac{1}{N})\mathbf{r}_{\tau,j}\right\|\right) \\
&\leq \mathbb{E}\left(\sum_{\tau=1}^{t}\sum_{j \in \mathcal{N}_i \cup \{i\}} \left|[\mathbf{W}^{t-\tau}]_{ij} - \frac{1}{N}\right| \|\mathbf{r}_{\tau,j}\|\right) \\
&\leq \sum_{\tau=1}^{t}\sum_{j \in \mathcal{N}_i \cup \{i\}} \left|[\mathbf{W}^{t-\tau}]_{ij} - \frac{1}{N}\right| \eta_\tau G \leq \sum_{\tau=1}^{t} \sqrt{N}\beta^{t-\tau}\eta_\tau G,
\end{aligned}$$

where the second inequality comes from Lemma 8 and the final inequality follows from $\sum_{j \in \mathcal{N}_i \cup \{i\}} |[\mathbf{W}^{t-\tau}]_{ij} - \frac{1}{N}| \leq \sqrt{N}\beta^{t-\tau}$(See Proposition 1 in (Nedic & Ozdaglar, 2009)). Due to $\mathbf{y}_{t+1,i} = \sum_{j \in \mathcal{N}_i \cup \{i\}} w_{ij}\mathbf{x}_{t+1,j}$ we also can have $\mathbb{E}(\|\mathbf{y}_{t+1,i} - \bar{\mathbf{x}}_{t+1}\|) \leq \sum_{j \in \mathcal{N}_i \cup \{i\}} w_{ij}\mathbb{E}(\|\mathbf{x}_{t+1,j} - \bar{\mathbf{x}}_{t+1}\|) \leq \sum_{\tau=1}^{t} \sqrt{N}\beta^{t-\tau}\eta_\tau G$.

$\square$

**Lemma 10.** *Consider our proposed Algorithm 1, if Assumption 1,2,3,5 hold and each set function $f_t$ is monotone submodular with curvature $c$ for any $t \in [T]$, then we can conclude that,*

$$
\left(\frac{1-e^{-c}}{c}\right) \sum_{t=1}^{T} F_t(\mathbf{x}_t^*) - \sum_{t=1}^{T} \mathbb{E}\Big(F_t(\bar{\mathbf{x}}_t)\Big)
$$
$$
\leq (3G + LDG)\Big(\sum_{t=1}^{T}\sum_{\tau=1}^{t} N^{\frac{3}{2}}\beta^{t-\tau}\eta_\tau\Big) + \sum_{t=1}^{T}\sum_{i\in\mathcal{N}}\frac{1}{\eta_t}\mathbb{E}\Big(\mathcal{D}_\phi(\mathbf{x}_t^*, \mathbf{y}_{t,i}) - \mathcal{D}_\phi(\mathbf{x}_t^*, \mathbf{x}_{t+1,i})\Big) + \frac{NG}{2}\sum_{t=1}^{T}\eta_t,
$$

*where $\mathbf{x}_t^*$ is the optimal solution of Eq.(3) and $D = \sup_{\mathbf{x},\mathbf{y}\in\mathcal{C}}\|\mathbf{x} - \mathbf{y}\|$ where $\mathcal{C} = \{\mathbf{x} \in [0,1]^n : \sum_{a\in\mathcal{V}_i} x_a \leq 1, \forall i \in \mathcal{N}\}$.*

*Proof.* From the Eq.(4) in Theorem 2, we have that

$$
\left(\frac{1-e^{-c}}{c}\right)F_t(\mathbf{x}_t^*) - F_t(\bar{\mathbf{x}}_t) \leq \langle \nabla F_t^s(\bar{\mathbf{x}}_t), \mathbf{x}_t^* - \bar{\mathbf{x}}_t\rangle
$$
$$
= \underbrace{\langle \nabla F_t^s(\bar{\mathbf{x}}_t) - \sum_{i\in\mathcal{N}}\Big(\nabla F_t^s(\mathbf{x}_{t,i}) \odot \mathbf{1}_{\mathcal{V}_i}\Big), \mathbf{x}_t^* - \bar{\mathbf{x}}_t\rangle}_{①}
$$
$$
+ \underbrace{\sum_{i\in\mathcal{N}}\langle \nabla F_t^s(\mathbf{x}_{t,i}) \odot \mathbf{1}_{\mathcal{V}_i}, \mathbf{x}_t^* - \mathbf{x}_{t,i}\rangle}_{②} \qquad (20)
$$
$$
+ \underbrace{\sum_{i\in\mathcal{N}}\langle \nabla F_t^s(\mathbf{x}_{t,i}) \odot \mathbf{1}_{\mathcal{V}_i}, \mathbf{x}_{t,i} - \bar{\mathbf{x}}_t\rangle}_{③},
$$

where $\odot$ denotes the coordinate-wise multiplication, i.e.,the $i$-th element of vector $\mathbf{x} \odot \mathbf{y}$ is $x_i y_i$, and $\mathbf{1}_{\mathcal{V}_i}$ denotes a $n$-dimensional vector where the entries at $\mathcal{V}_i$ is equal to 1 and all others are 0.

For ①, we have

$$
\langle \nabla F_t^s(\bar{\mathbf{x}}_t) - \sum_{i\in\mathcal{N}}\Big(\nabla F_t^s(\mathbf{x}_{t,i}) \odot \mathbf{1}_{\mathcal{V}_i}\Big), \mathbf{x}_t^* - \bar{\mathbf{x}}_t\rangle
$$
$$
\leq \left\|\nabla F_t^s(\bar{\mathbf{x}}_t) - \sum_{i\in\mathcal{N}}\Big(\nabla F_t^s(\mathbf{x}_{t,i}) \odot \mathbf{1}_{\mathcal{V}_i}\Big)\right\|_* \|\mathbf{x}_t^* - \bar{\mathbf{x}}_t\|
$$
$$
\leq \|\mathbf{x}_t^* - \bar{\mathbf{x}}_t\| \sum_{i\in\mathcal{N}}\Big(\|[\nabla F_t^s(\bar{\mathbf{x}}_t)]_{\mathcal{V}_i} - [\nabla F_t^s(\mathbf{x}_{t,i})]_{\mathcal{V}_i}\|_*\Big)
$$
$$
\leq \|\mathbf{x}_t^* - \bar{\mathbf{x}}_t\| \sum_{i\in\mathcal{N}}\Big(\|\nabla F_t^s(\bar{\mathbf{x}}_t) - \nabla F_t^s(\mathbf{x}_{t,i})\|_*\Big) \qquad (21)
$$
$$
\leq \|\mathbf{x}_t^* - \bar{\mathbf{x}}_t\| \sum_{i\in\mathcal{N}}\Big(L\|\bar{\mathbf{x}}_t - \mathbf{x}_{t,i}\|\Big)
$$
$$
\leq LD \sum_{i\in\mathcal{N}}\Big(\|\bar{\mathbf{x}}_t - \mathbf{x}_{t,i}\|\Big)
$$
$$
\leq LDG \sum_{\tau=1}^{t} N^{\frac{3}{2}}\beta^{t-\tau}\eta_\tau,
$$

where the fourth inequality follows from Assumption 5; the fifth comes from $D = \sup_{\mathbf{x},\mathbf{y}\in\mathcal{C}}\|\mathbf{x} - \mathbf{y}\|$ and the final inequality from Lemma 9.

For ③, from Assumption 5 and Lemma 9, we have,

$$
\mathbb{E}\Big(\sum_{i\in\mathcal{N}}\langle \nabla F_t^s(\mathbf{x}_{t,i}) \odot \mathbf{1}_{\mathcal{V}_i}, \mathbf{x}_{t,i} - \bar{\mathbf{x}}_t\rangle\Big) \leq G\sum_{i\in\mathcal{N}}\mathbb{E}\Big(\|\mathbf{x}_{t,i} - \bar{\mathbf{x}}_t\|\Big) \leq \sum_{\tau=1}^{t} GN^{\frac{3}{2}}\beta^{t-\tau}\eta_\tau. \qquad (22)
$$

As for ②, we have,

$$\mathbb{E}\Big( \sum_{i \in \mathcal{N}} \langle \nabla F_t^s(\mathbf{x}_{t,i}) \odot \mathbf{1}_{\mathcal{V}_i}, \mathbf{x}_t^* - \mathbf{x}_{t,i} \rangle \Big)$$

$$= \mathbb{E}\Big( \mathbb{E}\Big( \sum_{i \in \mathcal{N}} \langle \nabla F_t^s(\mathbf{x}_{t,i}) \odot \mathbf{1}_{\mathcal{V}_i}, \mathbf{x}_t^* - \mathbf{x}_{t,i} \rangle | \mathbf{x}_{t,i} \Big) \Big)$$

$$= \mathbb{E}\Big( \mathbb{E}\Big( \sum_{i \in \mathcal{N}} \langle \tilde{\nabla} F_t^s(\mathbf{x}_{t,i}) \odot \mathbf{1}_{\mathcal{V}_i}, \mathbf{x}_t^* - \mathbf{x}_{t,i} \rangle | \mathbf{x}_{t,i} \Big) \Big)$$

$$= \underbrace{\sum_{i \in \mathcal{N}} \mathbb{E}\Big( \langle \widetilde{\nabla} F_t^s(\mathbf{x}_{t,i}) \odot \mathbf{1}_{\mathcal{V}_i}, \mathbf{x}_t^* - \mathbf{x}_{t+1,i} \rangle \Big)}_{④}$$

$$+ \underbrace{\sum_{i \in \mathcal{N}} \mathbb{E}\Big( \langle \widetilde{\nabla} F_t^s(\mathbf{x}_{t,i}) \odot \mathbf{1}_{\mathcal{V}_i}, \mathbf{x}_{t+1,i} - \mathbf{y}_{t,i} \rangle \Big)}_{⑤}$$

$$+ \underbrace{\sum_{i \in \mathcal{N}} \mathbb{E}\Big( \langle \widetilde{\nabla} F_t^s(\mathbf{x}_{t,i}) \odot \mathbf{1}_{\mathcal{V}_i}, \mathbf{y}_{t,i} - \mathbf{x}_{t,i} \rangle \Big)}_{⑥}. \tag{23}$$

For ④, from Lemma 7, we know that

$$\mathbf{x}_{t+1,i} = \arg\min_{\mathbf{x} \in \mathcal{C}} \Big( - \langle \widetilde{\nabla} F_t^s(\mathbf{x}_{t,i}) \odot \mathbf{1}_{\mathcal{V}_i}, \mathbf{x} \rangle + \frac{1}{\eta_t} \mathcal{D}_\phi(\mathbf{x}, \mathbf{y}_{t,i}) \Big).$$

So, from the Lemma 5, we can show that

$$\eta_t \langle \mathbf{x}_{t+1,i} - \mathbf{x}, -\widetilde{\nabla} F_t^s(\mathbf{x}_{t,i}) \odot \mathbf{1}_{\mathcal{V}_i} \rangle \leq \mathcal{D}_\phi(\mathbf{x}, \mathbf{y}_{t,i}) - \mathcal{D}_\phi(\mathbf{x}, \mathbf{x}_{t+1,i}) - \mathcal{D}_\phi(\mathbf{x}_{t+1,i}, \mathbf{y}_{t,i}), \tag{24}$$

for any $\mathbf{x} \in \mathcal{C}$. If we set $\mathbf{x} = \mathbf{x}_t^*$ in Eq.(24), we have

$$④ \leq \frac{1}{\eta_t} \sum_{i \in \mathcal{N}} \mathbb{E}\Big( \mathcal{D}_\phi(\mathbf{x}_t^*, \mathbf{y}_{t,i}) - \mathcal{D}_\phi(\mathbf{x}_t^*, \mathbf{x}_{t+1,i}) - \mathcal{D}_\phi(\mathbf{x}_{t+1,i}, \mathbf{y}_{t,i}) \Big)$$

$$\leq \frac{1}{\eta_t} \sum_{i \in \mathcal{N}} \mathbb{E}\Big( \mathcal{D}_\phi(\mathbf{x}_t^*, \mathbf{y}_{t,i}) - \mathcal{D}_\phi(\mathbf{x}_t^*, \mathbf{x}_{t+1,i}) \Big) - \sum_{i \in \mathcal{N}} \mathbb{E}\left( \frac{\|\mathbf{x}_{t+1,i} - \mathbf{y}_{t,i}\|}{2\eta_t} \right). \tag{25}$$

For ⑤, by Young's inequality, we have that

$$⑤ \leq \sum_{i \in \mathcal{N}} \mathbb{E}\left( \frac{\|\mathbf{x}_{t+1,i} - \mathbf{y}_{t,i}\|}{2\eta_t} \right) + \sum_{i \in \mathcal{N}} \mathbb{E}\left( \frac{\eta_t}{2} \|\widetilde{\nabla} F_t^s(\mathbf{x}_{t,i}) \otimes \mathbf{1}_{\mathcal{V}_i}\|_* \right),$$

$$\leq \sum_{i \in \mathcal{N}} \mathbb{E}\left( \frac{\|\mathbf{x}_{t+1,i} - \mathbf{y}_{t,i}\|}{2\eta_t} \right) + \frac{\eta_t N G}{2}. \tag{26}$$

For ⑥, we have that,

$$⑥ \leq \Big( \sum_{i \in \mathcal{N}} \mathbb{E}\Big( \|\widetilde{\nabla} F_t^s(\mathbf{x}_{t,i}) \otimes \mathbf{1}_{\mathcal{V}_i}\|_* \|\mathbf{x}_{t,i} - \mathbf{y}_{t,i}\| \mid \mathbf{x}_{t,i}, \forall i \in \mathcal{N} \Big) \Big)$$

$$\leq \Big( \mathbb{E}(\|\tilde{\nabla} F_t^A(\mathbf{x}_{t,i})\|_*) \Big) \Big( \sum_{i \in \mathcal{N}} \mathbb{E}(\|\mathbf{x}_{t,i} - \bar{\mathbf{x}}_t\|) + \mathbb{E}(\|\mathbf{y}_{t,i} - \bar{\mathbf{x}}_t\|) \Big) \tag{27}$$

$$\leq 2 \sum_{\tau=1}^{t} G N^{\frac{3}{2}} \beta^{t-\tau} \eta_\tau.$$

Merging Eq.(21),(23),(22),(25),(26) and (27) into Eq.(20), we have that

$$\Big( \frac{1 - e^{-c}}{c} \Big) F_t(\mathbf{x}_t^*) - F_t(\bar{\mathbf{x}}_t) \leq \langle \nabla F_t^s(\bar{\mathbf{x}}_t), \mathbf{x}_t^* - \bar{\mathbf{x}}_t \rangle$$

$$\leq (3G + LD) \Big( \sum_{\tau=1}^{t} N^{\frac{3}{2}} \beta^{t-\tau} \eta_\tau \Big) + \frac{1}{\eta_t} \sum_{i \in \mathcal{N}} \mathbb{E}\Big( \mathcal{D}_\phi(\mathbf{x}_t^*, \mathbf{y}_{t,i}) - \mathcal{D}_\phi(\mathbf{x}_t^*, \mathbf{x}_{t+1,i}) \Big) + \frac{\eta_t N G}{2}.$$

As a result, we get the result in Lemma 10. □

Next, we prove an upper bound of $\sum_{t=1}^{T} \sum_{i \in \mathcal{N}} \frac{1}{\eta_t} \mathbb{E}\Big(\mathcal{D}_\phi(\mathbf{x}_t^*, \mathbf{y}_{t,i}) - \mathcal{D}_\phi(\mathbf{x}_t^*, \mathbf{x}_{t+1,i})\Big)$, that is,

**Lemma 11.** *If Assumption 1-5 hold, we have that*

$$\sum_{t=1}^{T} \sum_{i \in \mathcal{N}} \frac{1}{\eta_t} \mathbb{E}\Big(\mathcal{D}_\phi(\mathbf{x}_t^*, \mathbf{y}_{t,i}) - \mathcal{D}_\phi(\mathbf{x}_t^*, \mathbf{x}_{t+1,i})\Big) \leq \frac{NR^2}{\eta_{T+1}} + \sum_{t=1}^{T} \frac{KN}{\eta_{t+1}} \|\mathbf{x}_{t+1}^* - \mathbf{x}_t^*\|,$$

*where $\mathbf{x}_t^*$ is the optimal solution of Eq.(3), $R^2 := \sup_{\mathbf{x}, \mathbf{y} \in \mathcal{C}} \mathcal{D}_\phi(\mathbf{x}, \mathbf{y})$, and $\mathcal{C}$ is the constraint set in Eq.(3).*

*Proof.*

$$\sum_{t=1}^{T} \sum_{i \in \mathcal{N}} \frac{1}{\eta_t} \mathbb{E}\Big(\mathcal{D}_\phi(\mathbf{x}_t^*, \mathbf{y}_{t,i}) - \mathcal{D}_\phi(\mathbf{x}_t^*, \mathbf{x}_{t+1,i})\Big)$$

$$= \underbrace{\sum_{t=1}^{T} \sum_{i \in \mathcal{N}} \Big( \frac{1}{\eta_t} \mathbb{E}(\mathcal{D}_\phi(\mathbf{x}_t^*, \mathbf{y}_{t,i})) - \frac{1}{\eta_{t+1}} \mathbb{E}(\mathcal{D}_\phi(\mathbf{x}_{t+1}^*, \mathbf{y}_{t+1,i})) \Big)}_{①}$$

$$+ \underbrace{\sum_{t=1}^{T} \sum_{i \in \mathcal{N}} \Big( \frac{1}{\eta_{t+1}} \mathbb{E}\Big(\mathcal{D}_\phi(\mathbf{x}_{t+1}^*, \mathbf{y}_{t+1,i}) - \mathcal{D}_\phi(\mathbf{x}_t^*, \mathbf{y}_{t+1,i})\Big) \Big)}_{②}$$

$$+ \underbrace{\sum_{t=1}^{T} \sum_{i \in \mathcal{N}} \Big( \frac{1}{\eta_{t+1}} \mathbb{E}\Big(\mathcal{D}_\phi(\mathbf{x}_t^*, \mathbf{y}_{t+1,i}) - \mathcal{D}_\phi(\mathbf{x}_t^*, \mathbf{x}_{t+1,i})\Big) \Big)}_{③}$$

$$+ \underbrace{\sum_{t=1}^{T} \sum_{i \in \mathcal{N}} \Big( \frac{1}{\eta_{t+1}} - \frac{1}{\eta_t} \Big) \mathbb{E}\Big(\mathcal{D}_\phi(\mathbf{x}_t^*, \mathbf{x}_{t+1,i})\Big)}_{④}.$$

Firstly, we have $① \leq \frac{NR^2}{\eta_1}$ and $② \leq \sum_{t=1}^{T} \frac{KN}{\eta_{t+1}} \|\mathbf{x}_{t+1}^* - \mathbf{x}_t^*\|$ from Assumption 4.

Then, from the separate convexity, we have

$$③ = \sum_{t=1}^{T} \sum_{i \in \mathcal{N}} \left( \frac{1}{\eta_{t+1}} \mathbb{E}\Big(\mathcal{D}_\phi(\mathbf{x}_t^*, \mathbf{y}_{t+1,i}) - \mathcal{D}_\phi(\mathbf{x}_t^*, \mathbf{x}_{t+1,i})\Big) \right)$$

$$= \sum_{t=1}^{T} \sum_{i \in \mathcal{N}} \left( \frac{1}{\eta_{t+1}} \mathbb{E}\Big(\mathcal{D}_\phi(\mathbf{x}_t^*, \sum_{j \in \mathcal{N}_i \cup \{i\}} w_{ij} \mathbf{x}_{t+1,j}) - \mathcal{D}_\phi(\mathbf{x}_t^*, \mathbf{x}_{t+1,i})\Big) \right)$$

$$\leq \sum_{t=1}^{T} \left( \frac{1}{\eta_{t+1}} \mathbb{E}\Big( \sum_{i \in \mathcal{N}} \sum_{j \in \mathcal{N}_i \cup \{i\}} \Big( w_{ij} \mathcal{D}_\phi(\mathbf{x}_t^*, \mathbf{x}_{t+1,j}) \Big) - \sum_{i \in \mathcal{N}} \mathcal{D}_\phi(\mathbf{x}_t^*, \mathbf{x}_{t+1,i}) \Big) \right)$$

$$= \sum_{t=1}^{T} \left( \frac{1}{\eta_{t+1}} \mathbb{E}\Big( \sum_{i \in \mathcal{N}} \Big( (\sum_{j \in \mathcal{N}_i \cup \{i\}} w_{ji}) \mathcal{D}_\phi(\mathbf{x}_t^*, \mathbf{x}_{t+1,i}) \Big) - \sum_{i \in \mathcal{N}} D_\phi(\mathbf{x}_t^*, \mathbf{x}_{t+1,i}) \Big) \right)$$

$$= 0,$$

where the first inequality follows from Assumption 3, and the third inequality is due to $w_{ij} = w_{ji}$.

Moreover, we have $④ \leq NR^2 \Big( \frac{1}{\eta_{T+1}} - \frac{1}{\eta_1} \Big)$. We finally get

$$\sum_{t=1}^{T} \sum_{i \in \mathcal{N}} \frac{1}{\eta_t} \mathbb{E}\Big(D_\phi(\mathbf{x}^*, \mathbf{y}_{t,i}) - D_\phi(\mathbf{x}^*, \mathbf{x}_{t+1,i})\Big) \leq \frac{NR^2}{\eta_{T+1}} + \sum_{t=1}^{T} \frac{KN}{\eta_{t+1}} \|\mathbf{x}_{t+1}^* - \mathbf{x}_t^*\|.$$

$\square$

As a result, we can prove the following Lemma:

**Lemma 12.** *If Assumption 1-5 hold and each set function $f_t$ is monotone submodular with curvature $c$ for any $t \in [T]$, then*

$$\Big(\frac{1 - e^{-c}}{c}\Big) \sum_{t=1}^{T} F_t(\mathbf{x}_t^*) - \sum_{t=1}^{T} \mathbb{E}\Big(F_t\big(\sum_{i \in \mathcal{N}} \mathbf{x}_{t,i} \odot \mathbf{1}_{\mathcal{V}_i}\big)\Big)$$
$$\leq (4G + LDG)\Big(\sum_{t=1}^{T} \sum_{\tau=1}^{t} N^{\frac{3}{2}} \beta^{t-\tau} \eta_\tau\Big) + \frac{NR^2}{\eta_{T+1}} + \sum_{t=1}^{T} \frac{KN}{\eta_{t+1}} \|\mathbf{x}_{t+1}^* - \mathbf{x}_t^*\| + \frac{NG}{2} \sum_{t=1}^{T} \eta_t.$$

*Proof.* From the Lemma 10 and Lemma 11, we have that

$$\Big(\frac{1 - e^{-c}}{c}\Big) \sum_{t=1}^{T} F_t(\mathbf{x}_t^*) - \sum_{t=1}^{T} \mathbb{E}\Big(F_t(\bar{\mathbf{x}}_t)\Big)$$
$$\leq (3G + LDG)\Big(\sum_{t=1}^{T} \sum_{\tau=1}^{t} N^{\frac{3}{2}} \beta^{t-\tau} \eta_\tau\Big) + \frac{NR^2}{\eta_{T+1}} + \sum_{t=1}^{T} \frac{KN}{\eta_{t+1}} \|\mathbf{x}_{t+1}^* - \mathbf{x}_t^*\| + \frac{NG}{2} \sum_{t=1}^{T} \eta_t.$$

From the Assumption 5, we also can show that $|F_t(\mathbf{x}) - F_t(\mathbf{y})| \leq G\|\mathbf{x} - \mathbf{y}\|$ for any $t \in [T]$ such that we have $|F_t(\sum_{i \in \mathcal{N}} \mathbf{x}_{t,i} \odot \mathbf{1}_{\mathcal{V}_i}) - F_t(\bar{\mathbf{x}}_t)| \leq G\|\sum_{i \in \mathcal{N}} \mathbf{x}_{t,i} \odot \mathbf{1}_{\mathcal{V}_i} - \bar{\mathbf{x}}_t\| \leq G \sum_{i \in \mathcal{N}} \|\mathbf{x}_{t,i} - \bar{\mathbf{x}}_t\| \leq G\Big(\sum_{\tau=1}^{t} N^{\frac{3}{2}} \beta^{t-\tau} \eta_\tau\Big)$. Thus, we have that

$$\Big(\frac{1 - e^{-c}}{c}\Big) \sum_{t=1}^{T} F_t(\mathbf{x}_t^*) - \sum_{t=1}^{T} \mathbb{E}\Big(F_t\big(\sum_{i \in \mathcal{N}} \mathbf{x}_{t,i} \odot \mathbf{1}_{\mathcal{V}_i}\big)\Big)$$
$$\leq \Big(\frac{1 - e^{-c}}{c}\Big) \sum_{t=1}^{T} F_t(\mathbf{x}_t^*) - \sum_{t=1}^{T} \mathbb{E}\Big(F_t(\bar{\mathbf{x}}_t)\Big) + \big|\sum_{t=1}^{T} \mathbb{E}\Big(F_t\big(\sum_{i \in \mathcal{N}} \mathbf{x}_{t,i} \odot \mathbf{1}_{\mathcal{V}_i}\big) - F_t\big(\bar{\mathbf{x}}_t\big)\Big)\big|$$
$$\leq (4G + LDG)\Big(\sum_{t=1}^{T} \sum_{\tau=1}^{t} N^{\frac{3}{2}} \beta^{t-\tau} \eta_\tau\Big) + \frac{NR^2}{\eta_{T+1}} + \sum_{t=1}^{T} \frac{KN}{\eta_{t+1}} \|\mathbf{x}_{t+1}^* - \mathbf{x}_t^*\| + \frac{NG}{2} \sum_{t=1}^{T} \eta_t.$$

$\square$

From Calinescu et al. (2011); Chekuri et al. (2014), we know that the optimal value of continuous problem Eq.(3) is equal to the optimal value of the corresponding discrete submodular maximization Eq.(1), so we can set $\mathbf{x}_t^* := \mathbf{1}_{\mathcal{A}_t^*}$ where $\mathcal{A}_t^*$ is the maximizer of Eq.(1).

Next, we show a relationship between $\mathbb{E}\big(f_t\big(\cup_{i \in \mathcal{N}} \{a_{t,i}\}\big)\big)$ and $\mathbb{E}\big(F_t\big(\sum_{i \in \mathcal{N}} \mathbf{x}_{t,i} \odot \mathbf{1}_{\mathcal{V}_i}\big)\big)$.

**Lemma 13.** *If the function $f_t$ is monotone submodular and $a_{t,i}$ is the action taken by the agent $i \in \mathcal{N}$ at time $t$, then we have*

$$\mathbb{E}\Big(f_t\big(\cup_{i \in \mathcal{N}} \{a_{t,i}\}\big)\Big) \geq \mathbb{E}\Big(\big(F_t\big(\sum_{i \in \mathcal{N}} \mathbf{x}_{t,i} \odot \mathbf{1}_{\mathcal{V}_i}\big)\Big).$$

*Proof.* We prove this lemma by induction on $N = |\mathcal{N}|$. When $N = 1$, for any $t \in [T]$, we have that

$$\begin{aligned}
\mathbb{E}(F_t(\mathbf{x}_{t,1})) &\leq \mathbb{E}(F_t(\frac{\mathbf{x}_{t,1}}{\|\mathbf{x}_{t,1}\|_1})) \leq \mathbb{E}_{\mathcal{R} \sim \frac{\mathbf{x}_{t,1}}{\|\mathbf{x}_{t,1}\|_1}}\Big(f_t(\mathcal{R})\Big) \\
&\leq \mathbb{E}_{\mathcal{R} \sim \frac{\mathbf{x}_{t,1}}{\|\mathbf{x}_{t,1}\|_1}}\Big(\sum_{a \in \mathcal{R}} f_t(a)\Big) \\
&= \sum_{a \in \mathcal{V}} \frac{[\mathbf{x}_{t,1}]_a}{\|\mathbf{x}_{t,1}\|_1} f_t(a) \\
&= \mathbb{E}(f_t(a_{t,1})),
\end{aligned} \tag{28}$$

where the first inequality follows from the monotonicity of $f_t$ and $\|\mathbf{x}_{t,1}\|_1 \leq 1$; the second one from the submodularity of $f_t$ and Line 5-6 in Algorithm 1.

Now let $N > 1$. For any $\mathcal{R} \subseteq \mathcal{V}$, we denote $\mathcal{R}_1 = \mathcal{R} \setminus \mathcal{V}_N$ and $s_i = \sum_{a \in \mathcal{V}_i} [\mathbf{x}_{t,i}]_a$ for any $i \in \mathcal{N}$. Then, we have that

$$
\begin{aligned}
\mathbb{E}\Big(F_t\big(\sum_{i \in \mathcal{N}} \mathbf{x}_{t,i} \odot \mathbf{1}_{\mathcal{V}_i}\big)\Big) &\leq \mathbb{E}\Big(F_t\big(\sum_{i \in \mathcal{N}} \frac{\mathbf{x}_{t,i}}{s_i} \odot \mathbf{1}_{\mathcal{V}_i}\big)\Big) = \mathbb{E}_{\mathcal{R} \sim \sum_{i \in \mathcal{N}} \frac{\mathbf{x}_{t,i}}{s_i} \odot \mathbf{1}_{\mathcal{V}_i}}\Big(f_t(\mathcal{R})\Big) \\
&= \mathbb{E}_{\mathcal{R} \sim \sum_{i \in \mathcal{N}} \frac{\mathbf{x}_{t,i}}{s_i} \odot \mathbf{1}_{\mathcal{V}_i}}\Big(f_t(\mathcal{R}_1) + f_t(\mathcal{R}) - f_t(\mathcal{R}_1)\Big) \\
&= \mathbb{E}_{\mathcal{R} \sim \sum_{i \in \mathcal{N}} \frac{\mathbf{x}_{t,i}}{s_i} \odot \mathbf{1}_{\mathcal{V}_i}}\Big(f_t(\mathcal{R}) - f_t(\mathcal{R}_1)\Big) + \mathbb{E}_{\mathcal{R} \sim \sum_{i \in \mathcal{N}} \frac{\mathbf{x}_{t,i}}{s_i} \odot \mathbf{1}_{\mathcal{V}_i}}\Big(f_t(\mathcal{R}_1)\Big) \\
&= \mathbb{E}\Big(\mathbb{E}_{\mathcal{R} \sim \sum_{i \in \mathcal{N}} \frac{\mathbf{x}_{t,i}}{s_i} \odot \mathbf{1}_{\mathcal{V}_i}}\Big(f_t(\mathcal{R}) - f_t(\mathcal{R}_1)\Big|\mathcal{R}_1\Big)\Big) + \mathbb{E}_{\mathcal{R} \sim \sum_{i \in \mathcal{N}} \frac{\mathbf{x}_{t,i}}{s_i} \odot \mathbf{1}_{\mathcal{V}_i}}\Big(f_t(\mathcal{R}_1)\Big) \\
&\leq \mathbb{E}\Big(\mathbb{E}_{\mathcal{R} \sim \sum_{i \in \mathcal{N}} \frac{\mathbf{x}_{t,i}}{s_i} \odot \mathbf{1}_{\mathcal{V}_i}}\Big(f_t(\mathcal{R}_1 \cup \{a_{t,N}\}) - f_t(\mathcal{R}_1)\Big|\mathcal{R}_1\Big)\Big) + \mathbb{E}_{\mathcal{R} \sim \sum_{i \in \mathcal{N}} \frac{\mathbf{x}_{t,i}}{s_i} \odot \mathbf{1}_{\mathcal{V}_i}}\Big(f_t(\mathcal{R}_1)\Big) \\
&= \mathbb{E}_{\mathcal{R} \sim \sum_{i \in \mathcal{N}} \frac{\mathbf{x}_{t,i}}{s_i} \odot \mathbf{1}_{\mathcal{V}_i}}\Big(f_t(\mathcal{R}_1 \cup \{a_{t,N}\})\Big) \\
&\leq \mathbb{E}\Big(f_t\big(\cup_{i \in \mathcal{N}} \{a_{t,i}\}\big)\Big),
\end{aligned}
$$

where the first inequality follows from the monotonicity of $f_t$ and $s_i \leq 1$ and we get the second inequality follows from repeating the proof in Eq.(28) because $f_t(\mathcal{R}) - f_t(\mathcal{R}_1) = f_t(\mathcal{R}) - f_t(\mathcal{R} \setminus \mathcal{V}_N)$ is a submodular function over $\mathcal{V}_N$ for any fixed $\mathcal{R} \subseteq \mathcal{V}$ and $a_{t,N}$ is selected from the set $\mathcal{V}_N$ according to $\frac{[\mathbf{x}_{t,N}]_{\mathcal{V}_N}}{s_N}$. $\qquad\square$

Merging Lemma 13 into Lemma 12, we can get that

$$
\begin{aligned}
&\Big(\frac{1 - e^{-c}}{c}\Big) \sum_{t=1}^{T} f_t(\mathcal{A}_t^*) - \sum_{t=1}^{T} \mathbb{E}\Big(f_t\big(\cup_{i \in \mathcal{N}} \{a_{t,i}\}\big)\Big) \\
&\leq (4G + LDG)\Big(\sum_{t=1}^{T} \sum_{\tau=1}^{t} N^{\frac{3}{2}} \beta^{t-\tau} \eta_\tau\Big) + \frac{NR^2}{\eta_{T+1}} + \sum_{t=1}^{T} \frac{KN}{\eta_{t+1}} \|\mathbf{1}_{\mathcal{A}_{t+1}^*} - \mathbf{1}_{\mathcal{A}_t^*}\| + \frac{NG}{2} \sum_{t=1}^{T} \eta_t.
\end{aligned}
$$

We know that all norms in finite-dimensional real space are equivalent (Lax, 2014; Zhao & Lai, 2024), so $\|\mathbf{x}\| \leq C_2 \|\mathbf{x}\|_1$. Finally, we can verify the Eq.(8) in Theorem 3, that is,

$$
\begin{aligned}
&\Big(\frac{1 - e^{-c}}{c}\Big) \sum_{t=1}^{T} f_t(\mathcal{A}_t^*) - \sum_{t=1}^{T} \mathbb{E}\Big(f_t\big(\cup_{i \in \mathcal{N}} \{a_{t,i}\}\big)\Big) \\
&\leq (4G + LDG)\Big(\sum_{t=1}^{T} \sum_{\tau=1}^{t} N^{\frac{3}{2}} \beta^{t-\tau} \eta_\tau\Big) + \frac{NR^2}{\eta_{T+1}} + \sum_{t=1}^{T} \frac{KNC_2}{\eta_{t+1}} |\mathcal{A}_{t+1}^* \Delta \mathcal{A}_t^*| + \frac{NG}{2} \sum_{t=1}^{T} \eta_t.
\end{aligned}
$$

## E   PROOF OF THEOREM 4

In this section, we prove the Theorem 4.

*Proof.* When $g(x) = x \log(x)$, we know that, for any $\mathbf{b}, \mathbf{y} \in (0,1)^m$

$$
\mathcal{D}_{g,m}(\mathbf{b}, \mathbf{y}) = \sum_{i=1}^{m} \Big(b_i \log(\frac{b_i}{y_i})\Big) - \sum_{i=1}^{m} b_i + \sum_{i=1}^{m} y_i.
$$

Next, we consider the Lagrangian function, for any fixed $\mathbf{y}, \mathbf{z} \in (0,1)^n$,

$$
L(\mathbf{b}, \lambda) = \sum_{i=1}^{m} z_i b_i + \sum_{i=1}^{m} \Big(b_i \log(\frac{b_i}{y_i})\Big) - \sum_{i=1}^{m} b_i + \sum_{i=1}^{m} y_i + \lambda(\sum_{i=1}^{m} b_i - 1).
$$

Then, we have that

$$
\frac{\partial L(\mathbf{b}, \lambda)}{\partial b_i} = z_i + \log(\frac{b_i}{y_i}) + \lambda, \ \forall i \in [m]. \tag{29}
$$

Setting all equations in Eq.(29) to 0, we can get $b_i = y_i \exp(-z_i) \exp(-\lambda)$ for any $i \in [m]$ such that $L(\lambda) = -\sum_{i=1}^{m} \left( y_i \exp(-z_i) \right) \exp(-\lambda) - \lambda$. When $\sum_{i=1}^{m} \left( y_i \exp(-z_i) \right) \leq 1$, $L(0) = \max_{\lambda \geq 0} L(\lambda)$ such that the optimal solution $b_i^* = y_i \exp(-z_i)$. Similarly, when $\sum_{i=1}^{m} \left( y_i \exp(-z_i) \right) > 1$,

$$L \left( \log \left( \sum_{i=1}^{m} \left( y_i \exp(-z_i) \right) \right) \right) = \max_{\lambda \geq 0} L(\lambda),$$

so $b_i^* = y_i \exp(-z_i) \exp \left( -\log \left( \sum_{i=1}^{m} \left( y_i \exp(-z_i) \right) \right) \right) = \frac{y_i \exp(-z_i)}{\sum_{i=1}^{m} \left( y_i \exp(-z_i) \right)}.$ $\qquad \square$

## F    PROOF OF THEOREM 5

In this section, we present the proof of Theorem 5. Specially, we assume $\| \cdot \|$ is $l_1$ norm in this section. Like the Appendix D, we can show that

**Lemma 14.** *In Algorithm 2, if we set the constraint $\mathcal{C} = \{ \mathbf{x} \in [0,1]^n : \sum_{a \in \mathcal{V}_i} x_a \leq 1, \forall i \in \mathcal{N} \}$ and Assumption 2 holds,we have that, for any $t \in [T]$ and $i \in \mathcal{N}$, we have that, for any $t \in [T]$ and $i \in \mathcal{N}$, $\mathbf{x}_{t,i} \in \mathcal{C}$ and $\mathbf{y}_{t,i} \in \mathcal{C}$.*

**Lemma 15.** *In Algorithm 2, if we set the constraint $\mathcal{C} = \{ \mathbf{x} \in [0,1]^n : \sum_{a \in \mathcal{V}_i} x_a \leq 1, \forall i \in \mathcal{N} \}$ and Assumption 1 and 2 hold,we have that, for any $t \in [T]$ and $i \in \mathcal{N}$,*

$$\mathbf{x}_{t+1,i} = \arg\min_{\mathbf{x} \in \mathcal{C}} \left( -\langle \widetilde{\nabla} F_t^s(\mathbf{x}_{t,i}) \odot \mathbf{1}_{\mathcal{V}_i}, \mathbf{x} \rangle + \frac{1}{\eta_t} \mathcal{D}_{KL}(\mathbf{x}, \mathbf{y}_{t,i}) \right),$$

*where $\odot$ denotes the coordinate-wise multiplication, i.e.,the $i$-th element of vector $\mathbf{x} \odot \mathbf{y}$ is $x_i y_i$, and $\mathbf{1}_{\mathcal{V}_i}$ denotes a $n$-dimensional vector where the entries at $\mathcal{V}_i$ is equal to $1$ and all others are $0$.*

**Lemma 16.** *In Algorithm 2, if we set the constraint $\mathcal{C} = \{ \mathbf{x} \in [0,1]^n : \sum_{a \in \mathcal{V}_i} x_a \leq 1, \forall i \in \mathcal{N} \}$ and Assumption 1 and 5 hold, we have that*

$$\mathbb{E}(\|\mathbf{r}_{t,i}\|_1) = \mathbb{E}(\|\mathbf{x}_{t+1,i} - \mathbf{y}_{t,i}\|_1) \leq G\eta_t.$$

Moreover, we also define the following new symbols for Algorithm 2:

$$\bar{\mathbf{x}}_t = \frac{\sum_{i=1}^{N} \mathbf{x}_{t,i}}{N}, \quad \mathbf{x}_t^{cate} = [\mathbf{x}_{t,1}; \mathbf{x}_{t,2}; \ldots; \mathbf{x}_{t,N}] \in \mathbb{R}^{n*N};$$

$$\bar{\mathbf{y}}_t = \frac{\sum_{i=1}^{N} \mathbf{y}_{t,i}}{N}, \quad \mathbf{y}_t^{cate} = [\mathbf{y}_{t,1}; \mathbf{y}_{t,2}; \ldots; \mathbf{y}_{t,N}] \in \mathbb{R}^{n*N};$$

$$\bar{\hat{\mathbf{x}}}_t = \frac{\sum_{i=1}^{N} \hat{\mathbf{x}}_{t,i}}{N}, \quad \hat{\mathbf{x}}_t^{cate} = [\hat{\mathbf{x}}_{t,1}; \hat{\mathbf{x}}_{t,2}; \ldots; \hat{\mathbf{x}}_{t,N}] \in \mathbb{R}^{n*N};$$

$$\mathbf{r}_{t,i} = \mathbf{x}_{t+1,i} - \mathbf{y}_{t,i}, \quad \mathbf{r}_t^{cate} = [\mathbf{r}_{t,1}; \mathbf{r}_{t,2}; \ldots; \mathbf{r}_{t,N}] \in \mathbb{R}^{n*N};$$

Then, we also can show that

**Lemma 17.** *In Algorithm 2, under the Assumption 1, 2 and 5, we have that, for any $t \in [T]$ and $i \in \mathcal{N}$,*

$$\mathbb{E}((\|\mathbf{x}_{t+1,i} - \bar{\mathbf{x}}_{t+1}\|_1) \leq \sum_{\tau=1}^{t} \sqrt{N} \beta^{t-\tau} (1-\gamma)^{t-\tau} \eta_\tau G,$$

$$\mathbb{E}((\|\mathbf{y}_{t+1,i} - \bar{\mathbf{x}}_{t+1}\|_1) \leq \sum_{\tau=1}^{t} \sqrt{N} \beta^{t-\tau} (1-\gamma)^{t+1-\tau} \eta_\tau G + \gamma D,$$

*where $\beta = \max(|\lambda_2(\mathbf{W})|, |\lambda_N(\mathbf{W})|)$ is the second largest magnitude of the eigenvalues of the weight matrix $\mathbf{W}$ and $D = \sup_{\mathbf{x}, \mathbf{y} \in \mathcal{C}} \|\mathbf{x} - \mathbf{y}\|_1$ where $\mathcal{C} = \{ \mathbf{x} \in [0,1]^n : \sum_{a \in \mathcal{V}_i} x_a \leq 1, \forall i \in \mathcal{N} \}$.*

*Proof.* From the definition of $\mathbf{r}_{t,i}$ and $\hat{\mathbf{x}}_{t,i}$, we can conclude that

$$
\begin{aligned}
\mathbf{x}_{t+1,i} &= \mathbf{r}_{t,i} + \mathbf{y}_{t,i} = \mathbf{r}_{t,i} + \sum_{j \in \mathcal{N}_i \cup \{i\}} w_{ij} \hat{\mathbf{x}}_{t,j} \\
&= \mathbf{r}_{t,i} + \sum_{j \in \mathcal{N}_i \cup \{i\}} w_{ij} \left( (1-\gamma)\mathbf{x}_{t,i} + \frac{\gamma}{n} \mathbf{1}_n \right).
\end{aligned}
\tag{30}
$$

where the final equality follows from step 8 in Algorithm 2.

As a result, from the Eq.(30), we can show that

$$
\begin{aligned}
\mathbf{x}_{t+1}^{cate} &= \mathbf{r}_t^{cate} + (\mathbf{W} \otimes \mathbf{I}_n)\left( (1-\gamma)\mathbf{x}_t^{cate} + \frac{\gamma}{n} \mathbf{1}_{nN} \right) \\
&= \mathbf{r}_t^{cate} + (1-\gamma)(\mathbf{W} \otimes \mathbf{I}_n)\mathbf{x}_t^{cate} + \frac{\gamma}{n} \mathbf{1}_{nN} \\
&= \sum_{\tau=1}^{t} (1-\gamma)^{t-\tau}(\mathbf{W} \otimes \mathbf{I}_n)^{t-\tau}\left( \mathbf{r}_\tau^{cate} + \frac{\gamma}{n} \mathbf{1}_{nN} \right) \\
&= \sum_{\tau=1}^{t} (1-\gamma)^{t-\tau}(\mathbf{W}^{t-\tau} \otimes \mathbf{I}_n)\mathbf{r}_\tau^{cate} + \sum_{\tau=1}^{t} \frac{\gamma(1-\gamma)^{t-\tau}}{n} \mathbf{1}_{nN}.
\end{aligned}
\tag{31}
$$

If we define $\bar{\mathbf{x}}_t^{cate} = [\bar{\mathbf{x}}_t; \bar{\mathbf{x}}_t; \ldots; \bar{\mathbf{x}}_t] \in \mathbb{R}^{n*N}$ and from the Eq.(31), we also have that

$$
\begin{aligned}
\bar{\mathbf{x}}_{t+1}^{cate} &= \left( \frac{\mathbf{1}_N \mathbf{1}_N^T}{N} \otimes \mathbf{I}_N \right)\mathbf{x}_{t+1}^{cate} \\
&= \sum_{\tau=1}^{t} (1-\gamma)^{t-\tau}\left( \frac{\mathbf{1}_N \mathbf{1}_N^T}{N} \otimes \mathbf{I}_n \right)\mathbf{r}_\tau^{cate} + \sum_{\tau=1}^{t} \frac{\gamma(1-\gamma)^{t-\tau}}{n} \mathbf{1}_{nN}.
\end{aligned}
\tag{32}
$$

Then, from the Eq.(31) and Eq.(32), we have that , for any $i \in \mathcal{N}$,

$$
\mathbf{x}_{t+1,i} - \bar{\mathbf{x}}_{t+1} = \sum_{\tau=1}^{t} \sum_{j \in \mathcal{N}_i \cup \{i\}} (1-\gamma)^{t-\tau}\left( [\mathbf{W}^{t-\tau}]_{ij} - \frac{1}{N} \right)\mathbf{r}_{\tau,j}.
\tag{33}
$$

Eq.(33) indicates that

$$
\begin{aligned}
\mathbb{E}(\|\mathbf{x}_{t+1,i} - \bar{\mathbf{x}}_{t+1}\|_1) &= \mathbb{E}\left(\left\| \sum_{\tau=1}^{t} \sum_{j \in \mathcal{N}_i \cup \{i\}} (1-\gamma)^{t-\tau}\left( [\mathbf{W}^{t-\tau}]_{ij} - \frac{1}{N} \right)\mathbf{r}_{\tau,j} \right\|_1\right) \\
&\leq \mathbb{E}\left( \sum_{\tau=1}^{t} \sum_{j \in \mathcal{N}_i \cup \{i\}} (1-\gamma)^{t-\tau}\left| [\mathbf{W}^{t-\tau}]_{ij} - \frac{1}{N} \right| \|\mathbf{r}_{\tau,j}\|_1 \right) \\
&\leq \sum_{\tau=1}^{t} \sum_{j \in \mathcal{N}_i \cup \{i\}} \left| [\mathbf{W}^{t-\tau}]_{ij} - \frac{1}{N} \right|(1-\gamma)^{t-\tau}\eta_\tau G \\
&\leq \sum_{\tau=1}^{t} \sqrt{N}\beta^{t-\tau}(1-\gamma)^{t-\tau}\eta_\tau G,
\end{aligned}
$$

where the second inequality comes from Lemma 16 and the final inequality follows from $\sum_{j \in \mathcal{N}_i \cup \{i\}} \left| [\mathbf{W}^{t-\tau}]_{ij} - \frac{1}{N} \right| \leq \sqrt{N}\beta^{t-\tau}$. Due to $\mathbf{y}_{t+1,i} = \sum_{j \in \mathcal{N}_i \cup \{i\}} w_{ij}\hat{\mathbf{x}}_{t+1,j}$ we also can

have

$$\mathbb{E}((\|\mathbf{y}_{t+1,i} - \bar{\mathbf{x}}_{t+1}\|_1) \leq \sum_{j \in \mathcal{N}_i \cup \{i\}} w_{ij} \mathbb{E}((\|\hat{\mathbf{x}}_{t+1,j} - \bar{\mathbf{x}}_{t+1}\|_1)$$

$$= \sum_{j \in \mathcal{N}_i \cup \{i\}} w_{ij} \mathbb{E}((\|\big((1-\gamma)\mathbf{x}_{t,i} + \frac{\gamma}{n}\mathbf{1}_n\big) - \bar{\mathbf{x}}_{t+1}\|_1)$$

$$= (1-\gamma) \sum_{j \in \mathcal{N}_i \cup \{i\}} w_{ij} \mathbb{E}((\|\mathbf{x}_{t,i} - \bar{\mathbf{x}}_{t+1}\|_1) + \gamma \sum_{j \in \mathcal{N}_i \cup \{i\}} w_{ij} \mathbb{E}(\|\frac{1}{n}\mathbf{1}_n - \bar{\mathbf{x}}_{t+1}\|_1)$$

$$\leq \sum_{\tau=1}^{t} \sqrt{N} \beta^{t-\tau} (1-\gamma)^{t+1-\tau} \eta_\tau G + \gamma D.$$

$\square$

Like the Lemma 10, we can present a similar lemma for Algorithm 2, that is,

**Lemma 18.** *Consider our proposed Algorithm 2, if Assumption 1,2,3,5 hold and each set function $f_t$ is monotone submodular with curvature $c$ for any $t \in [T]$, then we can conclude that,*

$$\Big(\frac{1 - e^{-c}}{c}\Big) \sum_{t=1}^{T} F_t(\mathbf{x}^{*t}) - \sum_{t=1}^{T} \mathbb{E}\Big(F_t(\bar{\mathbf{x}}_t)\Big) \leq (2G + LD)\Big(\sum_{t=1}^{T} \sum_{i \in \mathcal{N}} \|\mathbf{x}_{t,i} - \bar{\mathbf{x}}_t\|_1\Big)$$

$$+ G\Big(\sum_{t=1}^{T} \sum_{i \in \mathcal{N}} \|\mathbf{y}_{t,i} - \bar{\mathbf{x}}_t\|_1\Big) + \sum_{t=1}^{T} \sum_{i \in \mathcal{N}} \frac{1}{\eta_t} \mathbb{E}\Big(D_{KL}(\mathbf{x}^*, \mathbf{y}_{t,i}) - D_{KL}(\mathbf{x}^*, \mathbf{x}_{t+1,i})\Big) + \frac{NG}{2} \sum_{t=1}^{T} \eta_t,$$

*where $\mathbf{x}_t^*$ is the optimal solution of Eq.(3) and $D = \sup_{\mathbf{x},\mathbf{y} \in \mathcal{C}} \|\mathbf{x} - \mathbf{y}\|$ where $\mathcal{C} = \{\mathbf{x} \in [0,1]^n : \sum_{a \in \mathcal{V}_i} x_a \leq 1, \forall i \in \mathcal{N}\}$.*

Then, we derive an upper bound for $\sum_{t=1}^{T} \sum_{i \in \mathcal{N}} \frac{1}{\eta_t} \mathbb{E}\Big(\mathcal{D}_{KL}(\mathbf{x}_t^*, \mathbf{y}_{t,i}) - \mathcal{D}_{KL}(\mathbf{x}_t^*, \mathbf{x}_{t+1,i})\Big)$, i.e.,

**Lemma 19.** *If Assumption 1,2,3 and 5 hold, we have that*

$$\sum_{t=1}^{T} \sum_{i \in \mathcal{N}} \frac{1}{\eta_t} \mathbb{E}\Big(\mathcal{D}_{KL}(\mathbf{x}_t^*, \mathbf{y}_{t,i}) - \mathcal{D}_{KL}(\mathbf{x}_t^*, \mathbf{x}_{t+1,i})\Big)$$

$$\leq \sum_{t=1}^{T} \sum_{i \in \mathcal{N}} \frac{1}{\eta_t} \mathbb{E}\Bigg(\sum_{j=1}^{n} [\mathbf{x}_t^*]_j \log\Big(\frac{[\hat{\mathbf{x}}_{t+1,i}]_j}{[\mathbf{y}_{t,i}]_j}\Big) + \sum_{j=1}^{n} \Big([\mathbf{y}_{t,i}]_j - [\mathbf{x}_{t+1,i}]_j\Big)\Bigg) + \sum_{t=1}^{T} \frac{2N^2\gamma}{\eta_t}.$$

*Proof.* From Nesterov (2013), we know that, for any two vector $\mathbf{x}, \mathbf{y} \in \mathbb{R}_+^n$,

$$\mathcal{D}_{KL}(\mathbf{x}, \mathbf{y}) = \sum_{j=1}^{n} [\mathbf{x}]_j \log\Big(\frac{[\mathbf{x}]_j}{[\mathbf{y}]_j}\Big) - \sum_{j=1}^{n} [\mathbf{x}]_j + \sum_{j=1}^{n} [\mathbf{y}]_j.$$

Thus, we can show that

$$\sum_{t=1}^{T} \sum_{i \in \mathcal{N}} \frac{1}{\eta_t} \mathbb{E}\Big(\mathcal{D}_{KL}(\mathbf{x}_t^*, \mathbf{y}_{t,i}) - \mathcal{D}_{KL}(\mathbf{x}_t^*, \mathbf{x}_{t+1,i})\Big)$$

$$= \sum_{t=1}^{T} \sum_{i \in \mathcal{N}} \frac{1}{\eta_t} \mathbb{E}\Bigg(\sum_{j=1}^{n} [\mathbf{x}_t^*]_j \log\Big(\frac{[\mathbf{x}_{t+1,i}]_j}{[\mathbf{y}_{t,i}]_j}\Big) + \sum_{j=1}^{n} \Big([\mathbf{y}_{t,i}]_j - [\mathbf{x}_{t+1,i}]_j\Big)\Bigg)$$

$$= \sum_{t=1}^{T} \sum_{i \in \mathcal{N}} \frac{1}{\eta_t} \mathbb{E}\Bigg(\sum_{j=1}^{n} [\mathbf{x}_t^*]_j \log\Big(\frac{[\hat{\mathbf{x}}_{t+1,i}]_j}{[\mathbf{y}_{t,i}]_j}\Big) + \sum_{j=1}^{n} [\mathbf{x}_t^*]_j \log\Big(\frac{[\mathbf{x}_{t+1,i}]_j}{[\hat{\mathbf{x}}_{t+1,i}]_j}\Big) + \sum_{j=1}^{n} \Big([\mathbf{y}_{t,i}]_j - [\mathbf{x}_{t+1,i}]_j\Big)\Bigg).$$

From Line 7 in Algorithm 2, we know that $[\hat{\mathbf{x}}_{t+1,i}]_j = (1-\gamma)[\mathbf{x}_{t+1,i}]_j + \frac{\gamma}{n}$. So if $[\mathbf{x}_{t+1,i}]_j \leq \frac{1}{n}$, $[\mathbf{x}_{t+1,i}]_j \leq [\hat{\mathbf{x}}_{t+1,i}]_j$ or $\log(\frac{[\mathbf{x}_{t+1,i}]_j}{[\hat{\mathbf{x}}_{t+1,i}]_j}) \leq 0$. As for $[\mathbf{x}_{t+1,i}]_j > \frac{1}{n}$ and $\gamma \leq \frac{1}{2}$,

$$\log(\frac{[\mathbf{x}_{t+1,i}]_j}{[\hat{\mathbf{x}}_{t+1,i}]_j}) = \log(\frac{[\mathbf{x}_{t+1,i}]_j}{(1-\gamma)[\mathbf{x}_{t+1,i}]_j + \frac{\gamma}{n}}) = \log\left(1 + \frac{\gamma([\mathbf{x}_{t+1,i}]_j - \frac{1}{n})}{(1-\gamma)[\mathbf{x}_{t+1,i}]_j + \frac{\gamma}{n}}\right)$$
$$\leq \frac{\gamma([\mathbf{x}_{t+1,i}]_j - \frac{1}{n})}{(1-\gamma)[\mathbf{x}_{t+1,i}]_j + \frac{\gamma}{n}} \leq 2\gamma,$$

where the final inequality follows from $[\mathbf{x}_{t+1,i}]_j - \frac{1}{n} \leq [\mathbf{x}_{t+1,i}]_j \leq 2(1-\gamma)[\mathbf{x}_{t+1,i}]_j \leq 2\left((1-\gamma)[\mathbf{x}_{t+1,i}]_j + \frac{\gamma}{n}\right)$.

Then, we have

$$\sum_{t=1}^{T}\sum_{i\in\mathcal{N}}\frac{1}{\eta_t}\mathbb{E}\Big(\mathcal{D}_{KL}(\mathbf{x}_t^*, \mathbf{y}_{t,i}) - \mathcal{D}_{KL}(\mathbf{x}_t^*, \mathbf{x}_{t+1,i})\Big)$$
$$\leq \sum_{t=1}^{T}\sum_{i\in\mathcal{N}}\frac{1}{\eta_t}\mathbb{E}\left(\sum_{j=1}^{n}[\mathbf{x}_t^*]_j\log(\frac{[\hat{\mathbf{x}}_{t+1,i}]_j}{[\mathbf{y}_{t,i}]_j}) + \sum_{j=1}^{n}\Big([\mathbf{y}_{t,i}]_j - [\mathbf{x}_{t+1,i}]_j\Big)\right) + \sum_{t=1}^{T}\frac{2N^2\gamma}{\eta_t}.$$

$\square$

**Lemma 20.** *If Assumption 1,2,3 and 5 hold, we have that*

$$\sum_{t=1}^{T}\sum_{i\in\mathcal{N}}\frac{1}{\eta_t}\mathbb{E}\left(\sum_{j=1}^{n}[\mathbf{x}_t^*]_j\log(\frac{[\hat{\mathbf{x}}_{t+1,i}]_j}{[\mathbf{y}_{t,i}]_j})\right) \leq \frac{N^2\log(\frac{n}{\gamma})}{\eta_{T+1}} + \sum_{t=1}^{T}\frac{N\log(\frac{n}{\gamma})}{\eta_{t+1}}\|\mathbf{x}_{t+1}^* - \mathbf{x}_t^*\|_1.$$

*Proof.*

$$\sum_{t=1}^{T}\sum_{i\in\mathcal{N}}\frac{1}{\eta_t}\mathbb{E}\left(\sum_{j=1}^{n}[\mathbf{x}_t^*]_j\log(\frac{[\hat{\mathbf{x}}_{t+1,i}]_j}{[\mathbf{y}_{t,i}]_j})\right)$$
$$= \sum_{t=1}^{T}\sum_{i\in\mathcal{N}}\frac{1}{\eta_t}\left(\mathbb{E}\Big(\sum_{j=1}^{n}[\mathbf{x}_t^*]_j\log(\frac{1}{[\mathbf{y}_{t,i}]_j})\Big) - \mathbb{E}\Big(\sum_{j=1}^{n}[\mathbf{x}_t^*]_j\log(\frac{1}{[\hat{\mathbf{x}}_{t+1,i}]_j})\Big)\right)$$
$$= \underbrace{\sum_{t=1}^{T}\sum_{i\in\mathcal{N}}\left(\frac{1}{\eta_t}\mathbb{E}\Big(\sum_{j=1}^{n}[\mathbf{x}_t^*]_j\log(\frac{1}{[\mathbf{y}_{t,i}]_j})\Big) - \frac{1}{\eta_{t+1}}\mathbb{E}\Big(\sum_{j=1}^{n}[\mathbf{x}_{t+1}^*]_j\log(\frac{1}{[\mathbf{y}_{t+1,i}]_j})\Big)\right)}_{①}$$
$$+ \underbrace{\sum_{t=1}^{T}\sum_{i\in\mathcal{N}}\frac{1}{\eta_{t+1}}\left(\mathbb{E}\Big(\sum_{j=1}^{n}[\mathbf{x}_{t+1}^*]_j\log(\frac{1}{[\mathbf{y}_{t+1,i}]_j})\Big) - \mathbb{E}\Big(\sum_{j=1}^{n}[\mathbf{x}_t^*]_j\log(\frac{1}{[\mathbf{y}_{t+1,i}]_j})\Big)\right)}_{②}$$
$$+ \underbrace{\sum_{t=1}^{T}\sum_{i\in\mathcal{N}}\frac{1}{\eta_{t+1}}\left(\mathbb{E}\Big(\sum_{j=1}^{n}[\mathbf{x}_t^*]_j\log(\frac{1}{[\mathbf{y}_{t+1,i}]_j})\Big) - \mathbb{E}\Big(\sum_{j=1}^{n}[\mathbf{x}_t^*]_j\log(\frac{1}{[\hat{\mathbf{x}}_{t+1,i}]_j})\Big)\right)}_{③}$$
$$+ \underbrace{\sum_{t=1}^{T}\sum_{i\in\mathcal{N}}\left(\frac{1}{\eta_{t+1}} - \frac{1}{\eta_t}\right)\mathbb{E}\Big(\sum_{j=1}^{n}[\mathbf{x}_t^*]_j\log(\frac{1}{[\hat{\mathbf{x}}_{t+1,i}]_j})\Big)}_{④}.$$

Firstly, from the convexity of function $\log(\frac{1}{x})$, we have

$$③ = \sum_{t=1}^{T} \sum_{i \in \mathcal{N}} \frac{1}{\eta_{t+1}} \left( \mathbb{E}\Big( \sum_{j=1}^{n} [\mathbf{x}_t^*]_j \log(\frac{1}{[\mathbf{y}_{t+1,i}]_j}) \Big) - \mathbb{E}\Big( \sum_{j=1}^{n} [\mathbf{x}_t^*]_j \log(\frac{1}{[\hat{\mathbf{x}}_{t+1,i}]_j}) \Big) \right)$$

$$= \sum_{t=1}^{T} \sum_{i \in \mathcal{N}} \frac{1}{\eta_{t+1}} \left( \mathbb{E}\Big( \sum_{j=1}^{n} [\mathbf{x}_t^*]_j \log(\frac{1}{\sum_{k \in \mathcal{N}_i \cup \{i\}} w_{ik}[\hat{\mathbf{x}}_{t+1,k}]_j}) \Big) - \mathbb{E}\Big( \sum_{j=1}^{n} [\mathbf{x}_t^*]_j \log(\frac{1}{[\hat{\mathbf{x}}_{t+1,i}]_j}) \Big) \right)$$

$$\leq \sum_{t=1}^{T} \sum_{i \in \mathcal{N}} \frac{1}{\eta_{t+1}} \left( \mathbb{E}\Big( \sum_{j=1}^{n} [\mathbf{x}_t^*]_j \sum_{k \in \mathcal{N}_i \cup \{i\}} w_{ik} \log(\frac{1}{[\hat{\mathbf{x}}_{t+1,k}]_j}) \Big) - \mathbb{E}\Big( \sum_{j=1}^{n} [\mathbf{x}_t^*]_j \log(\frac{1}{[\hat{\mathbf{x}}_{t+1,i}]_j}) \Big) \right)$$

$$= \sum_{t=1}^{T} \sum_{i \in \mathcal{N}} \frac{1}{\eta_{t+1}} \left( \Big( \sum_{k \in \mathcal{N}_i \cup \{i\}} w_{ki} \Big) \mathbb{E}\Big( \sum_{j=1}^{n} [\mathbf{x}_t^*]_j \log(\frac{1}{[\hat{\mathbf{x}}_{t+1,i}]_j}) \Big) - \mathbb{E}\Big( \sum_{j=1}^{n} [\mathbf{x}_t^*]_j \log(\frac{1}{[\hat{\mathbf{x}}_{t+1,i}]_j}) \Big) \right)$$

$$= 0.$$

Then, for ①, we can show that

$$① \leq \sum_{i \in \mathcal{N}} \frac{1}{\eta_1} \mathbb{E}\Big( \sum_{j=1}^{n} [\mathbf{x}_1^*]_j \log(\frac{1}{[\mathbf{y}_{1,i}]_j}) \Big) \leq \frac{N^2 \log(\frac{n}{\gamma})}{\eta_1},$$

where the final inequality follows from $[\mathbf{y}_{1,i}]_j \geq \frac{\gamma}{n}$ such that $\log(\frac{1}{[\mathbf{y}_{1,i}]_j}) \leq \log(\frac{n}{\gamma})$ and $\sum_{j=1}^{n} [\mathbf{x}_1^*]_j \leq N$.

As for ②, we can have that

$$\sum_{t=1}^{T} \sum_{i \in \mathcal{N}} \frac{1}{\eta_{t+1}} \left( \mathbb{E}\Big( \sum_{j=1}^{n} [\mathbf{x}_{t+1}^*]_j \log(\frac{1}{[\mathbf{y}_{t+1,i}]_j}) \Big) - \mathbb{E}\Big( \sum_{j=1}^{n} [\mathbf{x}_t^*]_j \log(\frac{1}{[\mathbf{y}_{t+1,i}]_j}) \Big) \right)$$

$$= \sum_{t=1}^{T} \sum_{i \in \mathcal{N}} \frac{1}{\eta_{t+1}} \left( \mathbb{E}\Big( \sum_{j=1}^{n} \Big( [\mathbf{x}_{t+1}^*]_j - [\mathbf{x}_t^*]_j \Big) \log(\frac{1}{[\mathbf{y}_{t+1,i}]_j}) \Big) \right)$$

$$\leq \sum_{t=1}^{T} \frac{N \log(\frac{n}{\gamma})}{\eta_{t+1}} \|\mathbf{x}_{t+1}^* - \mathbf{x}_t^*\|_1,$$

where the final inequality follows from $\log(\frac{1}{[\mathbf{y}_{t+1,i}]_j}) \leq \log(\frac{n}{\gamma})$. Moreover, we have ④$\leq$ $N^2 \log(\frac{n}{\gamma}) \Big( \frac{1}{\eta_{T+1}} - \frac{1}{\eta_1} \Big)$. We finally get

$$\sum_{t=1}^{T} \sum_{i \in \mathcal{N}} \frac{1}{\eta_t} \mathbb{E}\left( \sum_{j=1}^{n} [\mathbf{x}_t^*]_j \log(\frac{[\hat{\mathbf{x}}_{t+1,i}]_j}{[\mathbf{y}_{t,i}]_j}) \right) \leq \frac{N^2 \log(\frac{n}{\gamma})}{\eta_{T+1}} + \sum_{t=1}^{T} \frac{N \log(\frac{n}{\gamma})}{\eta_{t+1}} \|\mathbf{x}_{t+1}^* - \mathbf{x}_t^*\|_1.$$

$$\square$$

Next, from Line 13-18 in Algorithm 2, we know that, for any $i \in \mathcal{N}$, when $\sum_{a \in \mathcal{V}_i} \Big( [\mathbf{y}_{t,i}]_a \exp(\eta_t [\tilde{\nabla} F_t^s(\mathbf{x}_{t,i})]_a) \Big) \leq 1$, we can have $[\mathbf{x}_{t+1,i}]_a \geq [\mathbf{y}_{t,i}]_a$. As for $\sum_{a \in \mathcal{V}_i} \Big( [\mathbf{y}_{t,i}]_a \exp(\eta_t [\tilde{\nabla} F_t^s(\mathbf{x}_{t,i})]_a) \Big) > 1$, we have $\sum_{a \in \mathcal{V}_i} [\mathbf{x}_{t+1,i}]_a = 1$ such that $\sum_{j=1}^{n} \Big( [\mathbf{y}_{t,i}]_j - [\mathbf{x}_{t+1,i}]_j \Big) = \sum_{a \in \mathcal{V}_i} [\mathbf{y}_{t+1,i}]_a - 1 \leq 0$. As a result, we can conclude that

$$\sum_{t=1}^{T} \sum_{i \in \mathcal{N}} \frac{1}{\eta_t} \mathbb{E}\left( \sum_{j=1}^{n} \Big( [\mathbf{y}_{t,i}]_j - [\mathbf{x}_{t+1,i}]_j \Big) \right) \leq 0.$$

Finally, we get these result

$$\left(\frac{1-e^{-c}}{c}\right)\sum_{t=1}^{T}F_t(\mathbf{x}_t^*) - \sum_{t=1}^{T}\mathbb{E}\Big(F_t(\bar{\mathbf{x}}_t)\Big) \leq (3G^2 + LDG)\Big(\sum_{t=1}^{T}\sum_{\tau=1}^{t}N^{\frac{3}{2}}\beta^{t-\tau}(1-\gamma)^{t-\tau}\eta_\tau\Big)$$
$$+ G\gamma D + \frac{N^2\log(\frac{n}{\gamma})}{\eta_{T+1}} + \sum_{t=1}^{T}\frac{N\log(\frac{n}{\gamma})}{\eta_{t+1}}\|\mathbf{x}_{t+1}^* - \mathbf{x}_t^*\|_1 + \sum_{t=1}^{T}\frac{2N^2\gamma}{\eta_t}\cdot + \frac{NG}{2}\sum_{t=1}^{T}\eta_t.$$

Like Lemma 12, we also can verify that

**Lemma 21.** *In Algorithm 2, if Assumption 1,2,3,5 hold and each set function $f_t$ is monotone submodular with curvature c for any $t \in [T]$, then*

$$\left(\frac{1-e^{-c}}{c}\right)\sum_{t=1}^{T}F_t(\mathbf{x}_t^*) - \sum_{t=1}^{T}\mathbb{E}\Big(F_t\big(\sum_{i\in\mathcal{N}}\mathbf{x}_{t,i}\odot\mathbf{1}_{\mathcal{V}_i}\big)\Big) \leq (4G^2 + LDG)\Big(\sum_{t=1}^{T}\sum_{\tau=1}^{t}N^{\frac{3}{2}}\beta^{t-\tau}(1-\gamma)^{t-\tau}\eta_\tau\Big) + G\gamma D$$
$$+ \frac{N^2\log(\frac{n}{\gamma})}{\eta_{T+1}} + \sum_{t=1}^{T}\frac{N\log(\frac{n}{\gamma})}{\eta_{t+1}}\|\mathbf{x}_{t+1}^* - \mathbf{x}_t^*\|_1 + \sum_{t=1}^{T}\frac{2N^2\gamma}{\eta_t} + \frac{NG}{2}\sum_{t=1}^{T}\eta_t.$$

From Calinescu et al. (2011); Chekuri et al. (2014), we know that the optimal value of continuous problem Eq.(3) is equal to the optimal value of the corresponding discrete submodular maximization Eq.(1), so we can set $\mathbf{x}_t^* := \mathbf{1}_{\mathcal{A}_t^*}$ where $\mathcal{A}_t^*$ is the maximizer of Eq.(1).

Next, like Lemma 13, we also can show a relationship between $\mathbb{E}(f_t(\cup_{i\in\mathcal{N}}\{a_{t,i}\}))$ and $\mathbb{E}\big(F_t\big(\sum_{i\in\mathcal{N}}\mathbf{x}_{t,i}\odot\mathbf{1}_{\mathcal{V}_i}\big)\big)$.

**Lemma 22.** *If the function $f_t$ is monotone submodular and $a_{t,i}$ is the action taken via the agent $i \in \mathcal{N}$ at time t, then we have*

$$\mathbb{E}\Big(f_t\big(\cup_{i\in\mathcal{N}}\{a_{t,i}\}\big)\Big) \geq \mathbb{E}\Big(\big(F_t\big(\sum_{i\in\mathcal{N}}\mathbf{x}_{t,i}\odot\mathbf{1}_{\mathcal{V}_i}\big)\big)\Big).$$

Finally, we get

$$\left(\frac{1-e^{-c}}{c}\right)\sum_{t=1}^{T}f_t(\mathcal{A}_t^*) - \sum_{t=1}^{T}\mathbb{E}\Big(F_t\big(\cup_{i\in\mathcal{N}}\{a_{t,i}\}\big)\Big) \leq (4G^2 + LDG)\Big(\sum_{t=1}^{T}\sum_{\tau=1}^{t}N^{\frac{3}{2}}\beta^{t-\tau}(1-\gamma)^{t-\tau}\eta_\tau\Big) + G\gamma D$$
$$+ \frac{N^2\log(\frac{n}{\gamma})}{\eta_{T+1}} + \sum_{t=1}^{T}\frac{N\log(\frac{n}{\gamma})}{\eta_{t+1}}\|\mathbf{x}_{t+1}^* - \mathbf{x}_t^*\|_1 + \sum_{t=1}^{T}\frac{2N^2\gamma}{\eta_t} + \frac{NG}{2}\sum_{t=1}^{T}\eta_t.$$

