# OpenReview forum: "Near-Optimal Online Learning for Multi-Agent Submodular Coordination: Tight Approximation and Communication Efficiency"
_ICLR.cc/2025/Conference — ICLR 2025 Spotlight_

### Official Review · Reviewer_e6bZ · 2024-10-23

**Soundness:** 3
**Presentation:** 2
**Contribution:** 3
**Rating:** 6
**Confidence:** 3

**Summary:**

This paper presents two algorithms for solving the multi-agent online submodular maximization problem, accompanied with a convergence analysis for each. Both algorithms achieve near-optimal approximation ratio while requiring less restrictive assumptions about the graph structure (connected instead of complete). Notably, the second algorithm further gets rid of the projection step by using a specific Bergman divergence (KL divergence) within the mirror descent framework. The effectiveness of the proposed algorithms is also validated through a simulated multi-target tracking task.

**Strengths:**

1. The paper's contributions and the structure are clearly delivered, with additional remarks are provided to enhance understanding. The appendix is well-organized and easy to read.
2. The contributions are interesting and compelling. They advance prior work to a near-optimal approximation ration and substantially relaxing the assumptions regarding graph connectivity.

**Weaknesses:**

1. Overall, the writing could be improved. This paper is heavy on notations & definitions, and the authors should be more careful when introducing definitions/abbreviations/notations. Specific examples are listed below:

- Line 16: What do OSG, MA-OSMA, MA-OSEA stand for? The contribution "...skilfully (typo) harnessing the KL divergence..." is too technical and detailed for an abstract. A more high-level intuition is expected at this stage.

- Line 49: The abbreviation MA-OSM is introduced twice.

- Line 138: Typo "lowercase."

- "Notations" paragraph: On Line 142, avoid starting a sentence with math notations. The lowercase "s" is used without an explicit definition.

- Line 154: "MA-OSM" is introduced again.

- Line 225: Consider using a different notation than "F" for clarity.

- Theorem 2: Is the phrase "for any x,y" missing?

- Algorithm 1: Using $\eta_t$ suggests an adaptive step size, but it is set to a constant in Remark 8.

- Assumption 5: Is $\mathbb{E}[\tilde{\nabla}F_t^s(x)\mid x]=\nabla F_t^s(x)$ derived from the estimation procedure, or is it an assumption? Constant $G$ is not defined.

- Theorem 5: notation $\gamma$ should be redefined, as its first definition occurs early in the paper and the other one in Algorithm 2 appears only afterwards, which is shown later in the paper. Consider using a different set of parameters than $C_1,C_2,C_3$, which are already used in Theorem 3.

- Line 992: typo "y"->"x"

- Line 1058: "argmin"?

- Line 1100: is $=[y_{t,i}]_{\mathcal{V}\textbackslash \mathcal{V}_i}$ a typo?

- Line 1126: Typo - $x$ should be $y_{t,i}$ in the first term on RHS.

- Line 1029+1030: Typo $F^A$ should be $F^s$.

- Line 1183: Can you provide more specific references to the results in Nedic&Ozdaglar, 2009 and Horn&Johnson, 2012?

- Line 1291: Typo - $F^A$ should be $F^s$.

- Line 1374: Typo - "the first equation of Lemma 12."

- Line 1420: Typo - $f$ missing.

2. Algorithm 2: It would be beneficial to provide more insights into the difference between Algorithm 1&2.

3. The related work section is not informative enough to provide a high-level overview of earlier work, provide more detailed explanation and comparison of the works listed in Table 1 more in detail would be helpful.

4. Line 83: Please provide citations for these "well-established consensus techniques."

5. For the simulations, error bars are missing from the plot and 5 iterations seems insufficient for drawing robust conclusions.

**Questions:**

1. The comparison of prior work can be better clarified. For now, the contribution part in the introduction doesn't provide a clear picture of how the improvements are made and why they are novel.
2. The paper would benefit from more detailed insights into improvements and associated trade-offs? E.g. regarding the connectivity of the graph, while the prior work assume a complete graph, but they only require a directed acyclic graph (as indicated in Figure 1). In contrast, the completeness is no longer needed but the edges need to be undirected. Why is this trade-off beneficial? Another similar question comes from the regret bound: it is degrading from $\tilde{\mathcal{O}}(\sqrt{C_T*T})$ to $\mathcal{O}(\sqrt{C_T * T/(1-\beta)})$. Although the simulations suggest improved performance, a theoretical explanation of why this compromise in the regret bound is reasonable would be valuable. Could you provide more insights?
3. Could you elaborate on how to sample from different actions under the constraint "$\sum_{a\in\mathcal{V}_i}x_a\leq 1$" without requiring normalization?
4. Can you clarify the novelty of Theorem 2 compared to Corollary 7 in Zhang et al. 2024 in Remark 3? The results look very similar, apart from the difference in curvature.
5. Can you explain more on the difference between Distributed-CG and this paper? The two seem closely related but Distributed-CG is only mentioned in the appendix.
6. Line 1402: Can you provide a more detailed explanation of why the first inequality follows from the monotonicity of $f_t$? It is applied several time but the connection between the monotonicity of $F$ and monotonicity of $f$ is not clear
7. Does this paper offer any novel insights into optimization, beyond those already established in the literature?

---

> ### Author Response · Authors · 2024-11-21
>
> Thank you for your detailed and insightful comments. We are truly grateful for the time and effort you
> have dedicated to reviewing our manuscript. Your feedback is invaluable to us. Below, we will address the concerns you have raised in the sections of ``Questions`` and ``Weaknesses``.
>
> ---------------------
>
> ``Questions``:
>
> -------------------
>
> > **1.The comparison of prior work can be better clarified. For now, the contribution part in the introduction doesn't provide a clear picture of how the improvements are made and why they are novel.**
>
> Thank you very much for your insightful comments.
>
> Firstly, due to the limited space, we have chosen to place the main comparison of prior work in Appendix A.1. However, in Table 1, we have provided a detailed comparison with the most relevant OSG algorithms.
>
> Secondly, before summarizing our three contributions in our paper, we have detailed the main techniques and novelties employed in our algorithms, which constitute the cornerstone of  our improvements. Specifically:
>
> a) "We utilize the multi-linear extension to transform the discrete submodular maximization problem into a continuous optimization problem. This approach allows us to alleviate the stringent requirement for a complete communication graph by leveraging well-established consensus techniques in decentralized optimization."
>
> b) "We develop a surrogate function for the multilinear extension of submodular functions with curvature $c$. This innovation empowers us to surpass the sub-optimal $(\frac{1}{1+c})$-approximation stationary points."
>
> c) "We implement a distinct strategy for updating the selected probabilities associated with each agent's actions and those of others. This strategy requires agents to only assess the marginal gains of actions within their own action sets, thereby reducing the practical requirement on the observational scope of each agent."
>
> We hope these points will help you understand both the novel aspects and the improvements in our work.
>
> ---------------------------
>
> > **2.The paper would benefit from more detailed insights into improvements and associated trade-offs? e.g. regarding the connectivity of the graph, while the prior work assume a complete graph, but they only require a directed acyclic graph (as indicated in Figure 1). In contrast, the completeness is no longer needed but the edges need to be undirected. Why is this trade-off beneficial? Another similar question comes from the regret bound: it is degrading from  $\widetilde{O}(\sqrt{C_{T}T})$ to $O(\frac{\sqrt{C_{T}T}}{1-\beta})$. Although the simulations suggest improved performance, a theoretical explanation of why this compromise in the regret bound is reasonable would be valuable. Could you provide more insights?**
>
> Firstly, due to space limitations, our paper assumes a static undirected communication network among agents to simplify the description of the communication graph. However, the capabilities of the algorithmic framework we employ are far from it. We can efficiently extend our algorithms to time-varying and directed network topologies using consensus techniques from (Nedic & Olshevsky,
> 2014; Nedic et al., 2017), as indicated in our "Conclusions and Future Work" section. ***Therefore, this is not a trade-off but an enhancement.*** From the perspective of communication topology, our Algorithms 1 and 2 exhibit greater flexibility and efficiency compared to OSG.
>
> Secondly, to a certain extent, we can view our proposed Algorithms 1 and 2, as well as OSG, as the different trade-off paths between approximation ratio and regret. Specifically,
>
> a) Our Algorithms 1 and 2 slightly sacrifice the regret bound to ensure a tight approximation ratio and reduce the communication burden;
>
> b) On the contrary, OSG ensures a consistent $\widetilde{O}(\sqrt{C_{T}T})$ regret bound, but it will incur a significant communication overhead and a sub-optimal approximation guarantee.
>
> Finally, ***we  wish to emphasize that our chosen path significantly outperforms OSG.*** It is important to note that the theoretical results of OSG algorithm indicate that its regret bound will exhibit a deviation of $O(\log(T))$ compared to  $O(\sqrt{C_{T}T}))$.  Although the spectral information indeed affects the regret bound of our MA-OSMA, ensuring $\frac{1}{1-\beta}<C\log(T)$ for some constant $C$ is sufficient to guarantee that our MA-OSMA achieves superior performance in both regret bound and approximation ratio compared to the OSG algorithm. When the time-horizon $T$ is very large, a relatively sparse graph even can secure $\frac{1}{1-\beta}<<C\log(T)$ for some constant $C$.
>
> -------------------------------

---

> > ### Author Response · Authors · 2024-11-21
> >
> > >**3. Could you elaborate on how to sample from different actions under the constraint $\sum_{a\in V_i}x_a\le1$ without requiring normalization?**
> >
> > Thank you very much. This is an excellent question.
> >
> > When $\sum_{a\in V_i}x_a<1$, there is a probability of $\big(1-\sum_{a\in V_i}x_a\big)$ that we might sample an empty policy ($\emptyset$). Typically, when encountering an empty policy, we will encourage the agent to explore based on a certain random principle, such as randomly selecting an action from $V_i$. It is important to note that our submodular function is monotone, so this exploration strategy will not yield a worse outcome than an empty policy.
> >
> > -------------------------
> >
> > >**4.Can you clarify the novelty of Theorem 2 compared to Corollary 7 in Zhang et al. 2024 in Remark 3? The results look very similar, apart from the difference in curvature.**
> >
> > Thank you very much for your question.
> >
> > Please note that in Corollary 7 of f (Zhang et al., 2024), they only consider a weak diminishing return property of the set function and do not take into account the curvature information. The main difference between our paper and Corollary 7 is that we incorporate the curvature $c$.
> >
> > ***Although the final results may appear similar, we would like to emphasize that incorporating curvature information into the surrogate function associated with the multilinear extension is far from a trivial task.*** This process requires the development and  establishment of some new inequalities. A pivotal intermediate step in the derivation of the surrogate function involves a deep understanding of  the relationship between $\langle\mathbf{y},\nabla F(z*\mathbf{x})\rangle$, $F(\mathbf{y})$ and $F(z*\mathbf{x})$ for any $\mathbf{x},\mathbf{y}\in[0,1]^{n}$ and $z\in[0,1]$ where $F$  represents the multi-linear extension of some submodular function. In the previous articles (Zhang et al., 2022; 2024), the authors employed the following inequalities:
> >
> >  $ \langle\mathbf{y},\nabla F(z*\mathbf{x})\rangle$
> >
> > $=\langle(z*\mathbf{x})\lor\mathbf{y}-z*\mathbf{x},\nabla F(z*\mathbf{x})\rangle+\langle(z*\mathbf{x})\land\mathbf{y},\nabla F(z*\mathbf{x})\rangle$
> >
> > $\ge F((z*\mathbf{x})\lor\mathbf{y})-F(z*\mathbf{x})+\langle(z*\mathbf{x})\land\mathbf{y},\nabla F(z*\mathbf{x})\rangle$
> >
> > $=F(\mathbf{y})-F(z*\mathbf{x})+F((z*\mathbf{x})\lor\mathbf{y})-F(\mathbf{y})+\langle(z*\mathbf{x})\land\mathbf{y},\nabla F(z*\mathbf{x})\rangle$
> >
> > $\ge F(\mathbf{y})-F(z*\mathbf{x})$,
> >
> > where the first equality follows from $(z*\mathbf{x})\lor\mathbf{y}+(z*\mathbf{x})\land\mathbf{y}=(z*\mathbf{x})+\mathbf{y}$, the second inequality comes from the concave property of multi-linear extension of submodular function and the final inequality follows from the monotonicity of $F$, i.e., $F((z*\mathbf{x})\lor\mathbf{y})\ge F(\mathbf{y})$ and $\nabla F(z*\mathbf{x})\ge\textbf{0}$.
> >
> > Instead of directly discarding these two non-negative terms, namely, $F((z*\mathbf{x})\lor\mathbf{y})-F(\mathbf{y})$ and $\langle(z*\mathbf{x})\land\mathbf{y},\nabla F(z*\mathbf{x})\rangle$, in the previous inequality, we utilize the concept of curvature to develop the following two new inequalities
> > \begin{equation*}
> >     F\big((z*\mathbf{x})\lor\mathbf{y}\big)-F(\mathbf{y})\ge(1-c)\Big(F(z*\mathbf{x})-F\big((z*\mathbf{x})\land\mathbf{y}\big)\Big)
> > \end{equation*} in Lemma 3 of Appendix B and
> > \begin{equation*}
> >     \langle(z*\mathbf{x})\land\mathbf{y},\nabla F(z*\mathbf{x})\rangle\ge(1-c)F\big((z*\mathbf{x})\land\mathbf{y}\big)
> > \end{equation*} in Lines 1016-1020 of Lemma 4 of Appendix C, where $c$ is the curvature of submodular function. As a result, we can get the following new inequality:
> >
> > $\langle\mathbf{y},\nabla F(z*\mathbf{x})\rangle$
> >
> > $\ge F(\mathbf{y})-F(z*\mathbf{x})+F((z*\mathbf{x})\lor\mathbf{y})-F(\mathbf{y})+\langle(z*\mathbf{x})\land\mathbf{y},\nabla F(z*\mathbf{x})\rangle$
> >
> >  $\ge F(\mathbf{y})-F(z*\mathbf{x})+(1-c)\Big(F(z*\mathbf{x})-F\big((z*\mathbf{x})\land\mathbf{y}\big)\Big)+(1-c)F\big((z*\mathbf{x})\land\mathbf{y}\big)$
> >
> >  $=F(\mathbf{y})-cF(z*\mathbf{x}).$
> >
> > Inserting this inequality into the construction of surrogate function, we finally achieve the result of Theorem 2.
> >
> > -------------------------

---

> > > ### Author Response · Authors · 2024-11-21
> > >
> > > >**5. Can you explain more on the difference between Distributed-CG and this paper? The two seem closely related but Distributed-CG is only mentioned in the appendix.**
> > >
> > > Thank you very much for your question.
> > >
> > > Firstly, we would like to emphasize that Distributed-CG (Rezazadeh & Kia, 2023) and the algorithms presented in this paper (Algorithm 1 and Algorithm 2) employ completely different technical frameworks. Distributed-CG primarily utilizes the "Continuous Greedy" method introduced in (Calinescu et al., 2011) to optimize the multi-linear extension for the offline multi-agent submodular maximization(MA-SM) problem. This "Continuous Greedy" is a variant of the Frank-Wolfe algorithm. In contrast, our Algorithms 1 and 2 take a different path by designing new surrogate functions for the multi-linear extension and then performing gradient ascent on these surrogate functions. It's important to note that gradient ascent and Frank-Wolfe are two distinct optimization frameworks.
> > >
> > > Secondly, compared to the "surrogate-gradient-ascent" approach used in this paper, the "Continuous Greedy" or Distributed-CG algorithm has several limitations:
> > >
> > > a) ***Continuous Greedy method is not robust to stochastic noises.***  (Hassani et al., 2017)  provides a counterexample where replacing the exact gradient of the multi-linear extension in "Continuous Greedy" with a stochastic gradient will result in arbitrarily poor approximation guarantees. Moreover, from the definition of the multi-linear extension, its exact gradient typically requires exponential computational costs, making ``Continuous Greedy" impractical. ***To adapt "Continuous Greedy" for stochastic settings, we would have to employ batch stochastic gradient and variance reduction techniques, which significantly increase the number of queries to submodular functions.*** As shown in our Table 2, Distributed-CG requires $O(\frac{n\log(\frac{1}{\epsilon})}{\epsilon^{3}})$ value queries to achieve $(\frac{1-e^{-c}}{c})OPT-\epsilon$, whereas our Algorithms 1 and 2 only require only $O(\frac{n}{\epsilon^{2}})$ and $O(\frac{n\log(\frac{1}{\epsilon})}{\epsilon^{2}})$, respectively.
> > >
> > > b) ***Extending Distributed-CG to online settings will incur significant communication overhead.*** Unlike gradient ascent, which can be naturally extended to online settings, adapting Distributed-CG to an online scenario commonly requires a special meta-framework (Zhu et al., 2021). This meta-framework requires that agents must communicate with each other $\sqrt{T}$ times at each time spot. Note that when $T$ is very large, the requirement for substantial communication within a short time step is impractical. In contrast, our Algorithms 1 and 2 only require agents to communicate once per time step. This is also why we chose to design new surrogate functions and a mirror ascent method in this paper.
> > >
> > > In summary, Distributed-CG and the algorithms presented in this paper follow two distinct paths. Moreover, extending this algorithm to online settings would incur substantial communication costs and a greater number of value queries. Therefore, we initially decided not to utilize its framework to improve the OSG algorithm. Instead, we chose to design new surrogate functions and a novel mirror ascent method.
> > >
> > > --------------------------
> > >
> > > >**6. Line 1402: Can you provide a more detailed explanation of why the first inequality follows from the monotonicity of $f_{t}$? It is applied several time but the connection between the monotonicity of $F$ and monotonicity of $f$ is not clear.**
> > >
> > > Thank you very much for your question.
> > >
> > > Please note that prior to Line 1402, specifically in Appendix B, Lemma 1 (Lines 946-958), we have demonstrated that when a set function is monotone, its multi-linear extension is also monotone. For instance, from the third point of Lemma 1, we can know that:
> > > \begin{equation*}
> > >     \frac{\partial F(\mathbf{x})}{\partial x_{i}}=E_{\mathcal{R}\sim\mathbf{x}}\Big(f(\mathcal{R}\cup\{i\})-f(\mathcal{R}\setminus\{i\})\Big).
> > > \end{equation*}
> > > Given the monotonicity of $f$, we have$f(\mathcal{R}\cup\{i\})\ge f(\mathcal{R}\setminus\{i\})$, which implies that$\frac{\partial F(\mathbf{x})}{\partial x_{i}}\ge 0$ such that $F$ is monotone.
> > >
> > > ------------------

---

> ### Author Response · Authors · 2024-11-21
>
> >**7. Does this paper offer any novel insights into optimization, beyond those already established in the literature?**
>
> Thank you very much for your question.
>
> From our perspective, we think this paper may provide the following novel insights into the optimization community:
>
> a) The gradient of objective functions may not always serve as  a good direction for many non-convex optimizations. This point is evident from our Theorem 2, where we design an auxiliary function for the multi-linear extension of submodular functions. Specifically, we have theoretically demonstrated that by ascending along a new surrogate gradient, we may reach better stationary points.
>
> b) Consensus techniques in decentralized optimization can effectively integrate information and achieve the coordination among multi-agent scenarios. For instance, in this paper, although each agent can only access the marginal evaluations of actions within their own action set, our theoretical proofs for Algorithms 1 and 2 show that with the help of consensus techniques, we can achieve superior performance. It is important to clarify the distinction between the decentralized problem and the multi-agent scenario emphasized in this paper. Typically, in decentralized optimization, each local node is assumed to maintain its own local utility function, with the collective goal being to optimize the sum of these local functions. In contrast, within the scope of this paper, we assume that all agents share a common submodular function, but each agent is restricted to accessing a unique set of actions.
>
> ----------------------
> `` Weaknesses``:
>
> ---------------
> >**1. Overall, the writing could be improved. This paper is heavy on notations & definitions, and the authors should be more careful when introducing definitions/abbreviations/notations......**
>
>
> Thank you very much for your valuable feedback.  We will carefully review and correct the issues related to notations, definitions, and abbreviations as you have pointed out.
>
> -----------------------
> >**2. Algorithm 2: It would be beneficial to provide more insights into the difference between Algorithm 1&2?**
>
> Thank you for your suggestion.
>
> As mentioned in the text, the key difference between Algorithm 1 and Algorithm 2 lies in the incorporation of a ``mixing a uniform distribution" technique in Algorithm 2. Due to space limitations, we were unable to elaborate on this, but we are happy to provide further insights as follows.
>
> The main challenge with KL-divergence is that its gradient tends towards infinity at the boundary, which contradicts the Lipschitz condition stated in Assumption 4. To address this issue, we introduce a small perturbation in our decision vector to prevent it from reaching the boundary. Specifically, in Algorithm 2, at line 7, we incorporate a uniform distribution by adding $\frac{1}{n}\mathbf{1}$ to our decision vector. This ensures that our final decision keeps away from the boundary. Notably, $\frac{1}{n}\mathbf{1}$ can be interpreted as a uniform distribution across all $n$ actions.
>
> We hope this explanation clarifies the unique aspects of Algorithm 2 and its difference from Algorithm 1.
>
> ---------------------------
>
> >**3. The related work section is not informative enough to provide a high-level overview of earlier work, provide more detailed explanation and comparison of the works listed in Table 1 more in detail would be helpful.**
>
> Thank you for your suggestion. We appreciate your feedback on the related work section.
>
> For a detailed explanation and comparison of the previous works, please refer to our previous responses to the Question 1 and Question 5, where we have provided a thorough analysis and comparison of the works listed in Table 1 and Table 2.
>
> -----------------------
>
> >**4. Line 83: Please provide citations for these "well-established consensus techniques".**
>
> We apologize for not providing citations for the "well-established consensus techniques" at their first mention on Line 83. Although these relevant citations were included later in the text, we fully understand the importance of immediate attribution. We will add the appropriate citations at Line 83 in the revised version.
>
> ----------------------------
>
> >**5. For the simulations, error bars are missing from the plot and 5 iterations seems insufficient for drawing robust conclusions.**
>
>  Thank you for your valuable feedback on our simulation results. The primary theoretical contribution of our paper is that Algorithms 1 and 2, in expectation, can provide a better approximation guarantee than OSG. Therefore, our experiments only demonstrate the empirical performance in terms of the average case. The robustness of our multi-agent algorithms is indeed a topic worth exploring, such as regret bounds and approximation ratios under high probability. We will consider this as a future direction.
>
> -----------------------

---

> > ### Author Response · Authors · 2024-11-23
> >
> > Thank you very much for your suggestions. Regarding weakness 1, we would like to highlight some changes we have made in the revised paper.
> >
> > * I have removed the repeated introduction of the abbreviation MA-OSM in Line 49 and Line 155.
> > * We have changed "lower" to "lowercase" in Line 138.
> > * We have removed "skilfully harnessing the KL divergence" in Line 23 and Line 100, changing it to "which effectively utilizes".
> > * We have corrected the 's' to $|S|$ in Line 144.
> > * In Definition 2 of Line 225, we have replaced "F" with "G" to avoid confusion.
> > * We have added explanations for $\mathbf{x},\mathbf{y}$ in Theorem 2.
> > * Please note that Assumption 5 is merely a convenience for us to state Theorems 3 and 5. We have explained the values of the parameter $G$ under different norms in Appendix A.2.
> > * Please note that in Algorithm 2, $\gamma$ is a predefined parameter.
> > * In Line 992, we have changed $\mathbf{y}$ to $\mathbf{x}$.
> > * In Line 1126, we have replaced all instances of $\mathbf{x}$.
> > * In Lines 1130, 1131, and 1291, we have changed $F^{A}$ to $F^{s}$.
> > * Specific references to the results in Nedic&Ozdaglar, 2009 have been provided in Line 1184.
> > * From Line 1372 to 1381, we have added a more detailed explanation of the derivation of Lemma 12.
> > * In Line 1420, we have included the missing $f_t$.

---

> > > ### Comment · Reviewer_e6bZ · 2024-11-23
> > >
> > > Thanks for addressing my comments and providing detailed responses.  I am more convinced of this work's contribution and decide to increase my score accordingly.

---

### Official Review · Reviewer_qEPy · 2024-10-25

**Soundness:** 3
**Presentation:** 3
**Contribution:** 3
**Rating:** 6
**Confidence:** 2

**Summary:**

This paper proposes an algorithm for online multi-agent submodular optimization problem, which is then extended to a projection-free variant. The main contribution is that the proposed algorithm relaxes the complete graph assumption and achieves tighter approximation ratio, as compared to the SOTA algorithm.

**Strengths:**

The paper is overall well-written. The proposed algorithm removes the limitation of complete graph assumption as compared to the SOTA algorithm. It also achieves a tighter approximation ratio as compared to the SOTA algorithm.

**Weaknesses:**

1. The execution of the algorithm relies on the curvature parameter $c$. However, this parameter may be unknown beforehand.
2. In line 11 of the Alg. 1 and line 12 of the Alg. 2, the objective function needs to be evaluated for 2$\sum_i|\mathcal{V}_i|$ times. This can be very inefficient when the function $f_t$ is expensive to evaluate or the number of actions is large.
3. The experimental evaluation part is relatively weaker than the theoretical contribution. In the experiments, the paper did not verify the theoretical result, that is, some figure showing the ratio between empirical objective value and the optimal value.

**Questions:**

1. Can the authors comment on how to select $c$ in practice?
2. The algorithm introduces a set of hyperparameters such as step size $\eta_t$, mixing parameter, and the weight matrix. Can the authors comment on how to select these hyperparameters?
3. In figure 3, I can understand the utility column. But can the authors elaborate on the other two columns and why are they also interesting?

---

> ### Author Response · Authors · 2024-11-21
>
> Thank you for your detailed and insightful comments. We are truly grateful for the time and effort you
> have dedicated to reviewing our manuscript. Your feedback is invaluable to us. Below, we will address the concerns you have raised in the sections of ``Questions`` and ``Weaknesses``.
>
> ----------------
> ``Questions``:
>
> ------------
> >**1.Can the authors comment on how to select $c$ in practice?**
>
> This is a good question. we have also highlighted this point as one of the directions for future direction in our "Conclusions and Future Work" section.
>
> In practice, it is impossible to know the exact value of the curvature $c$. However, we know that $c$ lies within the interval $[0,1]$. An important property is that if a function has curvature $c$, it also has curvature $c'$ for any $c' \geq c$. Therefore, in our experiments, we set $c=1$ to ensure that our algorithm can achieve at least a $(1 - e^{-1})$-approximation guarantee.
>
> Beyond this, we have a rough idea regarding how to select the curvature $c$.  Since we know that curvature $c$ is within the interval $[0,1]$, we can divide this interval using a set of values ${\epsilon, 2\epsilon, \dots, 1}$, given an $\epsilon$ where we assume $\frac{1}{\epsilon}$ is an integer. We then maintain $\frac{1}{\epsilon}$ different versions of the MA-OSMA algorithm, each using one of these values as their curvature $c$. For example, we could denote them as MA-OSMA($\epsilon$), MA-OSMA($2\epsilon$), \dots, MA-OSMA($1$). We can treat these $\frac{1}{\epsilon}$ different versions of the MA-OSMA algorithm as a bandit problem with $\frac{1}{\epsilon}$ arms, where our goal is to find the optimal arm. Generally speaking, when $\epsilon$ is sufficiently small, we can guarantee a $\left(\frac{(1 - e^{-c})}{c} - \epsilon\right)$-approximation ratio. However, this approach requires running $\frac{1}{\epsilon}$ different versions of the MA-OSMA algorithm, which may impose a significant computational burden.
>
> ------------------
>
> >**2.The algorithm introduces a set of hyperparameters such as step size $\eta_{t}$, mixing parameter, and the weight matrix. Can the authors comment on how to select these hyperparameters?**
>
> Thanks for your question regarding the selection of hyperparameters in our algorithm. We have detailed these settings in Appendix A.3 of our manuscript.
>
> Specifically, for the weight matrix $\mathbf{W}$ in the context of a random graph, we assign values as follows: if the edge $(i,j)$ exists in the graph, we set $w_{ij} = \frac{1}{1 + \max(d_i, d_j)}$, where $d_i$ and $d_j$ represent the degrees of agents $i$ and $j$, respectively. If $(i,j)$ is not an edge and $i \neq j$, then $w_{i,j} = 0$. Lastly, we define $w_{i,i} = 1 - \sum_{j \in \mathcal{N}i} w_{i,j}$. For a complete graph, we set $w_{ij} = \frac{1}{N}$, where $N$ is the total number of agents. Besides, we have chosen $c = 1$, $\eta_t = \frac{1}{\sqrt{T}}$ and the mixing parameter $\gamma=\frac{1}{T^{2}}$ in the MA-OSMA and MA-OSEA algorithms.
>
> --------------------
>
> >**3. In figure 3, I can understand the utility column. But can the authors elaborate on the other two columns and why are they also interesting?**
>
> Thank you for your insightful question.
>
> In our experiments, we have adopted the commonly used "inverse of distance" as our primary metric for utility. In addition to this, we also consider the number of targets in proximity to agents and the average distance between agents and targets as valuable indicators for evaluating the effectiveness of tracking algorithms.
>
> To this end, we offer a comparison of our Algorithm 1, Algorithm 2, and the OSG algorithm based on these two indicators in the final two columns in Figure 3. The data clearly demonstrate that both of our algorithms outperform the OSG algorithm in terms of all these metrics. This comparison highlights the effectiveness of our approach in not only maximizing utility but also in optimizing the spatial distribution and proximity to targets.
>
> ----------
> ``Weaknesses``:
>
> ---------------
>
> >**1. The execution of the algorithm relies on the curvature parameter $c$. However, this parameter may be unknown beforehand.**
>
> Please refer to our response to the first question earlier.
>
> ------------------

---

> > ### Author Response · Authors · 2024-11-21
> >
> > >**2. In line 11 of the Alg. 1 and line 12 of the Alg. 2, the objective function needs to be evaluated for $2\sum_{i}|\mathcal{V}_{i}|$ times. This can be very inefficient when the function $f_t$ is expensive to evaluate or the number of actions is large.**
> >
> > Thank you for your feedback. This is a good question.
> >
> > To our knowledge, in recent years, the field of continuous optimization has developed many efficient methods for dealing with functions that are expensive to evaluate, such as zero-order optimization. Similarly, we also can apply these techniques to the algorithms in this paper. Next, we will present a straightforward  technique to reduce the complexity of function evaluations.
> >
> >
> > Please note that in our Algorithm 1 and Algorithm 2, we only require an unbiased estimate for the surrogate gradient. Therefore, a simple way to reduce the number of evaluations of the objective function is to have each agent $i$ uniformly select one action $a\in V_{i}$ and then use $\frac{1-e^{-c}}{c}|V_{i}|(f_t(\mathcal{R}\cup${$a$})$-f_t(\mathcal{R}\setminus${$a$})$)e_a$  to estimate $\nabla F_{t}^{s}$ where $e_a$ is the $a$-th basis vector in $\mathcal{R}^{|V_{i}|}$. In this method, each agent only needs to assess the function twice at every iteration.
> >
> > --------------
> >
> > >**3. The experimental evaluation part is relatively weaker than the theoretical contribution. In the experiments, the paper did not verify the theoretical result, that is, some figure showing the ratio between empirical objective value and the optimal value.**
> >
> > Thank you very much for your question.
> >
> > Firstly, we would like to emphasize that finding the optimal solution for submodular maximization problems is NP-hard (Bian et al., 2020). Secondly, in our experiments, we simulated a scenario with 20 agents, each having 24 candidate actions. To search for the optimal decision, we would need to enumerate $24^{20}$ different combinations of actions, which is computationally infeasible. Therefore, it is very hard to show the ratio between the empirical objective value and the optimal value.
> >
> > --------

---

> ### Comment · Reviewer_qEPy · 2024-11-25
> **Thanks for the response**
>
> I would like to thank the authors for the detailed responses. They address most of my concerns. And I would like to maintain my rating.

---

### Official Review · Reviewer_NhDa · 2024-11-03

**Soundness:** 4
**Presentation:** 4
**Contribution:** 4
**Rating:** 8
**Confidence:** 3

**Summary:**

This paper studies a multi-agent online submodular maximization problem (MA-OSM). In each step, each agent independently selects a decision from a unique decision set, queries local marginal gains of an underlying submodular function, and exchanges information with neighboring agents. By leveraging a novel surrogate function approach for the multi-linear extension of submodular functions, the paper proposes an online algorithm that achieves a tight approximation ratio relative to the offline solution under a connected communication graph, with a regret bound of $O\left(\sqrt{\frac{C_T T}{1 - \beta}}\right)$, where $C_T$ is the deviation of the maximizer sequence and $\beta$ is the spectral gap of the network. The paper further develops a projection-free variant of the first algorithm, and attains the same approximation ratio with a slight loss in the regret guarantee. Numerical results effectively illustrate the performance of the proposed algorithms compared to prior work, which has sub-optimal approximation ratios.

**Strengths:**

-  The proposed algorithms improve the approximation ratio of prior work from $\frac{1}{1+c}$ to the tight ratio $\frac{1-e^{-c}}{c}$. Additionally, this improvement is achieved with limited feedback requirements, rather than needing a complete communication graph.

- The use of surrogate functions to overcome limitations related to stationary points in the original multi-linear extension is both effective and inspiring.

- The paper is well presented, with a coherent flow, clear contributions, and strong results.

**Weaknesses:**

- There is some ambiguity regarding the communication complexity of the proposed algorithms. Although they attain a tight approximation ratio, it is unclear if the communication complexity is also tight. Based on the title, the paper seems to claim tight communication efficiency; however, this aspect is not clearly addressed. Additionally, the paper appears to assume a static communication graph, which may not apply to applications like multi-target tracking as discussed in the numerical section.

- In the proposed algorithm, each agent is required to query the local marginal gain oracle multiple times. While this is common in prior work, it is not evident if/how such an oracle would exist in practical applications.

**Questions:**

- Could you provide formal comments on the tightness of the communication complexity for both proposed algorithms?
- Since the proposed algorithms can be viewed as an online implementation of offline approximation algorithms, it would be beneficial to provide a formal comparison with the work of [Razazadeh and Kia 2023] in the main paper.
- The regret guarantee depends on $C_T$, the deviation of the maximizer sequence. Please specify how $C_T$ varies and impacts the regret guarantee.
- what does the subscript $d$ denote in the regret definition?

---

> ### Author Response · Authors · 2024-11-21
>
> Thank you for your detailed and insightful reviews. We sincerely appreciate the time and effort you
> have dedicated to reviewing our manuscript. Your feedback is invaluable to us. In the following, we will address the concerns you have raised in the sections of ``Questions`` and ``Weaknesses``.
>
> ------------
>  ``Questions``:
>
> ----------------
> > *  **Could you provide formal comments on the tightness of the communication complexity for both proposed algorithms?**
>
> We sincerely apologize for the confusion caused by our subtitle. We would like to clarify that, in this paper, we only claim that our $\frac{(1-e^{-c})}{c}$-approximation ratio is tight in a certain sense, as evidenced by Theorem 4.1. in (Vondrak, 2010). With respect to the communication graph, our main contribution is to relax the rigid requirement of a complete graph in the previous OSG algorithm, without asserting that our communication complexity is optimal.
>
> Designing an optimal communication graph for Multi-Agent Online Submodular Maximization(MA-OSM) Problem is indeed a very interesting problem. There have been numerous studies exploring "how to design the graph topology while maintain an optimal convergence rate", as seen in (Chow et al., 2016; Song et al., 2022; Vogels et al., 2023). Applying these techniques to our Algorithms 1 and 2 is certainly a promising direction.
>
> -----------------------
> >* **Since the proposed algorithms can be viewed as an online implementation of offline approximation algorithms, it would be beneficial to provide a formal comparison with the work of [Razazadeh and Kia 2023] in the main paper.**
>
> Thank you very much for your suggestion. We had initially considered providing a thorough comparison of our Algorithms 1 and 2 against the work of [Razazadeh and Kia 2023] in the offline scenario within the main body of the paper. However, due to space limitations, we had to include this comparison in Appendix A.1 and Table 2.
>
> In general, in the offline setting, compared to [Razazadeh and Kia 2023], our Algorithms 1 and 2 can achieve a $\frac{1-e^{-c}}{c}$-approximation with fewer queries to the submodular function. Specifically, [Razazadeh and Kia 2023] can achieve $(\frac{1-e^{-c}}{c})OPT-\epsilon$ at the cost of $O(\frac{n\log(\frac{1}{\epsilon})}{\epsilon^{3}})$ value queries to the submodular function. In contrast, our Algorithm 1 and Algorithm 2 only require inquiring the submodular objective $O(\frac{n}{\epsilon^{2}})$ and $O(\frac{n\log(\frac{1}{\epsilon})}{\epsilon^{2}})$ times to attain $(\frac{1-e^{-c}}{c})OPT-\epsilon$, if we view any incoming objective function $f_{t}$ as some fixed submodular set function $f$ and then return the set $\{\cup_{i\in\mathcal{N}}{a_{t,i}}\},\forall t\in[T]$ with probability $\frac{1}{T}$.
>
> -------------
>
> >* **The regret guarantee depends on  $C_{T}$, the deviation of the maximizer sequence. Please specify how $C_{T}$ varies and impacts the regret guarantee.**
>
> Thank you very much for your question. In this paper, we have not made any assumptions regarding the deviation of the maximizer sequence $C_{T}$, which means
> $C_{T}$ may evolve in an unpredictable direction.  For instance, if $C_{T}=O(T)$,  neither our Algorithms 1 and 2 nor the OSG can achieve superior regret performance. This to some extent reflects the well-known ``\emph{no free lunch}" theorem. Theoretically, as highlighted in (Besbes et al., 2015), if $C_{T}$ is linear in $T$, then no admissible policy can attain sub-linear regret.
>
> --------------
>
> >* **what does the subscript $d$ denote in the regret definition?**
>
> Thanks for your question. The subscript $d$ in the regret definition represents the dynamic regret. This $d$ is mainly used to distinguish from another common measure used to assess the performance of online algorithms, namely, static regret. We show the definition of static regret as follows:
>
> $\textbf{Reg}^{s}_{\alpha}(T)=\alpha\max_{\mathcal{A}}\Big(\sum_{t=1}^{T}f_{t}(\mathcal{A})\Big)-\sum_{t=1}^{T}f_{t}(\cup_{i\in\N}\{a_{t,i}\})$.
>
> Note that in static regret, we use the best static strategy as our benchmark. In contrast, in dynamic regret, we adopt a dynamically optimal strategy, that is,
>
> $\textbf{Reg}^{d}_{\alpha}(T)=\alpha\sum_{t=1}^{T}f_{t}(\mathcal{A}_{t}^{*})-\sum_{t=1}^{T}f_{t}(\cup_{i\in\N}\{a_{t,i}\})$.
>
> ------------------------------------------

---

> > ### Author Response · Authors · 2024-11-21
> >
> > ------------
> > ``Weaknesses``:
> >
> > -------------
> > >* **There is some ambiguity regarding the communication complexity of the proposed algorithms. Although they attain a tight approximation ratio, it is unclear if the communication complexity is also tight. Based on the title, the paper seems to claim tight communication efficiency; however, this aspect is not clearly addressed. Additionally, the paper appears to assume a static communication graph, which may not apply to applications like multi-target tracking as discussed in the numerical section.**
> >
> > Thank you very much for your question.
> >
> > Firstly, we would like to clarify that our paper does not claim that our communication complexity is tight. Our primary contribution is to relax the rigid requirement of a complete graph in the previous OSG algorithms. For details regarding the design of the communication graph, please refer to our response to the first question earlier.
> >
> > Additionally, to simplify the description of the algorithms and the communication graph, this paper assumes a static communication network. However, the capabilities of our algorithmic framework are far beyond this assumption. We can efficiently extend our algorithms to handle time-varying and directed network topologies by employing consensus techniques as referenced in (Nedic & Olshevsky,2014; Nedic et al., 2017). Note that we highlighted this point in our "Conclusions and Future Work" section.
> >
> > -----------------
> >
> > >* **In the proposed algorithm, each agent is required to query the local marginal gain oracle multiple times. While this is common in prior work, it is not evident if/how such an oracle would exist in practical applications.**
> >
> > This is a good question.
> >
> > Firstly, we admit that the assumption each agent can obtain the marginal evaluations of actions within their own action set is a simplification of real-world scenarios. This allows us to focus on designing efficient algorithms for multi-agent submodular coordination.
> >
> > However, in practice, in addition to the multi-agent decision-making process considered in this paper, we also need to develop a learning algorithm for agents to utilize the information they acquire to fit the current submodular objective, as shown in (Corah & Michael, 2021). Then, each agent employs their fitted submodular function to adjust their actions for subsequent decisions. So, typically, there do exist such an (approximate) marginal oracle. Therefore, one of our future directions is to incorporate the estimation errors from the learning algorithm into our Algorithm 1 and Algorithm 2, as outlined in  "Conclusions and Future Work" section.
> >
> > --------------------

---

> > > ### Comment · Reviewer_NhDa · 2024-11-24
> > >
> > > Thanks for your response and clarification. I will retain my score.

---

### Official Review · Reviewer_WfEW · 2024-11-03

**Soundness:** 3
**Presentation:** 2
**Contribution:** 3
**Rating:** 8
**Confidence:** 4

**Summary:**

This work proposes two algorithms MA-OSMA and MA-OSEA that collaboratively solve an online, multi-agent submodular optimization problem. Both algorithms produce $\left(\frac{1-e^{-c}}{c} \right)$ sub-optimal solutions in contrast to $\frac{1}{1+c}$ approximate solutions produced by state of the art methods. They also achieve a regret bound of $\tilde O\left( \sqrt\frac{C_TT}{1-\beta} \right)$. The MA-OSMA algorithm requires a costly projection step, this is then removed by using a KL divergence and "mixing uniform distribution" argument in MA-OESA. The method is then illustrated via numerical simulations.

**Strengths:**

The paper addresses an interesting and relevant problem - motivated by multi-agent robotic problems. The main contributions of the paper are:
- Improving on the state of the art performance bounds of the OSG algorithm by approving the approximation quality to $\frac{1-e^{-c}}{c}$ where $c$ encodes the curvature of the problem.
- The second algorithm, MA-OESA, removes the need for a projection step which allegedly is computationally expensive.

**Weaknesses:**

I believe there are several areas where this paper can be improved - mostly in the area of presentation and clarity of exposition.

- In section 2.1 the authors state that "each agent $i$ within $\mathcal N$ is equipped with a unique and discrete set of actions". But in the numerical section this seems to be contradicted as the simulation considers the case where each agent has exactly the same choices: up, down, left, right, diagonal etc. Are they unique or identical?!

- In the formulation (1), what is the set $\mathcal A$? Is it just a subset of $\mathcal V$?

- In terms of motivation, it is not clear why the algorithm from Vondrak or Bian et al, cannot be used to achieve the $\frac{1-e^{-c}}{c}$ bound.

- It's a bit weird to claim that your own algorithm "skillfully harnesses" something. I suggest removing this from the abstract and body.

- The explanation of how to circumvent the fact that KL-divergence is not Lipschitz is not clear at all. The text is full of jargon. What does "mixing a uniform distribution" mean? Neither Theorem 5 nor remark 9 seem to have any parameters from uniform distribution. Please provide some intuition about what is going on here.


- The OSG method should be described and compared to MA-OSMA and MA-OESA in a more direct way. What is the main difference between the algorithms, how do the necessary assumptions differ etc. Why is it that it only achieves $\frac{1}{1+c}$-accurate solutions? This will strengthen the numerical section where it is shown in simulations that the proposed methods out-perform it.

**Questions:**

- The paper lacks any discussion of how the performance of the two algorithms differ beyond what is in Table 1. How costly is the projection step? How do the two methods differ beyond a factor of $\log T$ regret?
- Does the spectral information show up anywhere in the regret bound of the OSG algorithm? Likewise, do any of the connection graph properties appear in the derivation of the approximation ratio in MA-OSMA or MA-OSEA. It seems like they need to appear somewhere and here they appear in the regret bound - was this an active choice? It seems like there is trade-off between approximation ratio and Regret.

---

> ### Author Response · Authors · 2024-11-21
>
> Thank you for your detailed and insightful reviews. We sincerely appreciate the time and effort you have dedicated to reviewing our manuscript. Your feedback is invaluable to us. Below, we will respond to the concerns you have raised in ``Questions`` and ``Weaknesses``.
>
> --------------
> ``Questions``:
>
> --------------
> >*  **The paper lacks any discussion of how the performance of the two algorithms differ beyond what is in Table 1. How costly is the projection step? How do the two methods differ beyond a factor of
> $\log(T)$ regret?**
>
> Firstly, in order to provide a clearer understanding of the computational cost associated with the general mirror projection in Algorithm 1, we will begin by introducing the standard methodological framework used to address a single constrained optimization problem.
>
> 1) We will consider the Lagrangian multiplier method, namely,
>
> $L(\mathbf{b},\lambda)=(-\langle \Big[\tilde{\nabla}F_{t}^{s}(\mathbf{x}_{(t,i)})\Big]_V,\mathbf{b}\rangle+(1/\eta_t)\mathcal{D}_g (\mathbf{b}, \mathbf{y}_t)+\lambda(\sum b_k-1)$;
>
> （***Please note that due to LaTeX symbol issues in OpenReview, some subscripts may not be displayed.***）
>
> 2) Then, fixing $\lambda\ge0$, we aim to find the optimal $\mathbf{b}^*(\lambda)=\arg\min L(\mathbf{b},\lambda)$;
>
> 3) Next, by substituting the optimal $\mathbf{b}^*(\lambda)$, we derive a one-dimensional function with respect to $\lambda$, i.e., $g(\lambda)=L(\mathbf{b}^*(\lambda),\lambda)$;
>
> 4) Finally, we find $\lambda^*=\arg\max g(\lambda)$ and obtain the optimal projection vector $\mathbf{b}^*=\mathbf{b}^*(\lambda^*)$.
>
> From the algorithmic framework we have described,  it is clear that the primary computational expense in the general mirror projection lies in the search for the optimal dual variable $\lambda^*=\arg\max g(\lambda)$. Even in the case of the Euclidean distance, the process of finding the optimal dual variable requires multiple iterations of binary search, as demonstrated in Appendix B in (Zhou et al., 2022). In contrast, when considering KL divergence, we can explicitly determine the optimal dual variable, as detailed in our Appendix E.
>
> Secondly, to mitigate the issue that the gradient of the KL divergence tends to infinity when approaching the boundaries,  we have incorporated a shift of $\frac{\gamma}{n}\mathbf{1}_{n}$ in Step 7 of Algorithm 2. This modification results in a regret deviation of $\log(T)$ for Algorithm 2.
>
> ------------
> >*  **Does the spectral information show up anywhere in the regret bound of the OSG algorithm? Likewise, do any of the connection graph properties appear in the derivation of the approximation ratio in MA-OSMA or MA-OSEA. It seems like they need to appear somewhere and here they appear in the regret bound - was this an active choice? It seems like there is trade-off between approximation ratio and Regret.**
>
> Thank you very much for your question.
>
> Firstly, as indicated in (Grimsman et al., 2018), the approximation ratio of OSG will deteriorate in proportion to the size of the largest independent group of agents within the communication graph, as detailed in our Table 1. Generally speaking, the number of nodes in the largest independent set is highly correlated with the spectral information. Thus, based on the current known results, we can only conclude that spectral information influences the approximation ratio of OSG. Similarly, our analysis of algorithms MA-OSMA and MA-OSEA only reveals that spectral information impacts their regret bounds.
>
> Secondly, to a certain extent, we can view our proposed Algorithms 1 and 2, as well as OSG, as the different trade-off paths between approximation ratio and regret. Specifically,
>
> a) Our Algorithms 1 and 2 slightly sacrifice the regret bound to ensure a tight approximation ratio and reduce the communication burden;
>
> b) On the contrary, OSG ensures a consistent $\widetilde{O}(\sqrt{C_{T}T})$ regret bound, but it will incur a significant communication overhead and a sub-optimal approximation guarantee.
>
> Finally, we  wish to emphasize that our chosen path significantly outperforms OSG. It is important to note that the theoretical results of OSG algorithm indicate that its regret bound will exhibit a deviation of $O(\log(T))$ compared to  $O(\sqrt{C_{T}T}))$.  Although the spectral information indeed affects the regret bound of our MA-OSMA, ensuring $\frac{1}{1-\beta}<C\log(T)$ for some constant $C$ is sufficient to guarantee that our MA-OSMA achieves superior performance in both regret bound and approximation ratio compared to the OSG algorithm. When the time-horizon $T$ is very large, a relatively sparse graph even can secure $\frac{1}{1-\beta}<<C\log(T)$ for some constant $C$.
>
> ---------------------------

---

> > ### Author Response · Authors · 2024-11-21
> >
> > -----------
> > ``Weaknesses``:
> >
> > --------------
> > >* **In section 2.1 the authors state that "each agent $i$ within $\mathcal{N}$ is equipped with a unique and discrete set of actions". But in the numerical section this seems to be contradicted as the simulation considers the case where each agent has exactly the same choices: up, down, left, right, diagonal etc. Are they unique or identical?**
> >
> > Thanks for your question.  It certainly highlights an aspect of our paper  that deserves further clarification.
> >
> > In the context of multi-target tracking tasks, each agent indeed shares the same movement space. ***However, it is crucial to emphasize that the "movement space" is conceptually distinct from the "action space" we consider.*** Generally speaking, the objective function of the multi-target tracking task is not solely dependent on the consistent  "movement space".  The key factor in shaping the objective function is  ***which agent carries out which particular movement***. To reflect the different role of each agent within the tracking system, we  commonly assign unique identity tags to each movement.
> >
> > To illustrate, in our experiments, we consider the movement decision space M={($\theta,s$): $s\in${5,10,15}units/s, $\theta\in${$\frac{\pi}{4},\frac{\pi}{2},\frac{3\pi}{4},\pi,\dots,2\pi$} } where $\theta$ and $s$ represent the movement angle and the speed, respectively. To identify the executor of the movement, we will label the movement space $M$ with identity tags. Consequently, for each agent $i\in${$1,2,\dots,20$}, its action space $V_{i}$ is constructed as follows: $V_{i}=M\times${$i$}$=${$(\theta,s,i)$: $s\in${5,10,15}units/s, $\theta\in${$\frac{\pi}{4},\frac{\pi}{2},\frac{3\pi}{4},\pi,\dots,2\pi$} }, where the action $(\theta,s,i)$ indicates that agent $i$ moves towards the direction of $\theta$ at a speed of $s$.
> >
> > With identification information included, these action sets are mutually disjoint, namely, $V_{i}\cap V_{j}$ for any $i\neq j\in${$1,2,\dots,20$}. Please note that this approach of distinguishing identities through tagging each decision is a widely adopted practice in multi-agent system, as exemplified in revenue maximization over multiple campaigns (Du et al., 2017; Aslay et al., 2017; Han et al., 2021), multi-round product marketing (Sun et al., 2018) and online multi-Ad slot allocation (Streeter et al.,2009).
> >
> > --------------
> > >* **In the formulation (1), what is the set $\mathcal{A}$? Is it just a subset of $\mathcal{V}$?**
> >
> > Yes, $\mathcal{A}$ is a subset of $\mathcal{V}$ and satisfies $|\mathcal{A} \cap \mathcal{V}_{i}| \leq 1$ for any $i\in\mathcal{N}$.
> >
> > -------------
> >
> > >* **In terms of motivation, it is not clear why the algorithm from Vondrak or Bian et al, cannot be used to achieve the $\frac{(1-e^{-c})}{c}$ bound.**
> >
> > Thank you very much for your question.
> >
> > Firstly, please note that t (Bian et al., 2017) investigates the performance of the greedy method under the simple cardinality constraint. If extending this method to our multi-agent online submodular maximization (MA-OSM) problem, it ultimately leads to the online sequential greedy (OSG) algorithm (Xu et al., 2023). Therefore, throughout the method in t (Bian et al., 2017), we cannot achieve the $\frac{(1-e^{-c})}{c}$ bound for MA-OSM.
> >
> > Secondly, note that  (Vondrak, 2010) employs the ***Continuous Greedy*** method (Calinescu et al.,
> > 2011) to optimize the multi-linear extension of a submodular function with curvature. However, this method requires the computation of the exact gradient of the multi-linear extension, which generally entails exponential computational costs. Thus, this approach is impractical. Moreover, the ***Continuous Greedy*** method is not robust to stochastic noises. (Hassani et al., 2017) provides a counterexample where replacing the exact gradient of the multi-linear extension in ***Continuous Greedy*** with a stochastic gradient leads to arbitrarily poor approximation guarantees. Therefore, directly applying the ***Continuous Greedy*** method from  (Vondrak, 2010)  to  MA-OSM problem, we also cannot achieve the $\frac{(1-e^{-c})}{c}$ bound.
> >
> > ---------
> >
> > >* **It's a bit weird to claim that your own algorithm "skillfully harnesses" something. I suggest removing this from the abstract and body.**
> >
> > Thank you very much for your suggestion. We agree that it would be more appropriate to remove the phrase"skillfully harnesses" from both the abstract and the body of the paper. We plan to revise the description to "a projection-free MA-OSEA algorithm utilizing the KL divergence by mixing a uniform distribution".
> >
> > -------------

---

> ### Author Response · Authors · 2024-11-21
>
> >* **The explanation of how to circumvent the fact that KL-divergence is not Lipschitz is not clear at all. The text is full of jargon. What does "mixing a uniform distribution" mean? Neither Theorem 5 nor remark 9 seem to have any parameters from uniform distribution. Please provide some intuition about what is going on here.**
>
> Thank you very much for your feedback. We would like to clarify the meaning of `mixing a uniform distribution’.
>
> The primary issue with KL-divergence is that as it approaches zero, its gradient tends towards infinity, which violates the Lipschitz condition. To circumvent this, we introduce a small perturbation in our decision vector to keep it away from the boundary. Specifically, in Algorithm 2, line 7, we mix a uniform distribution, namely, adding $\frac{1}{n}\textbf{1}$ to our decision vector. This ensures that our final decision stays away from the boundary. Note that $\frac{1}{n}\textbf{1}$ can be interpreted as a uniform distribution over all $n$ actions. The term ·mixing a uniform distribution' is commonly used in some bandit papers, which is why we adopted this phrasing. We apologize for any confusion this term may have caused.
>
> ---------------
>
> >*  **The OSG method should be described and compared to MA-OSMA and MA-OESA in a more direct way. What is the main difference between the algorithms, how do the necessary assumptions differ etc. Why is it that it only achieves $\frac{1}{1+c}$-accurate solutions? This will strengthen the numerical section where it is shown in simulations that the proposed methods out-perform it.**
>
> Thank you very much for your suggestion.
>
> Due to space limitations in the main paper, we were unable to provide a more detailed description of the OSG algorithm. However, we can offer a brief overview of its core idea here. The OSG algorithm is fundamentally based on the standard greedy method. It operates by allowing one agent to select the best action from its action set and then communicates this choice to the other agents. Subsequently, the second agent selects the best action from its own action set based on the current knowledge and shares its choice with the remaining agents. This greedy iteration continues in a sequential manner.
>
> Since the greedy method only can achieve a sub-optimal $\frac{1}{1+c}$-approximation for general submodular maximization problems (Fisher et al., 1978), the OSG algorithm also suffers from this limitation. This is the primary reason why OSG only achieves $\frac{1}{1+c}$-approximation solutions.
>
> ---------------------

---

> > ### Comment · Reviewer_WfEW · 2024-11-26
> > **Response**
> >
> > I thank the authors for their clarifications. I will updated my score.

---

### Official Review · Reviewer_H5Ct · 2024-11-04

**Soundness:** 3
**Presentation:** 3
**Contribution:** 3
**Rating:** 6
**Confidence:** 3

**Summary:**

The author(s) propose a continuous surrogate approximation of multi-agent submodular maximization problems and solve via mirror ascent approach to (i) achieve tight approximation, and (ii) reduce communication density among agents. The proposed approach is evaluated in multi-robot target tracking examples.

**Strengths:**

1. The contribution is novel and sound, and the theoretical results are strong.
2. The paper is well presented in general. The authors clearly motivated challenges existing in the literature and this work’s objectives.
3. The efficacy of the proposed approach is validated using sufficient amount of empirical evidence.

**Weaknesses:**

1. Problem formulation
- I think the author(s) should list the assumptions made in this section explicitly, such as those in lines 177 and 159.
- I think this section is a bit unconnected with the motivating example. I would appreciate a bit more introduction about why and how problems like multi-agent tracking can be formulated as online submodular maximization problems. For example, I am a bit confused here why the objective function is revealed after agents make decisions and why the objective function is guaranteed to be submodular?
2. Experiments:
- Is all the targets being strictly homogeneous in type a requirement from the algorithm? What if different targets are of different types?
3. Contribution: this work builds upon a few existing methods in the literature. The novelty of the approaches can be better specified; cf. Question 3.
4. While submodular optimization has a variety of applications, the problem studied in this work seems to be more specific to robotics applications. I think is relatively less relevant to this venue.
5. Other minor points:
- Line 70: “Furthermore” – hyperlink is redundant
- Figure 3: It can be shown in the caption or in the figure that 3 rows represent different target types. Right now, it’s a bit hidden in the text.

**Questions:**

1. Line 157: The assumption that different agents must have mutually disjoint action sets looks odd to me. Taking multi-target tracking task as an example, wouldn’t different agents totally have access to execute the same action?
2. Multi-linear extension: I would appreciate that if the author(s) could speak more about how good the continuous relaxation is compared to the original submodular maximization problem and under which conditions. Right now, the presentation just takes this relaxation for granted.
3. In Algorithm 1, what enables the relaxed requirement of a sparse communication graph stated in the contribution statement? Is this part of the author(s) contribution or something enabled by the fact of using existing online mirror ascent methods? A related point is that the author(s) claim that the novelty of their proposed non-oblivious surrogate function for the multi-linear extension objective is the curvature information. Is accounting for the curvature information in non-oblivious surrogate approximation of multi-linear extension a well-recognized challenge in the literature?
4. The requirement that the objective function needs to be monotone submodular seems to be strong. Can the author(s) elaborate on how restrictive this is?

---

> ### Author Response · Authors · 2024-11-21
>
> Thank you very much for your careful reading and constructive feedback. We are grateful for the time and effort you have dedicated to reviewing our manuscript. In what follows, we will address some of your concerns in ``Questions`` and ``Weaknesses``.
>
> ---------------
> ``Questions``:
>
> ---------------
> > **1. Line 157: The assumption that different agents must have mutually disjoint action sets looks odd to me. Taking multi-target tracking task as an example, wouldn’t different agents totally have access to execute the same action?**
>
> Thanks for your question.  It certainly highlights an aspect of our paper that deserves further clarification.
>
> In the context of multi-target tracking tasks, each agent indeed shares the same movement space. ***However, it is crucial to emphasize that the "movement space" is conceptually distinct from the "action space" we consider.*** Generally speaking, the objective function of the multi-target tracking task is not solely dependent on the consistent  "movement space".  The key factor in shaping the objective function is  ***which agent carries out which particular movement***. To reflect the different role of each agent within the tracking system, we  commonly assign unique identity tags to each movement.
>
> To illustrate, in our experiments, we consider the movement decision space M={($\theta,s$): $s\in${5,10,15}units/s, $\theta\in${$\frac{\pi}{4},\frac{\pi}{2},\frac{3\pi}{4},\pi,\dots,2\pi$} } where $\theta$ and $s$ represent the movement angle and the speed, respectively. To identify the executor of the movement, we will label the movement space $M$ with identity tags. Consequently, for each agent $i\in${$1,2,\dots,20$}, its action space $V_{i}$ is constructed as follows: $V_{i}=M\times${$i$}$=${$(\theta,s,i)$: $s\in${5,10,15}units/s, $\theta\in${$\frac{\pi}{4},\frac{\pi}{2},\frac{3\pi}{4},\pi,\dots,2\pi$} }, where the action $(\theta,s,i)$ indicates that agent $i$ moves towards the direction of $\theta$ at a speed of $s$.
>
> With identification information included, these action sets are mutually disjoint, namely, $V_{i}\cap V_{j}$ for any $i\neq j\in${$1,2,\dots,20$}. Please note that this approach of distinguishing identities through tagging each decision is a widely adopted practice in multi-agent system, as exemplified in revenue maximization over multiple campaigns (Du et al., 2017; Aslay et al., 2017; Han et al., 2021), multi-round product marketing (Sun et al., 2018) and online multi-Ad slot allocation (Streeter et al.,2009).
>
> ---------------
> > **2.Multi-linear extension: I would appreciate that if the author(s) could speak more about how good the continuous relaxation is compared to the original submodular maximization problem and under which conditions. Right now, the presentation just takes this relaxation for granted.**
>
> Thank you for your question. Next, we will briefly discuss the advantages of the multi-linear extension over the original submodular maximization problem from two aspects. Before we proceed, it is essential to emphasize that the superiority of multi-linear extension is widely recognized within the submodular optimization community.
>
> a) ***Beyond Cardinality Constraint***: When considering the simple cardinality constraint, t, (Nemhauser et al., 1978) demonstrated that the greedy algorithm can provide a tight $(1-1/e)$-approximation to the submodular maximization problem. However, when confronted with more complex matroid constraints, as shown in (Fisher et al., 1978), the greedy method only can guarantee a sub-optimal $1/2$-approximation to the submodular maximization problem. For a long period, how to achieve the tight $(1-1/e)$-approximation for the submodular maximization problem over matroid constraints remains an open question. It was not until  the seminal works of (Calinescu et al., 2011; Chekuri et al., 2014) introduced the multi-linear extension and a series of rounding methods to improve the greedy method. Therefore, a notable benefit of the multi-linear extension is its capacity to enhance the approximation ratio of the greedy method over complex combinatorial constraints, which is the primary motivation behind our adoption of the multi-linear extension in this paper.
>
>
> b) ***Adapting to complicated optimization scenarios***: In recent years, the field of continuous optimization has seen the emergence of various efficient algorithmic frameworks. Consequently, another advantage of the multi-linear extension is its ability to leverage gradient information, potentially making it compatible with numerous scenarios of continuous optimization ,  , such as distributed
> settings (Mokhtari et al., 2018; Zhu et al., 2021; Zhang et al., 2023), online learning (Chen et al.,
> 2018b;a; Zhang et al., 2022) and minimax optimization (Adibi et al., 2022; Zhang et al., 2024). In contrast, the tools available for discrete optimization are not only more limited but also often cumbersome.
>
> ---------------

---

> > ### Author Response · Authors · 2024-11-21
> >
> > > **3. In Algorithm 1, what enables the relaxed requirement of a sparse communication graph stated in the contribution statement? Is this part of the author(s) contribution or something enabled by the fact of using existing online mirror ascent methods? A related point is that the author(s) claim that the novelty of their proposed non-oblivious surrogate function for the multi-linear extension objective is the curvature information. Is accounting for the curvature information in non-oblivious surrogate approximation of multi-linear extension a well-recognized challenge in the literature?**
> >
> > Thank you for your insightful questions.
> >
> > Firstly, at the Line 8 of Algorithm 1, we employ the well-established consensus techniques (Nedic &Ozdaglar, 2009; Nedic & Olshevsky, 2014) from the field of decentralized optimization to reduce the requirement of a complete communication graph. We highlight this integration of consensus techniques in Lines 81-82 and Lines 94-95 of our paper. Although our Algorithm 1 can indeed be considered a multi-agent variant of online mirror ascent methods, the significant reduction in the need for a complete communication graph is primarily attributed to the sophisticated integration of consensus techniques that we have seamlessly merged into our approach.
> >
> > Secondly, it is crucial to stress that incorporating curvature information into the surrogate function associated with the multi-linear extension is  far from a trivial task. This process requires the development and  establishment of some new inequalities. A pivotal intermediate step in the derivation of the surrogate function involves a deep understanding of  the relationship between $\langle\mathbf{y},\nabla F(z*\mathbf{x})\rangle$, $F(\mathbf{y})$ and $F(z*\mathbf{x})$ for any $\mathbf{x},\mathbf{y}\in[0,1]^{n}$ and $z\in[0,1]$ where $F$  represents the multi-linear extension of some submodular function. In the previous articles (Zhang et al., 2022; 2024), the authors employed the following inequalities:
> >
> >  $ \langle\mathbf{y},\nabla F(z*\mathbf{x})\rangle$
> >
> > $=\langle(z*\mathbf{x})\lor\mathbf{y}-z*\mathbf{x},\nabla F(z*\mathbf{x})\rangle+\langle(z*\mathbf{x})\land\mathbf{y},\nabla F(z*\mathbf{x})\rangle$
> >
> > $\ge F((z*\mathbf{x})\lor\mathbf{y})-F(z*\mathbf{x})+\langle(z*\mathbf{x})\land\mathbf{y},\nabla F(z*\mathbf{x})\rangle$
> >
> > $=F(\mathbf{y})-F(z*\mathbf{x})+F((z*\mathbf{x})\lor\mathbf{y})-F(\mathbf{y})+\langle(z*\mathbf{x})\land\mathbf{y},\nabla F(z*\mathbf{x})\rangle$
> >
> > $\ge F(\mathbf{y})-F(z*\mathbf{x})$,
> >
> > where the first equality follows from $(z*\mathbf{x})\lor\mathbf{y}+(z*\mathbf{x})\land\mathbf{y}=(z*\mathbf{x})+\mathbf{y}$, the second inequality comes from the concave property of multi-linear extension of submodular function and the final inequality follows from the monotonicity of $F$, i.e., $F((z*\mathbf{x})\lor\mathbf{y})\ge F(\mathbf{y})$ and $\nabla F(z*\mathbf{x})\ge\textbf{0}$.
> >
> > Instead of directly discarding these two non-negative terms, namely, $F((z*\mathbf{x})\lor\mathbf{y})-F(\mathbf{y})$ and $\langle(z*\mathbf{x})\land\mathbf{y},\nabla F(z*\mathbf{x})\rangle$, in the previous inequality, we utilize the concept of curvature to develop the following two new inequalities
> > \begin{equation*}
> >     F\big((z*\mathbf{x})\lor\mathbf{y}\big)-F(\mathbf{y})\ge(1-c)\Big(F(z*\mathbf{x})-F\big((z*\mathbf{x})\land\mathbf{y}\big)\Big)
> > \end{equation*} in Lemma 3 of Appendix B and
> > \begin{equation*}
> >     \langle(z*\mathbf{x})\land\mathbf{y},\nabla F(z*\mathbf{x})\rangle\ge(1-c)F\big((z*\mathbf{x})\land\mathbf{y}\big)
> > \end{equation*} in Lines 1016-1020 of Lemma 4 of Appendix C, where $c$ is the curvature of submodular function. As a result, we can get the following new inequality:
> >
> > $\langle\mathbf{y},\nabla F(z*\mathbf{x})\rangle$
> >
> > $\ge F(\mathbf{y})-F(z*\mathbf{x})+F((z*\mathbf{x})\lor\mathbf{y})-F(\mathbf{y})+\langle(z*\mathbf{x})\land\mathbf{y},\nabla F(z*\mathbf{x})\rangle$
> >
> >  $\ge F(\mathbf{y})-F(z*\mathbf{x})+(1-c)\Big(F(z*\mathbf{x})-F\big((z*\mathbf{x})\land\mathbf{y}\big)\Big)+(1-c)F\big((z*\mathbf{x})\land\mathbf{y}\big)$
> >
> >  $=F(\mathbf{y})-cF(z*\mathbf{x}).$
> >
> > Inserting this inequality into the construction of surrogate function, we finally achieve the result of Theorem 2.
> >
> > ------------------

---

> ### Author Response · Authors · 2024-11-21
>
> > **4. The requirement that the objective function needs to be monotone submodular seems to be strong. Can the author(s) elaborate on how restrictive this is?**
>
> First, let's revisit the definition of a submodular function. A set function $f:2^{V}\rightarrow\mathcal{R}_{+}$ is submodular if and only if for any $S\subseteq T\subseteq V$ and $e\in V\setminus T$, the following inequality holds:
>
> $f(S\cup\{e\})-f(S)\ge f(T\cup\{e\})-f(T)$.
>
> This definition means that the incremental benefit derived from adding an element to a set will diminish as the size of set grows larger, which is a mathematical expression of the well-known "diminishing-returns" principle in economics. As a result, it is evident that submodular objectives are extensively pervasive in contemporary economic activities. In addition, maximization of submodular functions has found numerous applications in machine learning, including data summarization (Lin & Bilmes, 2010; 2011; Wei et al., 2013; 2015), product recommendation (Kempe et al., 2003; El-Arini et al., 2009; Mirzasoleiman et al., 2016a) and in-context learning (Kumari et al., 2024). ***Broadly speaking, we assume here that the objective function is monotone submodular, which offers extensive applicability and encompasses a broad range of applications in machine learning.***
>
> Second, ***across a variety of scenarios of multi-robot coordination, the assumption of a monotone submodular objective function is not only standard but also has been rigorously validated by a multitude of theoretical frameworks*** (Zhou et al., 2018; Qu et al., 2019; Corah & Michael, 2021; Schlotfeldt et al., 2021; Krause et al., 2008). Generally speaking, the goal of multi-agent coordination is to better monitor and track other objects.  From an information-theoretic perspective, it's all about gathering as much information as we can about other entities. In the context of information theory, entropy is a scientific concept that measures the expected amount of information needed to describe the state of the variable. Therefore, many tasks of multi-robot coordination (Corah & Michael, 2021;Zhou et al., 2018) embrace entropy as a metric to assess the information acquired and subsequently aim to (approximately) maximize it. It is important to note that the entropy function being monotone submodular has been highly validated via an extensive body of work, such as (Krause et al. (2008) ).
>
> Finally, it is crucial to emphasize that, in our experiments, we adopted the same approximation for entropy as detailed in  (Xu et al., 2023), assuming that the information acquired about each moving target is inversely proportional to the distance between the agent and the target. Despite this simplification, the objective remains a monotone submodular function.
>
> -------------------
> ``Weaknesses``:
>
> --------------------
> > **1. Problem formulation**
> >*  I think this section is a bit unconnected with the motivating example. I would appreciate a bit more introduction about why and how problems like multi-agent tracking can be formulated as online submodular maximization problems. For example, I am a bit confused here why the objective function is revealed after agents make decisions and why the objective function is guaranteed to be submodular?
>
> Thank you for your comments.
>
> To address your concerns, let's first revisit the multi-agent tracking scenarios. Specifically, at each time step $t\in[T]$, agents firstly select their directions of movement and speeds. Subsequently, the effectiveness of the agents' movement at time $t$ is evaluated based on the positions of targets at time $t+1$. Since the location information of the targets at time $t+1$ is not known in advance, we normally assume that the objective function is revealed after the agents have made their decisions.
>
> Regarding your question on why the objective function is guaranteed to be submodular, we refer you to our previous response to the fourth question.
>
> > **1. Problem formulation**
> > * I think the author(s) should list the assumptions made in this section explicitly, such as those in lines 177 and 159.
>
> Thank you for your suggestion.
> Initially, we had considered explicitly listing the constraints on the feedback of agents (Lines 177-178) as a distinct assumption. However, due to space limitations, we ultimately had to incorporate them within the Problem Formulation section.  In response to your feedback, we are considering the addition of a new "Assumption"   immediately following Line 181.

---

> > ### Author Response · Authors · 2024-11-21
> >
> > > **2. Experiments:**
> > > Is all the targets being strictly homogeneous in type a requirement from the algorithm? What if different targets are of different types?
> >
> > Thank you for your feedback.
> >
> > Firstly, I would like to clarify that our paper employs an online optimization framework (Hazan et al.,2016) to model our submodular coordination problem. Typically, the online optimization framework can accommodate highly adversarial and inconsistent environments, which means that our Algorithm 1 and Algorithm 2 do not require targets to be strictly homogeneous.
> >
> > Secondly, in scenarios where different targets are of different types, our algorithms still can achieve a $\frac{(1-e^{-c})}{c}$-approximation to the optimal decision at a regret bound of $\widetilde{O}(\sqrt{\frac{C_{T}T}{1-\beta}})$. However, there is "no free lunch". If we permit all targets to flee at very high speeds, the tracking performance of the optimal decision even will not be very good.
> >
> > ------------
> > >**3. Contribution: this work builds upon a few existing methods in the literature. The novelty of the approaches can be better specified; cf. Question 3.**
> >
> > Thank you for your question. Regarding the contribution and novelty, please refer to our previous response to the Question 3.
> >
> > ------------
> > > **4. While submodular optimization has a variety of applications, the problem studied in this work seems to be more specific to robotics applications. I think is relatively less relevant to this venue.**
> >
> > Thank you for your feedback.
> >
> > Firstly, we would like to emphasize that robotics is a critical application domain for both reinforcement learning and online optimization. Moreover, the ICLR/NeurIPS/ICML community has indeed published a substantial body of work on robotic learning and submodular maximization. Therefore, we believe that our paper is highly relevant to the venue of learning.
> >
> > Secondly, beyond multi-robot coordination, the multi-agent submodular maximization framework also encompasses "distributed data selection tasks" (Mirzasoleiman et al., 2016. In this context, each agent possesses a different dataset, such as images, and the goal of  agents is to collaboratively select a subset that can effectively represent the entire dataset.
> >
> > ----------------
> > >**5. Other minor points:**
> > >* Line 70: “Furthermore” – hyperlink is redundant
> > >* Figure 3: It can be shown in the caption or in the figure that 3 rows represent different target types. Right now, it’s a bit hidden in the text.
> >
> > Thank you for your valuable suggestions. We will address the two minor points you've raised in the revised version of the manuscript.
> >
> > -------------------------

---

> > > ### Comment · Reviewer_H5Ct · 2024-11-23
> > > **Thank You for Your Answers**
> > >
> > > Dear Authors,
> > >
> > > Thank you for your answers and clarification!

---

### Author Response · Authors · 2024-11-21
**References**

Adibi et al. (2022). Minimax optimization: The case of convex-submodular. In International Conference on Artificial Intelligence and Statistics (AISTATS), pp. 3556–3580. PMLR.

Aslay et al. (2017). Revenue maximization in incentivized social advertising. Proc. VLDB Endow., 10(11):1238–1249, August 2017.

Besbes et al. (2015). Non-stationary stochastic optimization. Operations research, 63(5):1227–1244, 2015.

Bian et al. (2017). Guarantees for greedy maximization of non-submodular functions with applications. In International conference on machine learning (ICML), pp. 498–507. PMLR.

Bian et al. (2020). Continuous Submodular Function Maximization. arXiv preprint arXiv:2006.13474.

Calinescu et al. (2011). Maximizing a monotone submodular function subject to a matroid constraint. SIAM Journal on Computing, 40(6):1740–1766, 2011.

Chekuri et al. (2014). Submodular function maximization via the multilinear relaxation and contention resolution schemes. SIAM Journal on Computing, 43(6): 1831–1879, 2014.

Chen et al. (2018a). Projection-Free Online Optimization with Stochastic Gradient: From Convexity to Submodularity. In International Conference on Machine Learning (ICML), pp. 814–823. PMLR.

Chen et al. (2018b). Online Continuous Submodular Maximization. In International Conference on Artificial Intelligence and Statistics (AISTATS), pp. 1896–1905. PMLR.

Chow et al. (2016). Expander graph and communication-efficient decentralized optimization. In 2016 50th Asilomar Conference on Signals, Systems and Computers, pp. 1715–1720. IEEE.

Corah and Michael (2021). Scalable Distributed Planning for Multi-Robot, Multi-Target Tracking. In 2021 IEEE/RSJ International Conference on Intelligent Robots and Systems (IROS), pp. 437–444. IEEE.

El-Arini et al. (2009). Turning down the noise in the blogosphere. In Proceedings of the 15th ACM SIGKDD international conference on Knowledge discovery and data mining, pp. 289–298.

Fisher et al. (1978). An analysis of approximations for maximizing submodular set functions—II. In Polyhedral Combinatorics, pp. 73–87. Springer.

Grimsman et al. (2018). The impact of information in distributed submodular maximization. IEEE Transactions on Control of Network Systems, 6(4):1334–1343.

Han et al. (2021). Efficient and Effective Algorithms for Revenue Maximization in Social Advertising. In Proceedings of the 2021 international conference on management of data, pp. 671–684.

Hassani et al. (2017). Gradient methods for submodular maximization. In Advances in Neural Information Processing Systems, pp. 5841–5851.

Hazan et al. (2016). Introduction to online convex optimization. Foundations and Trends® in Optimization, 2(3-4):157–325.

Kempe et al. (2003). Maximizing the spread of influence through a social network. In Proceedings of the ninth ACM SIGKDD International Conference on Knowledge Discovery and Data Mining, pp. 137–146.

Krause et al. (2008). Near-optimal sensor placements in gaussian processes: Theory, efficient algorithms and empirical studies. Journal of Machine Learning Research, 9(2).

Kumari et al. (2024). An end-to-end submodular framework for data-efficient in-context learning. In Findings of the Association for Computational Linguistics: NAACL 2024, pp. 3293–3308.

Lin and Bilmes (2010). Multi-document summarization via budgeted maximization of submodular functions. In Human Language Technologies: The 2010 Annual Conference of the North American Chapter of the Association for Computational Linguistics, pp. 912–920.

Lin and Bilmes (2011). A class of submodular functions for document summarization. In Proceedings of the 49th Annual Meeting of the Association for Computational Linguistics: Human Language Technologies, pp. 510–520.

Mirzasoleiman et al. (2016a). Fast constrained submodular maximization: Personalized data summarization. In International Conference on Machine Learning, pp. 1358–1367. PMLR.

Mirzasoleiman et al. (2016b). Distributed submodular maximization. The Journal of Machine Learning Research, 17(1):8330–8373.

Mokhtari et al. (2018). Decentralized submodular maximization: Bridging discrete and continuous settings. In International conference on machine learning, pp. 3616–3625. PMLR.

Nedi´c and Olshevsky (2014). Distributed optimization over time-varying directed graphs. IEEE Transactions on Automatic Control, 60(3):601–615.

Nedic and Ozdaglar (2009). Distributed subgradient methods for multi-agent optimization. IEEE Transactions on Automatic Control, 54(1):48–61.

Nedi´c et al. (2017). Achieving geometric convergence for distributed optimization over time-varying graphs. SIAM Journal on Optimization, 27(4):2597–2633.

Nemhauser et al. (1978). An analysis of approximations for maximizing submodular set functions—I. Mathematical Programming, 14(1):265–294.

Qu et al. (2019). Distributed greedy algorithm for multi-agent task assignment problem with submodular utility functions. Automatica, 105:206–215.

---

> ### Author Response · Authors · 2024-11-21
> **References**
>
> Rezazadeh and Kia (2023). Distributed strategy selection: A submodular set function maximization approach. Automatica, 153:111000.
>
> Schlotfeldt et al. (2021). Resilient active information acquisition with teams of robots. IEEE Transactions on Robotics, 38(1):244–261.
>
> Song et al. (2022). Communication-efficient topologies for decentralized learning with o(1) consensus rate. Advances in Neural Information Processing Systems, 35:1073–1085.
>
> Streeter et al. (2009). Online learning of assignments. Advances in neural information processing systems, 22.
>
> Vogels et al. (2023). Beyond spectral gap: the role of the topology in decentralized learning. Journal of Machine Learning Research, 24(355):1–31.
>
> Vondrák (2010). Submodularity and curvature: The optimal algorithm (combinatorial optimization and discrete algorithms). RIMS Kokyuroku Bessatsu B, 23:253–266.
>
> Wei et al. (2013). Using document summarization techniques for speech data subset selection. In Proceedings of the 2013 Conference of the North American Chapter of the Association for Computational Linguistics: Human Language Technologies, pp. 721–726.
>
> Wei et al. (2015). Submodularity in data subset selection and active learning. In International conference on machine learning, pp. 1954–1963. PMLR.
>
> Xu et al. (2023). Online submodular coordination with bounded tracking regret: Theory, algorithm, and applications to multi-robot coordination. IEEE Robotics and Automation Letters, 8(4):2261–2268.
>
> Zhang et al. (2022). Stochastic continuous submodular maximization: Boosting via non-oblivious function. In International Conference on Machine Learning, pp. 26116–26134. PMLR.
>
> Zhang et al. (2023). Communication-efficient decentralized online continuous dr-submodular maximization. In Proceedings of the 32nd ACM International Conference on Information and Knowledge Management, pp. 3330–3339.
>
> Zhang et al. (2024). Boosting gradient ascent for continuous dr-submodular maximization. arXiv preprint arXiv:2401.08330.
>
> Zhou et al. (2018). Resilient active target tracking with multiple robots. IEEE Robotics and Automation Letters, 4(1):129–136.
>
> Zhou et al. (2022). Probabilistic bilevel coreset selection. In International Conference on Machine Learning, pp. 27287–27302.
>
> Zhu et al. (2021). Projection-free decentralized online learning for submodular maximization over time-varying networks. Journal of Machine Learning Research, 22(51):1

---

### Meta-Review · Area_Chair_GnNh · 2024-12-17

**Metareview:**

The reviewers are unanimously positive about the intellectual contributions of this work. I have reviewed the manuscript and it provides a strong contribution to the theory of discrete optimization in the multi-agent setting, resolving some fundamental complexity barriers of prior art related to evaluation of projections. Its practical utility is also apparent in some applications. For these reasons, I recommend this paper be accepted.

**Additional Comments On Reviewer Discussion:**

The authors have substantively resolved all reviewer criticism of the paper and the reviewers have acknowledged this resolution. Moreover, a revision of the paper has been uploaded which incorporates these enhancements.

---

### Decision · Program_Chairs · 2025-01-22

Accept (Spotlight)